# Reasoning-Enhanced Healthcare Predictions with Knowledge Graph Community Retrieval

**Pengcheng Jiang**[*] **Cao Xiao**[†] **Minhao Jiang**[†] **Parminder Bhatia**[†]
**Taha Kass-Hout**[†] **Jimeng Sun**[*] **Jiawei Han**[*]

[*]UIUC  [†]GE HealthCare

## Abstract

Large language models (LLMs) have demonstrated significant potential in clinical decision support. Yet LLMs still suffer from hallucinations and lack fine-grained contextual medical knowledge, limiting their high-stake healthcare applications such as clinical diagnosis. Traditional retrieval-augmented generation (RAG) methods attempt to address these limitations but frequently retrieve sparse or irrelevant information, undermining prediction accuracy. We introduce KARE, a novel framework that integrates knowledge graph (KG) community-level retrieval with LLM reasoning to enhance healthcare predictions. KARE constructs a comprehensive multi-source KG by integrating biomedical databases, clinical literature, and LLM-generated insights, and organizes it using hierarchical graph community detection and summarization for precise and contextually relevant information retrieval. Our key innovations include: (1) a dense medical knowledge structuring approach enabling accurate retrieval of relevant information; (2) a dynamic knowledge retrieval mechanism that enriches patient contexts with focused, multi-faceted medical insights; and (3) a reasoning-enhanced prediction framework that leverages these enriched contexts to produce both accurate and interpretable clinical predictions. Extensive experiments demonstrate that KARE outperforms leading models by up to 10.8-15.0% on MIMIC-III and 12.6-12.7% on MIMIC-IV for mortality and readmission predictions. In addition to its impressive prediction accuracy, our framework leverages the reasoning capabilities of LLMs, enhancing the trustworthiness of clinical predictions.

## 1 Introduction

Large language models (LLMs) (Touvron et al., 2023a;b; OpenAI et al., 2024; Team et al., 2024) has revolutionized natural language processing, offering unprecedented capabilities in understanding and generating human-like text. In the healthcare domain, LLMs hold the potential to transform clinical decision-making by providing insights derived from vast amounts of medical data (Wornow et al., 2023; Yang et al., 2022). There has been many recent explorations on applying ML-based methods in healthcare domain (Choi et al., 2016; 2017; Shickel et al., 2018; Choi et al., 2018; Ma et al., 2020a; Gao et al., 2020; Zhang et al., 2021; Wu et al., 2023; Jiang et al., 2024a; Zhu et al., 2024a; Xu et al., 2024). However, deploying LLMs in clinical settings presents significant challenges, mainly because LLMs may produce hallucinations or incorrect information due to a lack of specialized medical knowledge. Traditional retrieval-augmented generation (RAG) techniques (Lewis et al., 2021), which aim to mitigate hallucinations by retrieving external knowledge, often fall short in healthcare applications. They tend to retrieve information that, while semantically similar in latent space, fails to provide meaningful clinical insights, leading to suboptimal outcomes for precise healthcare predictions (Shi et al., 2024; Magesh et al., 2024; Li et al., 2024). For instance, when dealing with the diagnosis of heart failure, a traditional RAG model might retrieve data on several conditions that are semantically similar, such as "acute coronary syndrome" or "ischemic heart disease" due to their close proximity in latent space. However, these conditions, while related, do not capture the specific nuances of heart failure, such as the impact of left ventricular ejection fraction or specific biomarkers like NT-proBNP levels.

Knowledge graphs (KGs) offer a promising solution by providing structured representations of medical knowledge, capturing complex relationships between clinical entities (Liu et al., 2019; Yasunaga

et al., 2022; Zhang et al., 2022). Integrating KGs with LLMs can enhance the models' reasoning capabilities and provide domain-specific knowledge essential for accurate healthcare predictions (Soman et al., 2024). However, previous studies have often lacked interpretability and failed to fully leverage the reasoning strengths of LLMs (Jiang et al., 2024a; Xu et al., 2024; Zhu et al., 2024a).

Graph community retrieval has been a proven technique in various domains, such as social network analysis (Fortunato, 2010; Jin et al., 2021) and recommendation systems (Salha et al., 2019) for efficiently extracting relevant and contextual information from large-scale graphs. Recent work like GraphRAG (Edge et al., 2024) has demonstrated the superior performance of graph community retrieval compared to naïve RAG in the query-focused summarization task. However, the application of graph retrieval for LLM-based healthcare prediction remains largely unexplored.

In this paper, we introduce KARE (**K**nowledge **A**ware **R**easoning-**E**nhanced Health**Care** Prediction), a new framework that combines KG community-level retrieval (e.g., retrieving relevant subgraphs) with LLM reasoning to improve healthcare prediction.

Our technical contributions can be summarized as follows:

1. **Multi-Source Medical Knowledge Structuring and Indexing:** We develop a novel method to construct and index multi-source medical concept KGs by integrating concept-specific knowledge derived from relationships among different concepts in patients' electronic health records (EHRs). We employ hierarchical graph community detection and summarization techniques to organize the KG into semantically meaningful communities, facilitating precise, fine-grained, and contextually relevant information retrieval.

2. **Context Augmentation with Dynamic Knowledge Retrieval from KG:** We propose a context augmentation technique that can dynamically enrich patient data with knowledge from relevant KG communities tailored to the patient context. By retrieving pre-summarized communities, we enrich the input to the LLMs with focused, multi-faceted medical insights, addressing the limitations of traditional RAG methods.

3. **Reasoning-Enhanced Clinical Prediction Framework:** We leverage the augmented patient context to enable LLMs to generate step-by-step reasoning chains, enhancing both interpretability and prediction accuracy in clinical tasks.

To evaluate the KARE framework, we conducted experiments on in-hospital mortality and hospital readmission prediction tasks using the MIMIC-III and MIMIC-IV datasets(Johnson et al., 2016; 2020). KARE significantly outperforms the best baseline models. Specifically, KARE achieves improvements over best baselines up to 10.8%, 15.0%, 12.6%, and 12.7% on the MIMIC-III mortality, MIMIC-III readmission, MIMIC-IV mortality, and MIMIC-IV readmission prediction tasks, respectively. By attaining higher prediction accuracy and leveraging reasoning capabilities, KARE enhances the trustworthiness of clinical decision support systems. The reasoning process incorporates valuable evidence from relevant medical knowledge, facilitating more informed and explainable predictions that are needed in clinical decision making.

## 2 Related Works

**Clinical Predictive Models.** Electronic health record (EHR) data have become invaluable in the medical field, supporting predictive tasks aimed at improving patient care and clinical outcomes (Cai et al., 2016; Ashfaq et al., 2019; Bhoi et al., 2021). The development of deep learning models (Hochreiter & Schmidhuber, 1997; Chung et al., 2014; Vaswani et al., 2017) has enabled researchers to capture complex patterns within structured EHR data. Models such as RETAIN (Choi et al., 2016), GRAM (Choi et al., 2017), and others (Nguyen et al., 2016; Choi et al., 2018; Ma et al., 2020a;b; Gao et al., 2020; Zhang et al., 2021; Yang et al., 2023b) have shown promise in various predictive tasks. However, traditional predictive models are often inflexible, requiring specific labeled training data and struggling to generalize beyond their original scope. This limitation is particularly problematic in the dynamic healthcare environment. To address this, there is growing interest in using LLMs for clinical predictive tasks. LLMs offer greater adaptability and potential to interpret diverse medical information, including unstructured text and knowledge graphs enabling more robust and versatile clinical decision support systems.

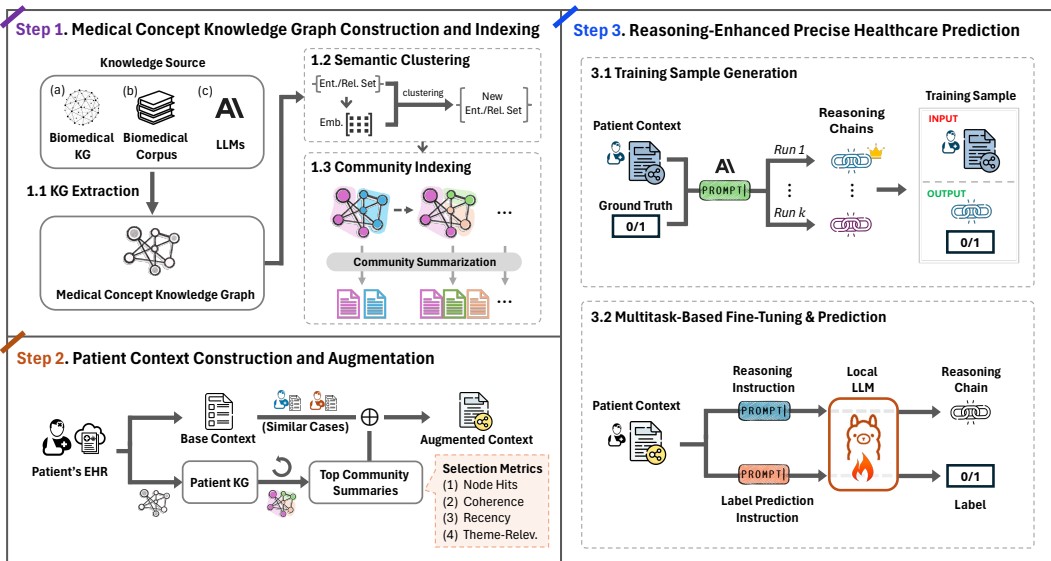

Figure 1: **A conceptual illustration of our KARE framework. Step 1** constructs a comprehensive medical concept knowledge graph by integrating information from multiple sources, organizing it into a hierarchical community structure. This structure allows for the generation of community summaries that facilitate precise knowledge retrieval. **Step 2** dynamically augments the patient's EHR context with relevant summaries from the knowledge graph, offering the LLM focused and relevant medical insights. **Step 3** generates training samples by employing an expert LLM to create reasoning chains based on the augmented patient context and ground truth labels. It then fine-tunes a local LLM using a multitask learning approach to produce interpretable reasoning chains and accurate predictions. By combining knowledge retrieval with LLM-driven reasoning, KARE significantly enhances the accuracy and reliability of clinical predictions.

**LLMs for Healthcare Predictions.** LLMs have revolutionized healthcare applications due to their advanced language understanding and generation capabilities (Xu et al., 2023; Kim et al., 2024; Bedi et al., 2024; Denecke et al., 2024). Recent works like MedRetriever (Ye et al., 2021), Graph-Care (Jiang et al., 2024a), RAM-EHR (Xu et al., 2024), EHR-KnowGen (Niu et al., 2024b), and EMERGE (Zhu et al., 2024a) have attempted to inject knowledge from retrieved literature or LLMs into patient representations, but they still lack interpretability and do not fully exploit the reasoning capabilities of LLMs. On the other hand, when directly applied to domain-specific tasks like EHR prediction, LLMs can produce significant errors and hallucinations due to the lack of integration of specialized domain knowledge (Zhu et al., 2024b; Xu et al., 2024; Cui et al., 2024; Shi et al., 2024; Chen et al., 2024). Recent works like KG-RAG (Soman et al., 2023) demonstrate the broader value of KG integration with LLMs in biomedical applications. Therefore, our work integrates KG community indexing and dynamic graph retrieval, compared to traditional RAG (Lewis et al., 2021; Niu et al., 2024a) and KGs, to construct and query fine-grained, precise knowledge, enhancing patient context. Furthermore, existing LLM-based methods often fail to fully harness the inherent reasoning capabilities of LLMs. Recent efforts (Cui et al., 2024; Shi et al., 2024) explored agentic frameworks for EHR prediction but rely on prompting that does not deeply engage with underlying EHR data patterns, resulting in suboptimal performance. Our approach distinguishes itself by fine-tuning a specialized, smaller LLM that incorporates reasoning abilities distilled from larger models.

# 3    KARE: KNOWLEDGE AWARE REASONING-ENHANCED FRAMEWORK

Our KARE framework (Figure 1) aims to improve healthcare predictions by combining relevant medical knowledge along with reasoning capabilities with LLMs. The following steps achieve this: (1) medical concept knowledge graph construction and indexing, (2) patient context construction and augmentation, and (3) reasoning-enhanced precise healthcare prediction.

## 3.1    STEP 1: MEDICAL CONCEPT KNOWLEDGE GRAPH CONSTRUCTION AND INDEXING

***Objective of Step 1*** is to create a medical knowledge base that is specifically tailored to electronic health record (EHR) data. Unlike most existing medical knowledge graphs, which are static and not

connected to the EHR data, KARE dynamically generates a high-quality knowledge base that can be used for retrieving and predicting information in later stages.

### 3.1.1 MEDICAL CONCEPT-SPECIFIC KNOWLEDGE GRAPH EXTRACTION

For each medical concept $c_i$ in the EHR coding system, we extract a concept-specific knowledge graph $G_{c_i} = (V_{c_i}, E_{c_i})$ globally tailored to the EHR datasets from three sources:

(a) **Biomedical KG** (e.g., UMLS (Bodenreider, 2004)): For each medical concept $c_i$ in EHR data, we extract a subgraph $G_{c_i}^{\text{KG}}$ by first iterating through the patient EHR dataset to collect the top $X$ co-existing concepts appeared in each patient's data, forming the set of related concepts $R_{c_i}$. We then find the shortest path $p_{ij}$ in the KG for each pair $(c_i, c_j)$ in $R_{c_i}$, with a specified maximum path length. $G_{c_i}^{\text{KG}} = (V_{c_i}^{\text{KG}}, E_{c_i}^{\text{KG}})$ is constructed by combining all these shortest paths, where $V_{c_i}^{\text{KG}}$ and $E_{c_i}^{\text{KG}}$ are the union of nodes and edges in all $p_{ij}$, respectively. See more details in Appendix B.1.

(b) **Biomedical Corpus** (e.g., PubMed (Canese & Weis, 2013)): We iterate through the EHR dataset and, for each visit of the patients, and collect all the involved medical concepts. We then retrieve the top $n$ documents from the corpus based on these medical concepts. For each retrieved document, we perform entity extraction and relation extraction to extract KG triples. The extracted triples are then added to the KG of the medical concepts mentioned in the document. By doing so, $G_{c_i}^{\text{BC}}$ is built for each concept $c_i$. We showcase more details in Appendix B.2.

(c) **LLMs**: We iterate through the EHR dataset and prompt the LLM to identify the relationships among the concepts that are helpful to the clinical predictions, where we allow the LLM to add intermediate relationships within two concepts. The process is detailed in Appendix B.3.

The final concept-specific KG $G_{c_i}$ is the union of the subgraphs from each source:

$$G_{c_i} = G_{c_i}^{\text{KG}} \cup G_{c_i}^{\text{BC}} \cup G_{c_i}^{\text{LLM}} \tag{1}$$

Finally, we integrate all concept-specific KGs for the medical concepts in our EHR coding system. The resulting knowledge graph $G' = (V', R', E')$ is defined as $G' = \bigcup_{c_i \in \mathbf{C}} G_{c_i}$ where $\mathbf{C}$ is the set of all medical concepts in the specified EHR coding system.

*Note*: Different from the KG construction method introduced by GraphCare (Jiang et al., 2024a), which retrieves sparse and random relationships from LLMs and biomedical KGs, our approach utilizes the EHR dataset to anchor the relevant relationships and interactions among the medical concepts present in patient data. This targeted focus allows us to construct a more relevant and context-tailored KG for clinical predictions.

### 3.1.2 SEMANTIC CLUSTERING

Semantic clustering in our KG addresses the challenge of differently named entities and relations from various sources that may refer to the same concept. We employ agglomerative clustering (Müllner, 2011) with an automatically determined optimal threshold. First, we generate text embeddings $\mathbf{e}_i = \text{TextEmbed}(v_i)$ for each entity $v_i \in V'$ and $\mathbf{e}_j = \text{TextEmbed}(r_j)$ for each relation $r_j \in R'$ using an LLM. To determine the optimal clustering thresholds $\theta_e$ and $\theta_r$ for entities and relations, we refer to the silhouette score (Shahapure & Nicholas, 2020; Jiang et al., 2024b), which considers both intra-cluster similarity and inter-cluster dissimilarity. We sample a subset of entities and relations, perform agglomerative clustering with varying distance thresholds, and select those yield the highest scores. We then cluster all entities and relations using their respective optimal thresholds. Each cluster is represented by the element closest to the cluster center, determined by the average embedding of all elements within the cluster. We create mappings $\phi_e : V' \to V$ and $\phi_r : R' \to R$ between the original entities/relations and their cluster representatives. Each triple $(h', r', t')$ in the original KG is mapped to its corresponding cluster representatives $(h, r, t) = (\phi_e(h'), \phi_r(r'), \phi_e(t'))$, resulting in a refined knowledge graph $G = (V, R, E)$.

### 3.1.3 HIERARCHICAL KG COMMUNITY DETECTION AND INDEXING

We organize the refined knowledge graph (KG) into a hierarchical structure of communities using the Leiden algorithm (Traag et al., 2019). This is done at multiple levels of granularity, from coarse to fine. We run the algorithm multiple times with different randomness parameters to explore diverse

community structures, and generate multi-theme summaries for each community, providing a more comprehensive understanding of the knowledge contained withtin the KG.

To keep computation manageable, we limit the maximum number of triples per community ($Z_c$) and the maximum number of triples per initial summary ($Z_s$).

For each community, we generate two types of summaries using an LLM (prompts used: Figure 12):

- *General summary*: A concise summarization of the medical concepts and relationships in the community without focusing on any specific theme.

- *Theme-specific summary*: A summary that highlights how the knowledge in the community relates to a specific theme (e.g., mortality prediction), if relevant. Figure 14 shows an example.

The summarization process depends on the community size:

- For small communities (size $\leq Z_s$), we directly summarize all triples.

- For large communities ($Z_s <$ size $\leq Z_c$), we shuffle and split the triples into subsets, summarize each subset, and then iteratively aggregate the summaries until we get a single comprehensive summary (prompt shown in Figure 13).

- For extremely large communities (size $> Z_c$), we do not generate summaries due to the limit of the LLM context window.

As we move up the hierarchy from fine to coarse levels, triples from small communities get merged into larger ones, which are then summarized using the same process.

The result is a hierarchical structure of communities at different granularities, each with theme-specific summaries. Running the Leiden algorithm multiple times with different randomness parameter gives us diverse communities, allowing entities to contribute to multiple summaries. This rich, multi-level representation of the KG is the foundation for later steps.

## 3.2 STEP 2: PATIENT CONTEXT CONSTRUCTION AND AUGMENTATION

***Objective of Step 2:*** This step constructs patient's EHR context with the highly relevant and fine-grained medical knowledge attached.

**Base Context Construction.** For a patient $p$, we construct a base context $\mathcal{B}_p$ with their EHR data with a standardized template. This context focuses on (1) task description, (2) the patient's conditions, procedures, and medications across different visits, and (3) similar patients to the target patient. For (3), two most similar patients are retrieved from the reference set (i.e., training data) based on the EHR similarity where one has the same label as patient $p$ and the other has a different label (Cui et al., 2024). Figure 11 shows an example of the base context and the template used.

**Context Augmentation.** To enrich the patient's base context with relevant information from the knowledge graph, we first construct a patient-specific knowledge graph $G_p$ by aggregating the concept-specific graphs $G_{c_i}$ (defined in Eq. 1) for all medical concepts $c_i$ in the patient's EHR, using the mappings $\phi_e$ and $\phi_r$ from §3.1.2:

$$G_p = \cup_{c_i \in \text{EHR}_p} \{\phi_e(h), \phi_r(r), \phi_e(t) \mid (h, r, t) \in G_{c_i}\} \tag{2}$$

From $G_p$, we derive two sets of nodes: $V_p^{\text{direct}}$, representing medical concepts that directly appear in the patient's EHR, and $V_p^{\text{indirect}}$, containing the remaining nodes in $G_p$.

We then introduce a combined relevance score for each community $C_k$ to select the most relevant summaries for context augmentation:

$$\begin{aligned} \text{Relevance}(C_k) = &\left(\mathcal{H}(C_k, V_p^{\text{direct}}) + \alpha \cdot \mathcal{H}(C_k, V_p^{\text{indirect}})\right) \times \text{Decay}(C_k, V_p^{\text{direct}}) \\ &\times \text{Coherence}(S_{C_k}, \mathcal{B}_p) \times \text{Recency}(C_k, V_p^{\text{direct}}) \times \text{ThemeRel}_\tau(C_k) \end{aligned} \tag{3}$$

In Eq. 3, $\mathcal{H}(C_k, V_p^{\text{direct}})$ and $\mathcal{H}(C_k, V_p^{\text{indirect}})$ calculate the normalized counts of direct and indirect node hits by comparing the nodes in community $C_k$ with the corresponding sets of direct and indirect nodes. The parameter $\alpha \in [0, 1)$ weights the importance of indirect hits relative to direct hits. The

decay function $\text{Decay}(C_k, V_p^{\text{direct}})$ reduces the contribution of previously hit nodes in community $C_k$ by a factor $\beta^{H(v)}$, where $\beta \in (0, 1]$ is a decay constant and $H(v)$ is the hit count of node $v$ in previous selections, considering only the direct nodes $V_p^{\text{direct}}$. Additional factors are defined as:

$$\text{Coherence}(S_{C_k}, \mathcal{B}_p) = 1 + \lambda_1 \cdot \cos\left(e(S_{C_k}), e(\mathcal{B}_p)\right) \tag{4}$$

$$\text{Recency}(C_k, V_p^{\text{direct}}) = 1 + \lambda_2 \cdot \frac{\sum_{v \in V_{C_k} \cap V_p^{\text{direct}}} \text{visit}(v)}{|V_{C_k} \cap V_p^{\text{direct}}|} \tag{5}$$

$$\text{ThemeRel}_\tau(C_k) = 1 + \frac{\lambda_3}{|V_{C_k}|} \sum_{v \in V_{C_k}} \max_{z \in \mathcal{T}_\tau} \cos\left(e(v), e(z)\right) \tag{6}$$

Here, $e(\cdot)$ denotes a text embedding function, $\cos(\cdot, \cdot)$ is the cosine similarity between embeddings, $\text{visit}(v)$ returns the visit index of node $v$, and $\lambda_1, \lambda_2, \lambda_3 \in [0, 1]$ control the weights of the metrics. The set $\mathcal{T}_\tau$ contains representative terms for the theme $\tau$ (e.g., {*end-stage*, *life-threatening*, ...} for mortality prediction), same as those used for *attention initialization* in (Jiang et al., 2024a).

The proposed metrics in Eq. 3 serve different purposes: node hits $\mathcal{H}$ ensure specificity to the patient's conditions, decay factor promotes diversity, coherence aligns the selected summaries with the patient's overall context, recency prioritizes more recent information, and theme relevance maintains task-oriented selection. In addition, we propose a Dynamic Graph Retrieval and Augmentation (DGRA) method to iteratively select the most relevant summaries to augment the patient's context. At each iteration, it performs as:

(1) Compute the relevance scores for all candidate communities $C_k \in \mathcal{C}$ using Eq. 3.

(2) Identify the community $C_{\text{best}}$ with the highest relevance score and add its summary $S_{C_{\text{best}}}$ to the set of selected summaries $S_p$.

(3) Increment the hit count $H(v)$ for each node $v$ in $V_{C_{\text{best}}}$, which will impact the decay in future relevance calculations.

(4) Remove $C_{\text{best}}$ from the candidate communities $\mathcal{C}$, ensuring it is not reconsidered in subsequent iterations.

---

**Algorithm 1** Dynamic Graph Retrieval and Augmentation

---

**Input:** Set of communities $\mathcal{C}$, patient graph $G_p$, base context $\mathcal{B}_p$, desired number of summaries $N$
**Output:** Augmented patient context $\mathcal{A}_p$
Initialize $S_p \leftarrow \emptyset$
Initialize hit counts $H(v) \leftarrow 0$ for each node $v \in V_p^{\text{direct}}$
**while** $|S_p| < N$ **do**
  Compute Relevance$(C_k)$ for all $C_k \in \mathcal{C}$ using Eq. 3
    Select $C_{\text{best}} \leftarrow \arg\max_{C_k \in \mathcal{C}} \text{Relevance}(C_k)$
    Add $S_{C_{\text{best}}}$ to $S_p$: $S_p \leftarrow S_p \cup S_{C_{\text{best}}}$
    For each $v \in V_{C_{\text{best}}} \cap V_p^{\text{direct}}$, $H(v) \leftarrow H(v) + 1$
    Remove $C_{\text{best}}$ from $\mathcal{C}$: $\mathcal{C} \leftarrow \mathcal{C} \setminus C_{\text{best}}$
**end**
Augment patient context: $\mathcal{A}_p = \mathcal{B}_p \oplus S_p$
**return** $\mathcal{A}_p$

---

The process continues until $N$ summaries have been selected. The final augmented patient context $\mathcal{A}_p$ is obtained by concatenating the base context $\mathcal{B}_p$ with the selected summaries.

By dynamically updating the node hits and recalculating relevance scores at each iteration, we prioritize communities that contribute new and valuable information. This ensures that the augmented context includes the most relevant and diverse information from the KG, tailored to the patient's specific conditions and the prediction task.

## 3.3 STEP 3: REASONING-ENHANCED PRECISE HEALTHCARE PREDICTION

***Objective of Step 3:*** This step trains an LLM capable of predicting healthcare outcome while generating a reasoning process, using the augmented patient context constructed in the previous step.

### 3.3.1 TRAINING SAMPLE GENERATION

Inspired by recent rationale distillation approaches (Kang et al., 2024; Kwon et al., 2024; Jiang et al., 2024b), we employ an LLM to generate reasoning chains in a unified format for each patient $p$ and task $\tau$. This process involves entering (1) the task description $\mathcal{D}_\tau$ (e.g., Figure 15), (2) the augmented patient context $\mathcal{A}_p$, and (3) the corresponding ground truth label $y_{p,\tau}^*$ into the LLM. The specific prompt utilized for the reasoning chain (training sample) generation is showcased in Figure 16 in Appendix. The LLM generates $K$ reasoning chains $\rho_{p,\tau,k}$ along with confidence levels. We select

Table 1: Statistics of pre-processed EHR datasets. "#": "the number of", "/ patient": "per patient".

| | MIMIC-III-Mort. | | | MIMIC-III-Read. | | | MIMIC-IV-Mort. | | | MIMIC-IV-Read. | | |
|---|---|---|---|---|---|---|---|---|---|---|---|---|
| | Train | Valid | Test | Train | Valid | Test | Train | Valid | Test | Train | Valid | Test |
| # Patients (Samples) | 7730 | 991 | 996 | 7730 | 991 | 996 | 8018 | 996 | 986 | 8029 | 958 | 1013 |
| # Visits / Patient | 1.56 | 1.60 | 1.61 | 1.56 | 1.60 | 1.61 | 1.26 | 1.30 | 1.21 | 1.26 | 1.28 | 1.25 |
| # Conditions / Patient | 23.27 | 23.92 | 25.89 | 23.27 | 23.92 | 25.89 | 14.34 | 15.30 | 13.59 | 13.62 | 14.21 | 13.21 |
| # Procedures / Patient | 6.22 | 6.56 | 7.17 | 6.22 | 6.56 | 7.17 | 2.96 | 3.08 | 2.84 | 2.89 | 2.96 | 2.81 |
| # Medications / Patient | 54.79 | 55.77 | 63.73 | 54.79 | 55.77 | 63.73 | 30.66 | 32.86 | 28.40 | 28.74 | 30.61 | 27.59 |

the reasoning chain with the highest confidence, ensuring that only the most reliable explanations are used. The final set of training data for each patient-task pair is then $\{(\mathcal{D}_\tau, \mathcal{A}_p, \rho_{p,\tau}^{\text{best}}, y_{p,\tau}^*)\}$, where $\rho_{p,\tau}^{\text{best}}$ is the reasoning chain with the highest confidence level.

### 3.3.2 MULTITASK-BASED FINE-TUNING AND PREDICTION

We fine-tune a relatively small local LLM (e.g., a 7B-parameter model) to perform both reasoning chain generation and label prediction for each patient $p$ and healthcare prediction task $\tau$ (such as mortality or readmission prediction). The model is trained using inputs that consist of the task description $\mathcal{D}_\tau$ and the augmented patient context $\mathcal{A}_p$, with an prepended instruction indicating whether to generate a reasoning chain or predict the label. These inputs and outputs are formatted according to the templates shown in Figure 17 in the Appendix.

During fine-tuning, when instructed to generate a reasoning chain (with the prefix `[Reasoning]`), the model aligns its output with the reasoning chain $\rho_{p,\tau}^{\text{best}}$ obtained from the previous step. When instructed to predict the label (with the prefix `[Label Prediction]`), it aligns its output with the ground truth label $y_{p,\tau}^*$. We minimize the cross entropy loss across both tasks, encouraging the development of shared representations that enhance performance in both reasoning and prediction.

In the prediction phase, given a new patient $p_{\text{new}}$ and task $\tau$, we provide $\mathcal{A}_{p_{\text{new}}}, \tau$, and the appropriate instruction to the fine-tuned model. Based on the instruction, the model can either generate the reasoning chain $\rho_{p_{\text{new}},\tau}$ or predict the label $y_{p_{\text{new}},\tau}$. This flexible approach allows us to obtain detailed reasoning when necessary or perform efficient label prediction, leveraging the multitask training to effectively handle both tasks during inference.

## 4 EXPERIMENTS

### 4.1 EXPERIMENTAL SETTING

**Tasks.** In this work, we focus on the following EHR-based prediction tasks:

- *Mortality Prediction.* This task estimates mortality outcome for next visit, defined as $f : (x_1, x_2, \ldots, x_{t-1}) \to y[x_t]$, where $y[x_t] \in \{0, 1\}$ is patient's survival status during visit $x_t$.

- *Readmission Prediction.* This task predicts if patient will be readmitted within $\sigma$ days, defined as $f : (x_1, x_2, \ldots, x_{t-1}) \to y[\varphi(x_t) - \varphi(x_{t-1})]$, where $y \in \{0, 1\}$, $\varphi(x_t)$ is timestamp of visit $x_t$, and $y[\varphi(x_t) - \varphi(x_{t-1})] = 1$ if $\varphi(x_t) - \varphi(x_{t-1}) \le \sigma$, else 0. $\sigma$ is set to 15 in this study.

**Datasets.** We utilize the publicly available MIMIC-III (Johnson et al., 2016) (v1.4) and MIMIC-IV (Johnson et al., 2020) (v2.0) EHR datasets, and use PyHealth (Yang et al., 2023a) for preprocessing. MIMIC-III is processed (full set) using the same approach as GraphCare (Jiang et al., 2024a). For MIMIC-IV mortality prediction, we retain 2,152 patients with a label of 1 (mortality), excluding 54 patients with more than 10 visits. We then randomly (seed=42) sample unique patients with a label of 0, each having no more than 10 visits, until reaching a sample size of 10,000. For MIMIC-IV readmission prediction, we randomly (seed=42) select 5,000 unique patients with a label of 1 (will be readmitted) and 5,000 with a label of 0. Both datasets are split into training, validation, and test sets in a 0.8/0.1/0.1 ratio by patient, ensuring that all samples from the same patient are confined to a single subset, preventing data leakage. We use Clinical Classifications Software (CCS) for condition/procedure mappings and the Anatomical Therapeutic Chemical classification system at the third level (ATC3) for medication mapping, with the resulting statistics presented in Table 1.

**Evaluation Metrics.** We employ four standard binary classification metrics: (1) **Accuracy**, measuring overall correct predictions; (2) **Macro-F1**, providing a balanced measure for imbalanced

Table 2: Comparative analysis of mortality and readmission predictions using MIMIC-III and MIMIC-IV datasets. "pos" indicates the proportion of positive samples (label = 1) in the test set. Metrics like Macro F1 and Sensitivity are emphasized by an asterisk (*) due to their importance for handling imbalanced datasets. Results are averaged by multiple runs: 30 for ML-based, 10 for LM+ML-based, and 3 for LLM-based methods with different random seeds. We **highlight** the highest value for each metric.

| | | MIMIC-III | | | | | | | |
| | | Mortality Prediction (pos = 5.42%) | | | | Readmission Prediction (pos = 54.82%) | | | |
| Type | Models | Accuracy | Macro F1* | Sensitivity* | Specificity | Accuracy | Macro F1 | Sensitivity | Specificity |
|---|---|---|---|---|---|---|---|---|---|
| ML | GRU (Chung et al., 2014) | 92.7 | 50.7 | 3.7 | 97.8 | 62.2 | 61.5 | 68.9 | 54.0 |
| | Transformer (Vaswani et al., 2017) | 92.7 | 51.9 | 5.6 | 97.6 | 58.8 | 58.2 | 65.0 | 51.3 |
| | RETAIN (Choi et al., 2016) | 92.4 | 50.6 | 3.7 | 97.6 | 59.1 | 56.9 | 74.9 | 40.0 |
| | GRAM (Choi et al., 2017) | 92.4 | 50.2 | 5.2 | 95.2 | 61.8 | 60.4 | 74.9 | 46.4 |
| | Deepr (Nguyen et al., 2016) | 91.9 | 51.0 | 3.7 | 98.2 | 62.6 | 62.1 | 66.7 | 57.6 |
| | TCN (Bai et al., 2018) | 91.6 | 53.2 | 9.3 | 96.4 | 63.4 | 62.7 | 70.7 | 54.7 |
| | ConCare (Ma et al., 2020b) | 94.6 | 48.6 | 0.0 | **100.0** | 59.2 | 59.0 | 61.5 | 56.4 |
| | AdaCare (Ma et al., 2020a) | 90.6 | 54.1 | 9.1 | 97.6 | 61.6 | 60.5 | 70.8 | 50.3 |
| | GRASP (Zhang et al., 2021) | 93.7 | 49.9 | 1.9 | 98.9 | 61.3 | 59.5 | 74.9 | 44.8 |
| | StageNet (Gao et al., 2020) | 90.5 | 50.5 | 5.6 | 95.4 | 60.5 | 60.0 | 65.1 | 54.9 |
| | KerPrint (Yang et al., 2023b) | 92.4 | 52.2 | 9.8 | 94.7 | 63.5 | 62.1 | 68.0 | 56.1 |
| LM+ML | GraphCare (Jiang et al., 2024a) | 94.9 | 58.3 | 17.2 | 97.1 | 65.4 | 64.1 | 70.3 | 57.8 |
| | RAM-EHR (Xu et al., 2024) | 94.4 | 59.6 | 14.8 | 98.9 | 64.8 | 63.5 | 74.7 | 52.4 |
| | EMERGE (Zhu et al., 2024a) | 94.1 | 57.7 | 13.2 | 98.4 | 63.7 | 62.0 | 68.0 | 55.9 |
| LLM | Zero-shot (LLM: Claude 3.5 Sonnet) | | | | | | | | |
| | w/ EHR context only | 89.5 | 50.4 | 6.4 | 94.4 | 54.3 | 35.4 | 98.9 | 0.2 |
| | w/ Classic RAG[a] | 89.9 | 51.2 | 10.2 | 92.8 | 53.2 | 34.6 | 91.2 | 1.4 |
| | w/ KARE-augmented context[b] | 92.3 | 54.6 | 14.2 | 94.6 | 56.3 | 43.8 | 93.9 | 10.6 |
| | Few-Shot (LLM: Claude 3.5 Sonnet) | | | | | | | | |
| | w/ exemplar only (N=2)[c] | 88.7 | 49.5 | 5.6 | 93.4 | 52.7 | 42.2 | 87.0 | 11.1 |
| | w/ exemplar only (N=4) | 88.0 | 49.2 | 5.6 | 92.7 | 53.6 | 44.7 | 84.0 | 15.7 |
| | w/ EHR-CoAgent[d] (Cui et al., 2024) | 87.4 | 51.7 | 13.0 | 91.8 | 55.2 | 46.1 | 78.2 | 20.1 |
| | w/ KARE-augmented context | 91.5 | 53.5 | 13.7 | 94.0 | 57.1 | 49.3 | 75.5 | 27.2 |
| | Fine-tuned (LLM: Mistral-7B-Instruct-v0.3) | | | | | | | | |
| | Backbone | 90.4 | 53.0 | 11.4 | 94.3 | 57.6 | 57.6 | 50.5 | 66.3 |
| | w/ Classic RAG | 90.1 | 51.4 | 12.5 | 91.6 | 60.2 | 59.9 | 56.1 | 64.5 |
| | **KARE** (ours) | **95.3** | **64.6** | **24.7** | 98.3 | **73.9** | **73.7** | **76.7** | **70.7** |

| | | MIMIC-IV | | | | | | | |
| | | Mortality Prediction (pos = 19.16%) | | | | Readmission Prediction (pos = 46.50%) | | | |
| Type | Models | Accuracy | Macro F1* | Sensitivity* | Specificity | Accuracy | Macro F1 | Sensitivity | Specificity |
|---|---|---|---|---|---|---|---|---|---|
| ML | GRU (Chung et al., 2014) | 88.7 | 76.4 | 42.9 | 99.6 | 62.4 | 62.2 | 68.3 | 56.2 |
| | Transformer (Vaswani et al., 2017) | 83.7 | 71.3 | 47.1 | 92.3 | 61.3 | 61.3 | 63.0 | 59.5 |
| | RETAIN (Choi et al., 2016) | 84.8 | 73.8 | 52.4 | 92.4 | 62.8 | 62.6 | 68.7 | 56.6 |
| | GRAM (Choi et al., 2017) | 86.4 | 74.4 | 50.6 | 93.9 | 62.5 | 62.5 | 67.4 | 57.8 |
| | Deepr (Nguyen et al., 2016) | 89.2 | 78.9 | 50.8 | 98.2 | 59.2 | 59.2 | 57.0 | 61.5 |
| | TCN (Bai et al., 2018) | 89.9 | 79.2 | 47.6 | **99.9** | 63.6 | 63.5 | 72.2 | 56.1 |
| | ConCare (Ma et al., 2020b) | 89.8 | 78.9 | 47.1 | **99.9** | 59.8 | 59.8 | 63.5 | 56.6 |
| | AdaCare (Ma et al., 2020a) | 88.7 | 78.2 | 50.3 | 97.8 | 62.9 | 62.9 | 58.4 | 67.7 |
| | GRASP (Zhang et al., 2021) | 89.9 | 79.1 | 47.6 | 99.8 | 59.7 | 59.6 | 53.1 | 66.7 |
| | StageNet (Gao et al., 2020) | 88.1 | 77.8 | 51.9 | 96.7 | 62.8 | 62.7 | 62.6 | 62.9 |
| | KerPrint (Yang et al., 2023b) | 88.7 | 79.8 | 53.1 | 98.0 | 63.5 | 63.3 | 67.0 | 60.1 |
| LM+ML | GraphCare (Jiang et al., 2024a) | 91.5 | 80.3 | 57.8 | 96.6 | 65.7 | 65.5 | 66.2 | 65.0 |
| | RAM-EHR (Xu et al., 2024) | 90.5 | 78.4 | 52.6 | 97.0 | 65.5 | 65.5 | 64.0 | 67.0 |
| | EMERGE (Zhu et al., 2024a) | 90.7 | 78.3 | 53.4 | 96.6 | 63.3 | 63.2 | 61.5 | 64.9 |
| LLM | Zero-shot (LLM: Claude 3.5 Sonnet) | | | | | | | | |
| | w/ EHR context only | 80.5 | 47.0 | 2.7 | 98.7 | 49.4 | 45.7 | 81.8 | 21.5 |
| | w/ Classic RAG[a] | 81.0 | 49.9 | 8.1 | 94.6 | 49.0 | 44.2 | 83.2 | 18.8 |
| | w/ KARE-augmented context[b] | 83.2 | 54.3 | 12.7 | 96.3 | 52.3 | 49.7 | 80.6 | 27.7 |
| | Few-Shot (LLM: Claude 3.5 Sonnet) | | | | | | | | |
| | w/ exemplar only (N=2)[c] | 80.8 | 46.7 | 2.1 | 99.5 | 49.3 | 44.7 | 84.0 | 19.1 |
| | w/ exemplar only (N=4) | 81.6 | 49.9 | 5.3 | 99.8 | 49.0 | 44.1 | 84.3 | 18.2 |
| | w/ EHR-CoAgent[d] (Cui et al., 2024) | 81.0 | 55.5 | 13.8 | 97.0 | 51.2 | 46.3 | 78.4 | 24.0 |
| | w/ KARE-augmented context | 84.5 | 57.4 | 15.8 | 97.6 | 54.1 | 51.9 | 75.2 | 34.1 |
| | Fine-tuned (LLM: Mistral-7B-Instruct-v0.3) | | | | | | | | |
| | Backbone | 92.2 | 83.1 | 65.0 | 96.2 | 56.1 | 46.7 | 23.1 | **76.2** |
| | w/ Classic RAG | 92.5 | 83.8 | 63.2 | 99.8 | 58.8 | 46.7 | 56.7 | 57.5 |
| | **KARE** (ours) | **94.1** | **90.4** | **73.2** | 99.8 | **73.9** | **73.8** | **85.6** | 63.7 |

[a] We retrieve up to ten documents from 30 M PubMed abstracts that are most similar to the base context. Dense retrieval is applied with Nomic (Nussbaum et al., 2024) (dim=768).
[b] Context is augmented as described in Section 3.2 using our constructed KG communities. Up to ten community summaries are appended as supplementary information.
[c] N=2 means that we retrieve two most similar patients (one from positive and the other from negative samples), similar to the strategy introduced in (Cui et al., 2024).
[d] Since the code for EHR-CoAgent was not open-sourced at the time of our paper submission, we implemented it ourselves, as detailed in Appendix C.

datasets; (3) **Sensitivity**, quantifying the model's ability to identify high-risk patients; and (4) **Specificity**, assessing accuracy in identifying low-risk patients. Detailed discussion of metric selection and computation is provided in Appendix E.

**Baselines.** We compare to three categories of baselines: (1) ML-based methods: GRU (Chung et al., 2014), Transformer (Vaswani et al., 2017), RETAIN (Choi et al., 2016), GRAM (Choi et al., 2017), Deepr (Nguyen et al., 2016), TCN (Bai et al., 2018), StageNet (Gao et al., 2020), ConCare (Ma et al., 2020b), AdaCare (Ma et al., 2020a), GRASP (Zhang et al., 2021), and KerPrint (Yang et al., 2023b); (2) LM + ML-based methods: GraphCare (Jiang et al., 2024a), RAM-EHR (Xu et al., 2024), and EMERGE (Zhu et al., 2024a); and (3) LLM-based methods: zero-shot and few-shot prompting with the advanced LLM Claude 3.5 Sonnet (Anthropic, 2024), and a few-shot-based method EHR-CoAgent (Cui et al., 2024). We showcase the details of baseline implementation in Appendix C.

**Implementation Details.** We utilize Scikit-learn (Pedregosa et al., 2018) for agglomerative clustering and Graspy (Chung et al., 2019) for the hierarchical Leiden algorithm. Semantic similarity cal-

Table 3: Ablation study of fine-tuning components. Results are averaged by 3 runs with different seeds.

| Similar Patients | Retrieved Knowledge | Reasoning | MIMIC-III-Mortality | | | | MIMIC-III-Readmission | | | |
|---|---|---|---|---|---|---|---|---|---|---|
| | | | Accuracy | Macro F1 | Sensitivity | Specificity | Accuracy | Macro F1 | Sensitivity | Specificity |
| ✗ | ✗ | ✗ | 90.4 | 53.0 | 11.4 | 94.3 | 57.6 | 57.6 | 50.5 | 66.3 |
| ✗ | ✗ | ✓ | 93.1 | 58.4 | 15.8 | 97.5 | 65.5 | 64.7 | 62.3 | 67.7 |
| ✗ | ✓ | ✓ | 95.3 | 64.6 | 24.7 | 98.3 | 72.8 | 72.6 | 74.7 | 70.6 |
| ✓ | ✓ | ✓ | 93.6 | 61.3 | 18.4 | 98.6 | 73.9 | 73.7 | 76.7 | 70.7 |

| Similar Patients | Retrieved Knowledge | Reasoning | MIMIC-IV-Mortality | | | | MIMIC-IV-Readmission | | | |
|---|---|---|---|---|---|---|---|---|---|---|
| | | | Accuracy | Macro F1 | Sensitivity | Specificity | Accuracy | Macro F1 | Sensitivity | Specificity |
| ✗ | ✗ | ✗ | 92.2 | 83.1 | 65.0 | 96.2 | 56.1 | 46.7 | 23.1 | 76.2 |
| ✗ | ✗ | ✓ | 93.3 | 85.4 | 67.3 | 97.5 | 64.7 | 62.1 | 69.3 | 55.9 |
| ✗ | ✓ | ✓ | 93.8 | 89.6 | 74.5 | 98.8 | 72.2 | 71.9 | 81.1 | 64.0 |
| ✓ | ✓ | ✓ | 94.1 | 90.4 | 73.2 | 99.9 | 73.9 | 73.8 | 85.6 | 63.7 |

culations are performed using the Nomic embedding (Nussbaum et al., 2024) for PubMed abstracts and the text-embedding-3-large model from Azure OpenAI for all other purposes. The optimal thresholds for semantic clustering are determined to be $\theta_e = \theta_r = 0.14$. We generate community summaries using $Z_s = 20$ and $Z_c = 150$, and employ hyperparameters $\alpha = 0.1$, $\beta = 0.7$, $\lambda_1 = 0.2$, $\lambda_2 = 0.2$, and $\lambda_3 = 0.3$ for patient context augmentation. Claude 3.5 Sonnet, accessed via the Amazon Bedrock platform[1], is used as our expert LLM for generating reasoning chain training samples. Our fine-tuning framework is implemented using the TRL (von Werra et al., 2020), Transformers (Wolf et al., 2020), and FlashAttention-2 (Dao, 2024), with Mistral-7B-Instruct-v0.3 (Jiang et al., 2023) as our local LLM. We provide step-by-step implementation details in Appendix D. [2]

## 4.2 EXPERIMENTAL RESULTS

**Main Results.** Table 2 presents the main results and highlights several key observations: (1) KARE consistently outperforms all other methods across every dataset and task; (2) the naïve RAG model sometimes fails to enhance zero-shot performance, while our method effectively augments context, leading to improved zero-shot predictions; (3) our context augmentation method is comparable to the state-of-the-art EHR-CoAgent in few-shot scenarios; and (4) our approach identifies more unique patterns, particularly excelling in correctly predicting true positives for mortality prediction, which other supervised models struggle to capture. We place some case studies in Appendix F.

*Note*: In mortality prediction using MIMIC-III/IV, sensitivity is crucial because positive cases are significantly fewer than negative ones, increasing the risk of overfitting. Accurately predicting positive cases is essential. Our model's specificity is not always the highest, as efforts to enhance model's overall capability can sometimes lead to misclassification of negative cases as positive. This is a well-known trade-off between sensitivity and specificity (Zweig & Campbell, 1993; Powers, 2020). Conversely, for readmission prediction, where datasets are balanced, the model is expected to perform equally well on both positive and negative samples.

**Ablation Study of Fine-tuning Components.** We perform an ablation study to assess the individual contributions of each component in boosting the performance of our fine-tuned model, as illustrated in Table 3 and Figure 2. The results in Table 3 show that all components (similar patients, retrieved knowledge, and reasoning) contribute positively to performance in most cases. However, in highly imbalanced datasets like MIMIC-III Mortality, including similar patients can degrade performance. This is likely because the retrieved patients for positive cases (label = 1) tend to be less similar when positive samples are scarce. Additionally, the absence of these components makes the fine-tuned model more prone to label bias, as seen in MIMIC-III Mortality and MIMIC-IV Readmission. Figure 2 further illustrates

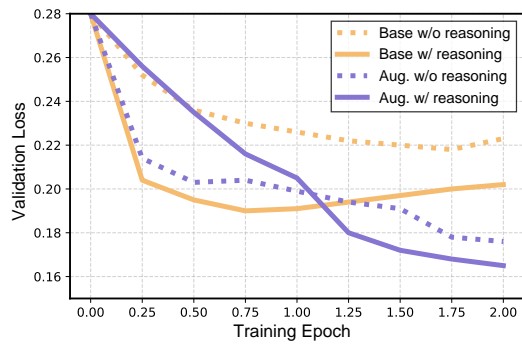

Figure 2: Validation loss of the label prediction during the fine-tuning with different settings. Loss is computed every 1/4 epoch. Task: mortality prediction on MIMIC-IV. "Base" and "Aug." denote base context and augmented context, respectively.

---

[1]The use of Amazon Bedrock is authorized by MIMIC: https://physionet.org/news/post/gpt-responsible-use
[2]Our code is available at: https://github.com/pat-jj/KARE

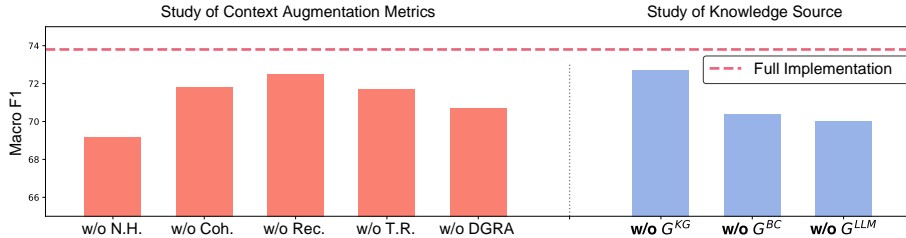

Figure 3: Ablation study of (*left*) the metrics we proposed for patient context augmentation, and (*right*) the KG used as the knowledge source. N.H., Coh., Rec., and T.R. denote node hits, coherence, recency, and theme relevance, respectively. Tested task: MIMIC-IV-Readmission.

that fine-tuning with base context leads to early overfitting compared to fine-tuning with augmented context. Moreover, adding reasoning as a multitask objective accelerates convergence for models using base context, whereas it slows convergence when applied to models with augmented context. This suggests that learning reasoning over more information-rich contexts is more challenging, but ultimately results in a lower final loss once mastered.

**Effect of Context Augmentation Components.** The LHS of Figure 3 compares the contribution of each metric proposed for community summary selection in patient context augmentation. The study shows that node hits is the most critical factor, followed by the DGRA algorithm, theme relevance, coherence, and recency, with each playing a distinct role in enhancing the final performance.

**Effect of the Knowledge Source.** The RHS of Figure 3 shows how removing individual knowledge sources affects the model's performance on the MIMIC-IV readmission task. While all KGs improve predictions, removing $G^{KG}$ causes the smallest performance drop, whereas removing $G^{LLM}$ leads to the largest decline. This highlights the importance of the LLM-extracted KG, likely due to its contextually relevant, clinically specific relationships. The UMLS-derived KG contributes less, likely because code mapping introduces sparsity by generalizing fine-grained concepts into more abstract categories (e.g., mapping "acute myocardial infarction" to "cardiovascular diseases"). This generalization limits the exploration of detailed relationships within the large KG. Future work will explore methods for retrieving knowledge with more fine-grained concepts from biomedical KGs.

**Benefit of Multitask Learning.** We compare our multitask learning approach, which treats reasoning chain generation and outcome prediction as separate tasks, with a "Two-In-One" method that only outputs the concatenated reasoning chain and ground-truth label. As shown in Table 4, multitask learning significantly outperforms the "Two-In-One" approach

Table 4: Comparison of two strategies for fine-tuning LLM with reasoning chain and label.

| | **MIMIC-IV-Mortality** | | **MIMIC-IV-Readmission** | |
|---|---|---|---|---|
| Strategy | Macro F1 | Sensitivity | Accuracy | Macro F1 |
| Multitask | 90.4 | 73.2 | 73.9 | 73.8 |
| "Two-In-One" | 86.5 | 68.0 | 67.2 | 65.4 |

for both mortality and readmission prediction on MIMIC-IV. This demonstrates that decoupling tasks allows better capture of each component's nuances, yielding more robust patient representations. This framework enables the LLM to specialize in generating quality reasoning chains while making accurate predictions, resulting in a more effective and interpretable model.

## 5 CONCLUSION

We propose KARE, a novel framework that combines community-based knowledge graph retrieval with large language model reasoning to enhance healthcare predictions. KARE constructs a comprehensive knowledge graph, employs hierarchical community detection, and dynamically augments patient context with fine-grained, contextually relevant information. By fine-tuning a specialized smaller LLM, KARE generates interpretable reasoning chains for accurate predictions. Experiments on MIMIC-III and MIMIC-IV datasets demonstrate KARE's superiority over state-of-the-art methods for mortality and readmission prediction tasks. Future work will focus on scaling KARE to more challenging healthcare tasks and exploring its applicability to other scientific domains, where integrating knowledge graphs and powerful language models can potentially drive groundbreaking scientific progress. We discuss ethics, broader impacts, and limitations in Appendix A. Human evaluation of KARE-generated reasoning chains is presented in Appendix H.

## 6 ACKNOWLEDGEMENTS

Research was supported in part by US DARPA INCAS Program No. HR0011-21-C0165 and BRIES Program No. HR0011-24-3-0325, National Science Foundation IIS-19-56151, the Molecule Maker Lab Institute: An AI Research Institutes program supported by NSF under Award No. 2019897, and the Institute for Geospatial Understanding through an Integrative Discovery Environment (I-GUIDE) by NSF under Award No. 2118329.

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

**Contents of Appendix**

## A    ETHICS, BROADER IMPACTS, AND LIMITATIONS

**Ethics:** Our work involves analysis of electronic health record (EHR) data, which contains sensitive personal medical information. To ensure the ethical handling of this data, we conducted all interactions between the language models and the EHR data through Amazon Bedrock[34], which provides rigorous compliance standards and privacy protection measures. This allowed us to fully leverage the capabilities of the LLMs while maintaining strict confidentiality of the patient data.

**Broader Impacts:** Our framework, KARE, demonstrates the potential for integrating knowledge graphs and language model reasoning to enhance clinical decision support systems. By providing more accurate and interpretable predictions for critical outcomes like mortality and readmission, KARE could assist healthcare providers in identifying high-risk patients who may require additional interventions or closer monitoring. This could ultimately lead to improved patient outcomes and more efficient allocation of healthcare resources. However, it is important to recognize that our models are intended to augment, rather than replace, the judgment of healthcare professionals. The predictions should be considered as additional data points to inform clinical decision making, not as definitive diagnoses or treatment recommendations.

**Limitations:** While KARE achieves promising results, there are several limitations to consider. First, our evaluation is based on the MIMIC-III and MIMIC-IV datasets, which represent a specific patient population from a single hospital system in the United States. The generalizability of our findings to other patient populations or healthcare settings may be limited. Second, our knowledge graphs are constructed from a subset of biomedical databases, literature, and language model outputs, and may not capture the full breadth of medical knowledge. Expanding the knowledge sources and improving the knowledge extraction and integration processes could further enhance the performance of our models. Third, our framework relies on the outputs of large language models, which are known to have biases and can generate hallucinations. While we have taken steps to mitigate these issues, such as collaborating with medical experts to validate the extracted knowledge, there remains a risk of the models producing incorrect or biased predictions in some cases. Fourth, our work is based on English biomedical literature and EHR data, and we did not evaluate the performance of KARE on data in other languages. The applicability and effectiveness of our approach for non-English clinical settings requires further investigation. Ongoing research is needed to develop more robust and reliable language models for clinical applications across diverse languages and populations.

In conclusion, KARE represents an important step towards leveraging knowledge graphs and language model reasoning for improved clinical predictions. However, further research is needed to address the limitations and ensure the safe and responsible deployment of such models in real-world healthcare settings. We encourage future work to focus on enhancing the generalizability, interpretability, and robustness of these approaches, as well as engaging with healthcare stakeholders to develop guidelines for the ethical and effective use of AI in clinical decision support.

## B    DETAILS OF KNOWLEDGE GRAPH CONSTRUCTION

### B.1    KG EXTRACTION FROM LARGE BIOMEDICAL KG

To construct concept-specific knowledge graphs from the Unified Medical Language System (UMLS), we follow a multi-step process:

1. **Extracting Co-existing Concepts from EHR Data**: We iterate through the patient EHR dataset and collect the top $X$ ($X = 20$ in our implementation) most frequently co-existing concepts for each unique medical concept based on their co-occurrence in patient records.

2. **Mapping Concepts to UMLS CUIs**: We map the medical concepts from the EHR data to their corresponding Concept Unique Identifiers (CUIs) in UMLS. This involves mapping condition and procedure concepts to CCS codes, then to ICD-9 codes, and finally to UMLS CUIs. Drug concepts are directly mapped to ATC codes and then to UMLS CUIs.

---

[3] https://docs.aws.amazon.com/bedrock/latest/userguide/what-is-bedrock.html

[4] https://physionet.org/news/post/gpt-responsible-use

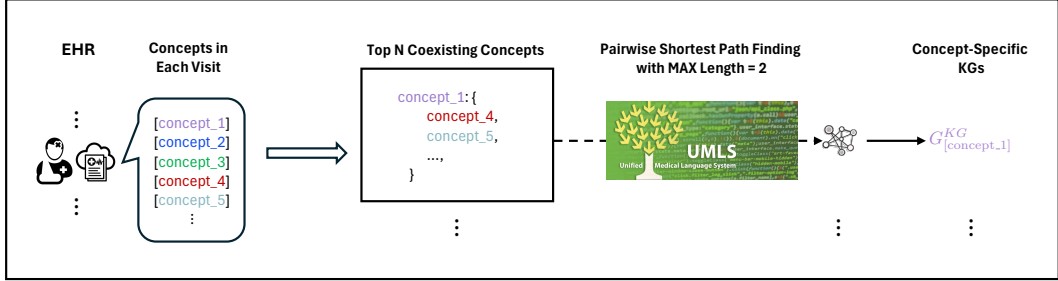

Figure 4: Our pipeline to construct concept-specific KG $G^{KG}$ with bio KG (UMLS) and EHR.

3. **Extracting Subgraphs from UMLS**: For each medical concept $c_i$ and its top $X$ co-existing concepts $R_{c_i}$, we extract a concept-specific subgraph $G_{c_i}^{\text{KG}}$ from UMLS using a bidirectional shortest path finding algorithm. The algorithm parameters are set as follows:

   - `max_length=7`: The maximum length of the shortest paths considered between concepts.
   - `max_paths=40`: The maximum number of shortest paths to retrieve for each pair of concepts.
   - `max_nodes=12000`: The maximum number of nodes to explore during the bidirectional search.

   The bidirectional shortest path finding algorithm is implemented as follows:

---

**Algorithm 2** Bidirectional Shortest Path Finding

---

**Input:** Graph $G$, start node $s$, end node $t$, max length $l$, max paths $p$, max nodes $n$
**Output:** Set of shortest paths $P$
Initialize forward queue $Q_f \leftarrow \{[s]\}$ and backward queue $Q_b \leftarrow \{[t]\}$
Initialize forward visited dict $V_f \leftarrow \{s : [s]\}$ and backward visited dict $V_b \leftarrow \{t : [t]\}$
Initialize paths $P \leftarrow \emptyset$ and nodes explored $N \leftarrow 0$
**while** $Q_f$ and $Q_b$ and $|P| < p$ and $N < n$ **do**
  Expand forward path $\pi_f \leftarrow Q_f.\text{popleft}()$ and last node $u \leftarrow \pi_f[-1]$
  **if** $u \in V_b$ **then**
    | $P \leftarrow P \cup \{\pi_f + V_b[u][:: -1][1 :]\}$
  **end**
  Expand backward path $\pi_b \leftarrow Q_b.\text{popleft}()$ and last node $u \leftarrow \pi_b[-1]$
  **if** $u \in V_f$ **then**
    | $P \leftarrow P \cup \{V_f[u] + \pi_b[:: -1][1 :]\}$
  **end**
  **if** $len(\pi_f) < 2l$ or $len(\pi_b) < 2l$ **then**
    | Expand neighbors and update $Q_f, Q_b, V_f, V_b$
  **end**
**end**
**return** $P$

---

The extracted shortest paths for each pair $(c_i, c_j)$ are combined to form the concept-specific subgraph $G_{c_i}^{\text{KG}}$. The nodes $V_{c_i}^{\text{KG}}$ and edges $E_{c_i}^{\text{KG}}$ of the subgraph are the union of all nodes and edges in the extracted paths.

By following this process, we construct a set of concept-specific knowledge graphs $\{G_{c_i}^{\text{KG}}\}$ that capture the relevant relationships and contextual information for each medical concept $c_i$ based on the UMLS knowledge graph and real-world co-occurrence patterns in patient EHR data.

The number of the resulting KG triples from UMLS is 29,434.

## B.2  KG EXTRACTION FROM LARGE BIOMEDICAL CORPUS

To construct concept-specific knowledge graphs, we process the EHR dataset and the PubMed Abstracts corpus using the following pipeline (Figure 5):

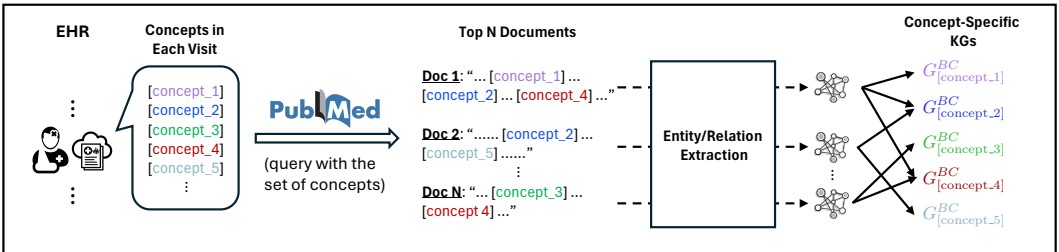

Figure 5: Our pipeline to construct concept-specific KG $G^{BC}$ from biomedical corpus with EHR.

1. **Concept Set Extraction from EHR Data**: We iterate through the EHR dataset and, for each patient visit, we collect all the involved medical concepts (conditions, procedures, and drugs) into a concept set. This results in a list of concept sets, where each set represents the concepts associated with a specific patient visit.

2. **Filtering Similar Concept Sets**: To reduce redundancy and computational overhead, we filter out highly similar concept sets based on their concept multi-hot vector representation. We use the `CountVectorizer` from scikit-learn (Pedregosa et al., 2018) to create a vocabulary of unique concepts and transform each concept set into a multi-hot vector. We then compare the symmetric difference between pairs of concept sets and filter out sets that differ by fewer than a specified similarity threshold (5 in our case). This step helps to eliminate nearly duplicate concept sets while retaining a diverse range of concepts for knowledge graph construction.

   This process results in 26,134 concept sets in our experiment.

3. **Dense Retrieval of Relevant PubMed Abstracts**: For each filtered concept set, we retrieve the top $n$ ( $n = 3$ in our case) most relevant documents from PubMed Abstracts using dense retrieval. Considering the expensive time consumption to retrieve abstracts from the full corpus (with 30 million abstracts), we randomly (seed=42) select 1/10 abstracts as the subset corpus to proceed. We employ the `nomic-ai/nomic-embed-text-v1.5` model to embed both the concepts and the PubMed abstracts into a dense vector space (dim=768). We then compute the cosine similarity between the concept set embedding and the abstract embeddings to identify the most relevant documents. The dense retrieval process is optimized for efficiency by processing the embeddings in chunks and utilizing GPU acceleration when available.

4. **Triple Extraction from PubMed Abstracts**: For each retrieved PubMed abstract, we perform entity extraction and relation extraction to extract knowledge graph triples. We use a large language model (Claude 3.5 Sonnet in our case), to identify relationships between the concepts mentioned in the abstract. The LLM is provided with a prompt that includes the abstract text, the list of relevant concepts, and an example of the desired triple format. The prompt instructs the LLM to extract at most 10 informative and logically sound triples for each abstract, focusing on relationships closely related to the provided concepts. The extracted triples follow the format `[ENTITY1, RELATIONSHIP, ENTITY2]`, where the entities are replaced with the exact concept terms when applicable. The prompt we used for the triple extraction is:

> Given a medical text and a list of important concepts, extract relevant relationships between the concepts from the text (if present). For each triple, if an entity matches one of the given concepts, replace the entity with the exact concept term.
> Focus on generating high-quality triples closely related to the provided concepts. Aim to extract at most 10 triples for each text. Each triple should follow this format: [ENTITY1, RELATIONSHIP, ENTITY2]. Ensure the triples are informative and logically sound.
>
> Example:
> Text:
> Asthma is a chronic respiratory condition characterized by inflammation and narrowing of the airways, leading to breathing difficulties. Common symptoms include wheezing, coughing, shortness of breath, and chest tightness. Triggers can vary but often include allergens, air pollution, exercise, and respiratory infections. Management typically involves a combination of long-term control medications, such as inhaled corticosteroids, and

> quick-relief medications like short-acting beta-agonists. Recent research has focused on personalized treatment approaches, including biologics for severe asthma and the role of the microbiome in asthma development and progression. Proper inhaler technique and adherence to medication regimens are crucial for effective management. Asthma action plans, developed in partnership with healthcare providers, help patients manage symptoms and exacerbations.
>
> Concepts:
> [asthma, inflammation, airways, wheezing, coughing, inhaled corticosteroids, short-acting beta-agonists, allergens, respiratory infections]
>
> Extracted triples:
> [[asthma, is a, chronic respiratory condition], [asthma, characterized by, inflammation of airways], [inflammation, causes, narrowing of airways], [narrowing of airways, leads to, breathing difficulties], [wheezing, is a symptom of, asthma], [coughing, is a symptom of, asthma], [allergens, can trigger, asthma], [respiratory infections, can trigger, asthma], [inhaled corticosteroids, used for, long-term control of asthma], [short-acting beta-agonists, provide, quick relief in asthma]]
>
> Text:
> {text}
>
> Concepts:
> {concepts}
>
> Extracted triples:

5. **Knowledge Graph Construction**: The extracted triples from each abstract are added to the knowledge graph of the medical concepts mentioned in the document. This process builds a concept-specific knowledge graph $G_{c_i}^{\text{BC}}$ for each concept $c_i$, incorporating relevant information from the PubMed corpus. The resulting knowledge graphs capture the relationships and contextual information associated with each medical concept.

By following this pipeline, we construct a comprehensive set of concept-specific knowledge graphs that integrate information from both EHR data and the PubMed corpus. These knowledge graphs serve as a valuable resource for EHR-based downstream tasks, such as patient representation learning and predictive modeling.

The number of resulting KG triples from the PubMed Abstracts is 259,938.

### B.3 KG Extraction from Large Language Models

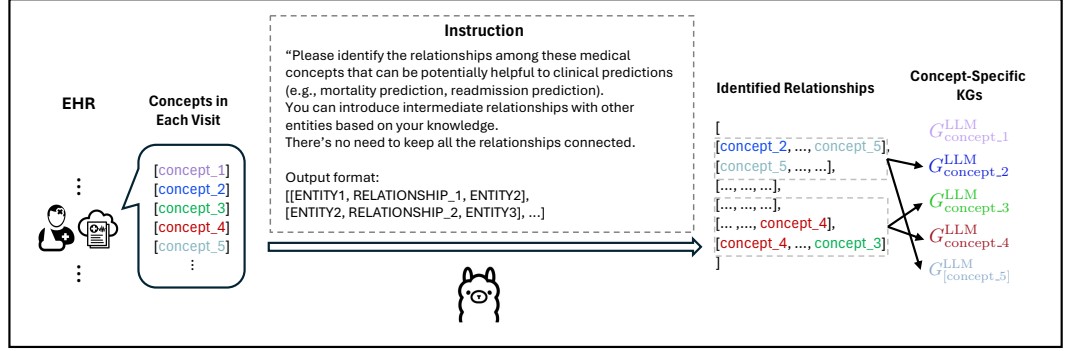

Figure 6: Our pipeline to construct concept-specific KG with LLM and EHR.

To extract concept-specific knowledge graphs from large language models (LLMs), we follow a process similar to the initial steps of the "KG extraction from corpus" pipeline (Appendix B.2):

1. **Concept Set Extraction from EHR Data**: We iterate through the EHR dataset and, for each patient visit, we collect all the involved medical concepts (conditions, procedures, and drugs) into a concept set. This results in a list of concept sets, where each set represents the concepts associated with a specific patient visit.

2. **Filtering Similar Concept Sets**: To reduce redundancy and computational overhead, we filter out highly similar concept sets based on their concept multi-hot vector representation, as described in the previous subsection. This step helps to eliminate nearly duplicate concept sets while retaining a diverse range of concepts for knowledge graph construction.

3. **Prompting LLMs for Relationship Extraction**: For each filtered concept set, we prompt a large language model (Claude 3.5 Sonnet in our case) to identify relationships among the medical concepts that can be potentially helpful for clinical predictions, such as mortality prediction and readmission prediction. The LLM is encouraged to introduce intermediate relationships with other entities based on its knowledge, and there is no requirement to keep all the relationships connected. The prompt instructs the LLM to use the original names of the provided concepts in the output, which should follow the format `[[ENTITY1, RELATIONSHIP_1, ENTITY2], [ENTITY2, RELATIONSHIP_2, ENTITY3], ...]`. The prompt we used for the relationship extraction is:

   > Please identify the relationships among these medical concepts that can be potentially helpful to clinical predictions (e.g., mortality prediction, readmission prediction) as many as possible.
   >
   > You can introduce intermediate relationships with other entities based on your knowledge.
   >
   > Consider how these concepts would interact with others to be useful for clinical predictions. There's no need to keep all the relationships connected.
   >
   > For the concepts provided in the list, you MUST use the their original name without any changes. Please output only the list of triples without any other information.
   >
   > Output format:
   > [[ENTITY1, RELATIONSHIP_1, ENTITY2],
   > [ENTITY2, RELATIONSHIP_2, ENTITY3], ...]
   >
   > Medical Concepts:
   > concepts
   >
   > Output:

4. **Knowledge Graph Construction**: The extracted triples for each concept set are used to construct a concept-specific knowledge graph $G_{c_i}^{\text{LLM}}$ for each concept $c_i$. For each concept $c_i$ in the concept set, we store the connected 3-hop subgraph sourced from $c_i$ to its corresponding concept-specific knowledge graph $G_{c_i}^{\text{LLM}}$. This step ensures that only the triples directly or indirectly connected to the concept $c_i$ are included in its concept-specific knowledge graph. This process is iteratively performed for all the concepts in the concept set. The resulting knowledge graphs capture the relationships and contextual information associated with each medical concept based on the knowledge embedded in the large language model.

By leveraging the knowledge embedded in large language models, this process allows us to construct concept-specific knowledge graphs that incorporate a broad range of information beyond what is explicitly stated in the EHR data or biomedical literature. These knowledge graphs can provide valuable insights and support various downstream tasks in the clinical domain.

The number of resulting KG triples from the LLM is 315,492.

## B.4 CASE STUDIES OF KG CONSTRUCTION

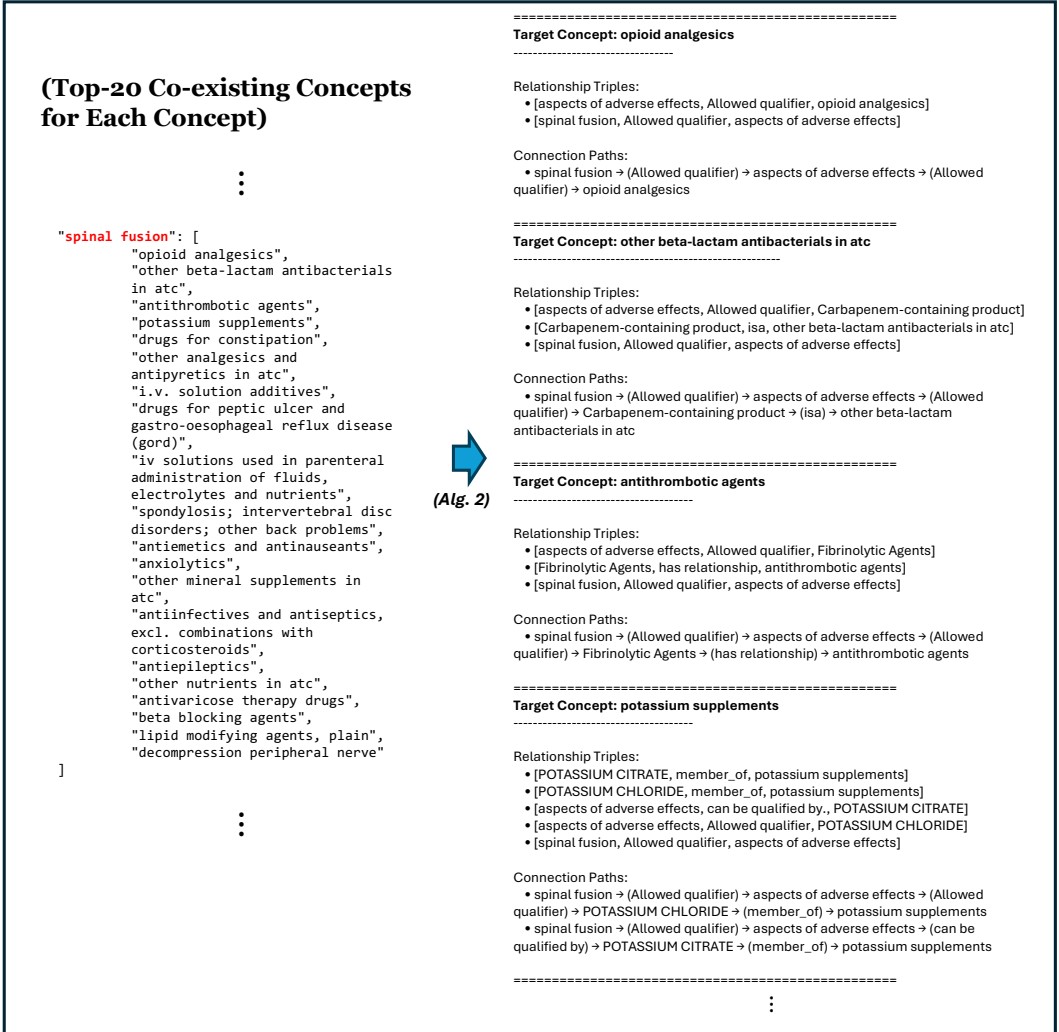

Figure 7: An example of concept KG extraction from biomedical KG (UMLS).

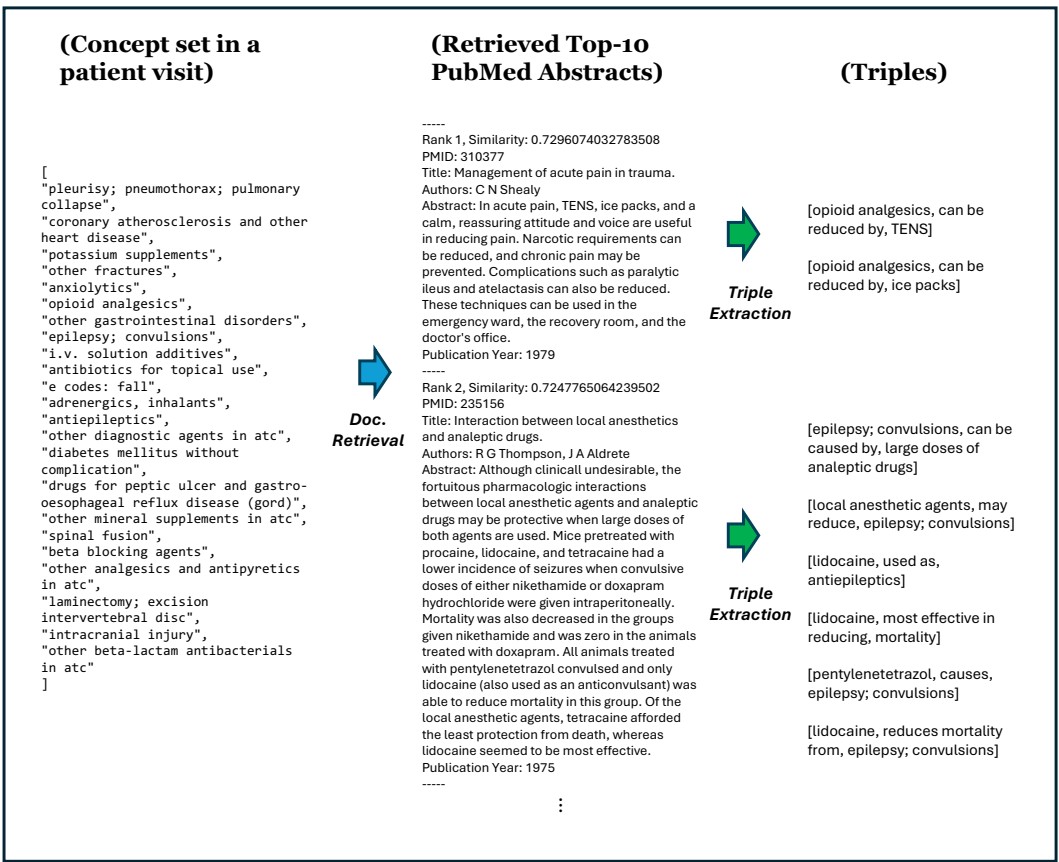

Figure 8: An example of concept KG extraction from Corpus (PubMed Abstract).

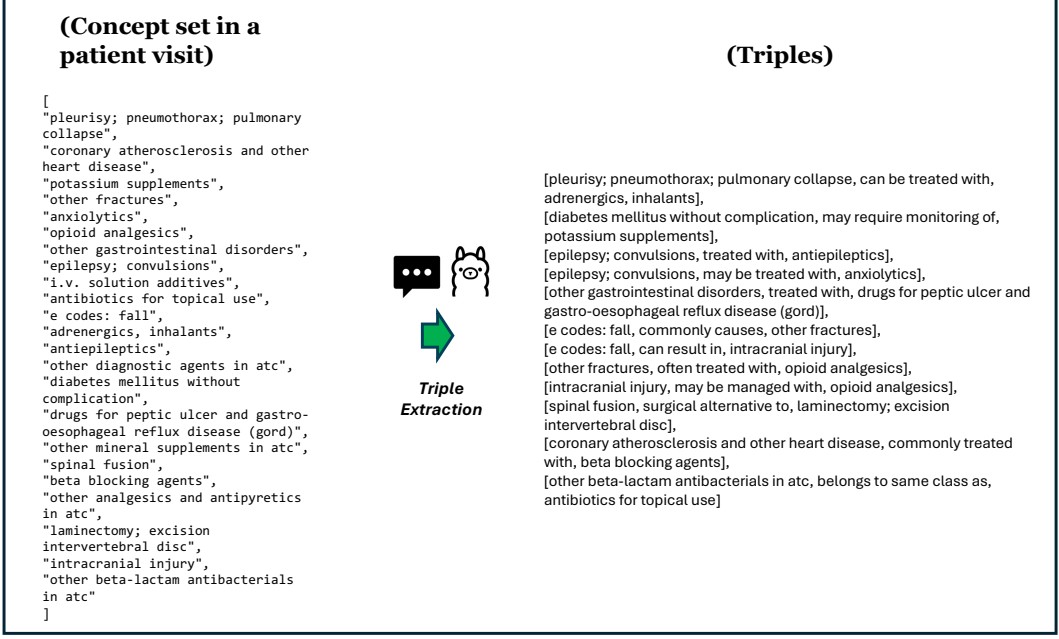

Figure 9: An example of oncept KG extraction from LLM (Claude-3.5).

## C  BASELINE IMPLEMENTATIONS

**ML-based Models.** To ensure a fair performance comparison, we implemented all machine learning (ML)-based Electronic Health Record (EHR) models using PyHealth (Yang et al., 2023a). Since GRAM (Choi et al., 2017) and KerPrint (Yang et al., 2023b) were not yet integrated into PyHealth, we separately implemented these models within the `pyhealth.models` module. Our implementations were based on the original codebases for GRAM[5] and KerPrint[6]. For consistency across all ML-based models, we set the embedding size to 256. We trained the models using a learning rate of 1e-4 and employed an early stopping mechanism based on validation loss to prevent overfitting.

**LM+ML Models.** We implement GraphCare (Jiang et al., 2024a) using their official codebase[7] with their default setting for each component. We use `text-embedding-3-large` (an improved version of `text-embedding-ada-002` used in the original implementation) as the embedding model for the embedding initialization, and use their proposed BAT as the base GNN model. We implement RAM-EHR (Xu et al., 2024) using their codebase[8] with the settings mentioned in the implementation details in their paper. We implement EMERGE (Zhu et al., 2024a) fully following the implementation details provided in their paper (with the LLMs Clinical-LongFormer, BGE-M3, Qwen 1.5-7B Chat, and DeepSeek-V2 Chat used for different purposes in the pipeline).

**LLM-based Methods.** For zero-shot and few-shot prompting-based EHR prediction with the LLM, we utilize the template presented in Table 7 which includes *task description*, *task-specific instruction*, *patient base context*, *supplementary information* (based on retrieval), and *Patient References* (similar patients). Unlike the structured format used for reasoning-chain generation, the reasoning here is presented in a free-style manner, which, as our study indicates, results in better performance.

We implement the EHR-CoAgent (Cui et al., 2024) approach as described in their paper[9], which combines the strengths of predictive agent reasoning and critical agent instruction to create a accurate few-shot prediction system for our tasks. The implementation consists of two main components: a predictor agent and a critic agent.

The predictor agent is responsible for generating predictions and providing explanatory reasoning based on the input EHR data. Given a patient's medical history, the predictor agent analyzes the relevant information and generates the most likely prediction along with a step-by-step explanation of its reasoning process. The prompt used for the predictor agent is as follows:

---

Given the following task description, patient EHR context, task instructions, and similar patients, please make a prediction with reasoning.

\# Task \#
[Task Definition] + [Regulator]

\# Patient EHR Context \#
[Patient's Context (Base)]

\# Task Instructions (Guidelines) \#
[Refined Guidelines if iteration > 1 else Initial Guidelines]

\# Similar Patients \#
[Top-K similar patients' contexts]

Give the prediction and reasoning in the following format:
\# Reasoning \#

---

[5]https://github.com/mp2893/gram
[6]https://github.com/xyxpku/KerPrint
[7]https://github.com/pat-jj/GraphCare
[8]https://github.com/ritaranx/RAM-EHR
[9]*Note*: While EHR-CoAgent was not originally designed for mortality or readmission prediction tasks, we have made minor custom modifications to adapt it to our specific use cases.

---

[Your reasoning here]

# Prediction #
[Your prediction here (1/0)]

Output:

---

where *Task Definition* is a brief definition of the task (e.g., Mortality Prediction Task: Objective: Predict the mortality outcome for a patient's subsequent hospital visit based solely on conditions, procedures, and medications. Labels: 1 = mortality, 0 = survival). We introduce a *Regulator* to import prior knowledge of the dataset to avoid the LLM to over-focus on improving true positive or true negative. For example, we set "***Must to Notice:** Only the patients with extremely very high risk of mortality should be predicted as 1.* " as the regulator for the mortality prediction task. The existence of the regulator significantly affect the final result for imbalanced datasets, as shown in Table 5. This is because that the instruction-updating approach used by EHR-CoAgent tends to excessively penalize false positives to produce instructions that boost true positives, especially in cases of imbalanced data like mortality prediction in the two datasets. However, this can compromise true negatives and accuracy in such scenarios.

Table 5: Significant performance difference between EHR-CoAgent w/ regulator and w/o regulator.

|  | MIMIC-III Mortality | | | | MIMIC-IV Mortality | | | |
|---|---|---|---|---|---|---|---|---|
|  | Accuracy | Macro F1 | Sensitivity | Specificity | Accuracy | Macro F1 | Sensitivity | Specificity |
| EHR-CoAgent w/ Regulator | 87.4 | 51.7 | 13.0 | 91.7 | 81.0 | 55.5 | 13.8 | 97.0 |
| EHR CoAgent w/o Regulator | 53.6 | 39.7 | 51.9 | 53.6 | 72.4 | 61.8 | 51.3 | 77.4 |

*Refined Guidelines* are the instructions refined by the critic agent, while *Initial Guidelines* are the seed instructions, which is as same as the task descriptions we used for other methods.

The critic agent, on the other hand, plays a different role in the EHR-CoAgent framework. It observes the predictor agent's outputs alongside the ground truth labels and identifies error patterns and discrepancies in the predictor's reasoning process. Based on this analysis, the critic agent refine the instructions to improve the reasoning process of the predictor. The prompt used for the critic agent is as follows:

---

You are an assistant who is good at self-reflection, gaining experience, and summarizing criteria. By reflecting on failure predictions that are given below, your task is to reflect on these incorrect predictions, compare them against the ground truth, and formulate criteria and guidelines to enhance the accuracy of future predictions.
The original instructions are provided under "# Task Instructions (Guidelines) #". Your task is to refine the instructions based on the discrepancies between the predictions and the ground truth.

# Input Data #
[input data batch with prompts, predictions, and ground-truth labels]

# Instructions #
1. Please always remember that the predictions above are all incorrect. You should always use the ground truth as the final basis to discover many unreasonable aspects in the predictions and then summarize them into experience and criteria.
2. Identify why the wrong predictions deviated from the ground truth by examining discrepancies in the medical history analysis.
3. Determine key and potential influencing factors, reasoning methods, and relevant feature combinations that could better align predictions with the ground truth.
4. The instructions should be listed in distinct rows, each representing a criteria or guideline.
5. The instructions should be generalizable to multiple samples, rather than specific to individual samples.
6. Conduct detailed analysis and write criteria based on the input samples, rather than writing some criteria without foundation.

---

> 7. Please note that the criteria you wrote should not include the word "ground truth".
>
> Your output should be the new set of guidelines under "# Task Instructions (Guidelines) #" that can be used to improve the predictor's reasoning process.
>
> Output:

which is mostly the same as the one in their paper (Cui et al., 2024).

Our implementation of EHR-CoAgent follows an iterative refinement process, where the predictor agent generates predictions, the critic agent analyzes incorrect predictions and refines the instruction, which is consolidated and integrated into the predictor's prompts for the next round. The incorrect predictions are divided into batches, and the critic agent refines the instructions for each batch. The instructions from all batches is then consolidated using the LLM to identify the most important and recurring insights across the entire refined instruction list. This consolidated new guidelines are integrated into the predictor's prompts for the next round, allowing the system to effectively improve its prediction performance. We iterate the process 5 times.

The consolidated instructions are generated using the following prompt:

> Given the following set of guidelines, please consolidate the insights into a concise and coherent set of guidelines for refining the predictor's reasoning process.
>
> # Set of Guidelines #
> [A Batch of Guidelines]
>
> # Instructions #
> 1. Analyze the provided guidelines and identify common themes, patterns, and key insights.
> 2. Synthesize the insights into a consolidated set of guidelines that capture the most important and recurring aspects.
> 3. Ensure that the consolidated guidelines are clear, concise, and actionable to refine the predictor's reasoning process.
> 4. Create a numbered list of the consolidated guidelines in the same format as the original guidelines.
>
> Output:

The consolidated instructions are then recursively consolidated until the final list size is smaller than 10. This is done to ensure that the consolidated instructions can be effectively integrated into the predictor's prompts for the next round, considering the limited context window size of the LLM.

By incorporating the consolidated instructions into the predictor's prompts, the EHR-CoAgent approach enables an iterative refinement process to improve the accuracy predictions.

# D    IMPLEMENTATION DETAILS OF KARE

## D.1    STEP 1: MEDICAL CONCEPT KNOWLEDGE GRAPH CONSTRUCTION

**Step 1.1: Medical Concept-Specific Knowledge Graph Extraction**

We use UMLS, PubMed Abstracts, and Claude 3.5 Sonnet as the sources for knowledge graph extraction from Biomedical KG, Biomedical Corpus, and Large Language Model, respectively. The extraction details are showcased in Appendix B.1, B.2, and B.3, respectively.

For UMLS, we utilize the "Full Release" version under "2024AA Full UMLS Release Files"[10]. For dense retrieval from PubMed abstracts, we utilize the local embedding model Nomic (dimension = 768) (Nussbaum et al., 2024). We use Amazon Bedrock[11] to access the Claude model.

The resulting KG triples from UMLS, PubMed Abstracts, and Sonnet are 29,434, 259,938, and 315,492, respectively.

**Step 1.2: Semantic Clustering**

For semantic clustering of entities and relations in the KG we build above: we (1) first use the text-embedding-3-large model (dimension = 1024) from Azure OpenAI to retrieve the text embeddings of entities and relations; and (2) use Scikit-learn (Pedregosa et al., 2018) to perform agglomerative clustering based on those embeddings. The optimal cosine distance thresholds $\theta_e$ and $\theta_r$ are both found to be 0.14, resulting in 513,867 triples in total after clustering.

**Step 1.3: Hierarchical KG Community Detection and Indexing**

We employ Graspy (Chung et al., 2019) to implement the hierarchical Leiden algorithm, setting the maximum size for each top-level community (max_cluster_size) to 5.

To enhance community diversity, the algorithm is run 25 times with different randomness at each iteration, resulting in unique 59,832 communities (with different combinations of triples) where there are 40,934 communities with the size smaller than 20 ($Z_s$), and 57,247 communities with the size smller than 150 ($Z_c$).

Using Claude 3.5 Sonnet as the LLM, we generate 147,264 community summaries (including both general and theme-specific summaries) with the prompts shown in Figure 12 and 13.

## D.2    STEP 2: PATIENT CONTEXT CONSTRUCTION AND AUGMENTATION

We use the template as shown in Figure 11 to construct patient's base context based on their EHR.

To retrieve the relevant medical knowledge for context augmentation, we set $\alpha = 0.1$, $\beta = 0.7$, $\lambda_1 = 0.2$, $\lambda_2 = 0.2$, and $\lambda_3 = 0.3$ as the hyperparameters, each tuned in the range of $[0, 1]$.

## D.3    STEP 3: REASONING-ENHANCED PRECISE HEALTHCARE PREDICTION

**Step 3.1: Training Sample Generation**

To generate reasoning chain training samples, we leverage Claude 3.5 Sonnet as our expert LLM. Ensuring EHR data protection and ethical use is paramount; therefore, all LLM interactions are conducted via the Amazon Bedrock platform[12], a cloud infrastructure that allows us to fully harness LLM capabilities while maintaining strict privacy measures. The maximum output length for Sonnet is set as 4,096 tokens. We use the prompt in Figure 16 for the reasoning chain generation here.

**Step 3.2 Multitask-Based Fine-Tuning and Prediction**

Our fine-tuning framework is implemented using the TRL (von Werra et al., 2020), Transformers (Wolf et al., 2020), and FlashAttention-2 (Dao, 2024) Python libraries. We use Mistral-7B-Instruct-

---

[10]https://www.nlm.nih.gov/research/umls/licensedcontent/umlsknowledgesources.html

[11]https://docs.aws.amazon.com/bedrock/

[12]The use of Amazon Bedrock is authorized by MIMIC: https://physionet.org/news/post/gpt-responsible-use

v0.3 (Jiang et al., 2023) as our local LLM, full-parameter fine-tuned using DeepSpeed (Rasley et al., 2020) with the following configurations:

| Parameter | Value |
|---|---|
| model_name_or_path | mistralai/Mistral-7B-Instruct-v0.3 |
| torch_dtype | bfloat16 |
| use_flash_attention_2 | true |
| preprocessing_num_workers | 12 |
| bf16 | true |
| gradient_accumulation_steps | 4 |
| gradient_checkpointing | true |
| learning_rate | 5.0e-06 |
| max_seq_length | 6000 |
| num_train_epochs | 3 |
| per_device_train_batch_size | 1 |
| lr_scheduler_type | cosine |
| warmup_ratio | 0.1 |

Table 6: LLM Fine-tuning Configuration Parameters

*Hardware Information*: The experiments were conducted on a system with an AMD EPYC 7513 32-Core Processor and 1.0 TB of RAM. The setup includes eight NVIDIA A100 80GB PCIe GPUs, each with 81920 MiB of memory, providing a total of 640 GB GPU memory. The system's root partition has 32 GB of storage.

Training of each model runs on eight NVIDIA A100 GPUs and typically completes within five hours. After the training process, we select the best performing model checkpoint based on validation loss to perform the prediction.

# E    EVALUATION METRICS

In this work, we employ four key evaluation metrics to assess model performance:

## E.1    PRIMARY METRICS

1. **Accuracy**: Measures the overall proportion of correct predictions:

$$\text{Accuracy} = \frac{\text{TP} + \text{TN}}{\text{TP} + \text{TN} + \text{FP} + \text{FN}} \tag{7}$$

2. **Macro-F1**: Provides a balanced measure that is particularly important for imbalanced datasets:

$$\text{Macro-F1} = \frac{2 \times \text{Precision} \times \text{Recall}}{\text{Precision} + \text{Recall}} \tag{8}$$

where Precision = TP/(TP + FP) and Recall = TP/(TP + FN)

3. **Sensitivity** (True Positive Rate): Quantifies the model's ability to correctly identify high-risk patients:

$$\text{Sensitivity} = \frac{\text{TP}}{\text{TP} + \text{FN}} \tag{9}$$

4. **Specificity** (True Negative Rate): Assesses the model's accuracy in identifying low-risk patients:

$$\text{Specificity} = \frac{\text{TN}}{\text{TN} + \text{FP}} \tag{10}$$

where TP = True Positives, TN = True Negatives, FP = False Positives, and FN = False Negatives.

## E.2    CHOICE OF METRICS

While metrics like AUROC (Area Under the Receiver Operating Characteristic) and AUPRC (Area Under the Precision-Recall Curve) are commonly used for imbalanced classification tasks, they are not suitable for our LLM-based approach for several reasons:

1. **LLM Probability Limitations**: LLMs compute next-token probabilities over their entire vocabulary rather than binary class probabilities. These probabilities:

- Are distributed across the full vocabulary rather than just binary classes
- Depend on how different LLMs encode the same label ("0"/"1") using different tokens
- Are not directly comparable to class probabilities output by traditional ML models

2. **Clinical Relevance**: Our chosen metrics provide direct clinical interpretability:

- Sensitivity is crucial for identifying high-risk patients who require immediate attention
- Specificity helps avoid unnecessary interventions for low-risk patients
- Together, they effectively evaluate performance on imbalanced datasets using only final predictions

Our metric selection aligns with recent LLM-based EHR prediction works, such as EHR-CoAgent (Cui et al., 2024).

The effectiveness of our metric choice is demonstrated in the MIMIC-III/IV mortality prediction tasks, where despite high class imbalance (positive rates of 5.42%/19.16%), KARE achieves significantly higher sensitivity (24.7%/73.2%) compared to baselines while maintaining high specificity (98.3%/99.8%).

# F CASE STUDY

## F.1 REASONING-ENHANCED PREDICTION BY OUR FINE-TUNED MODEL

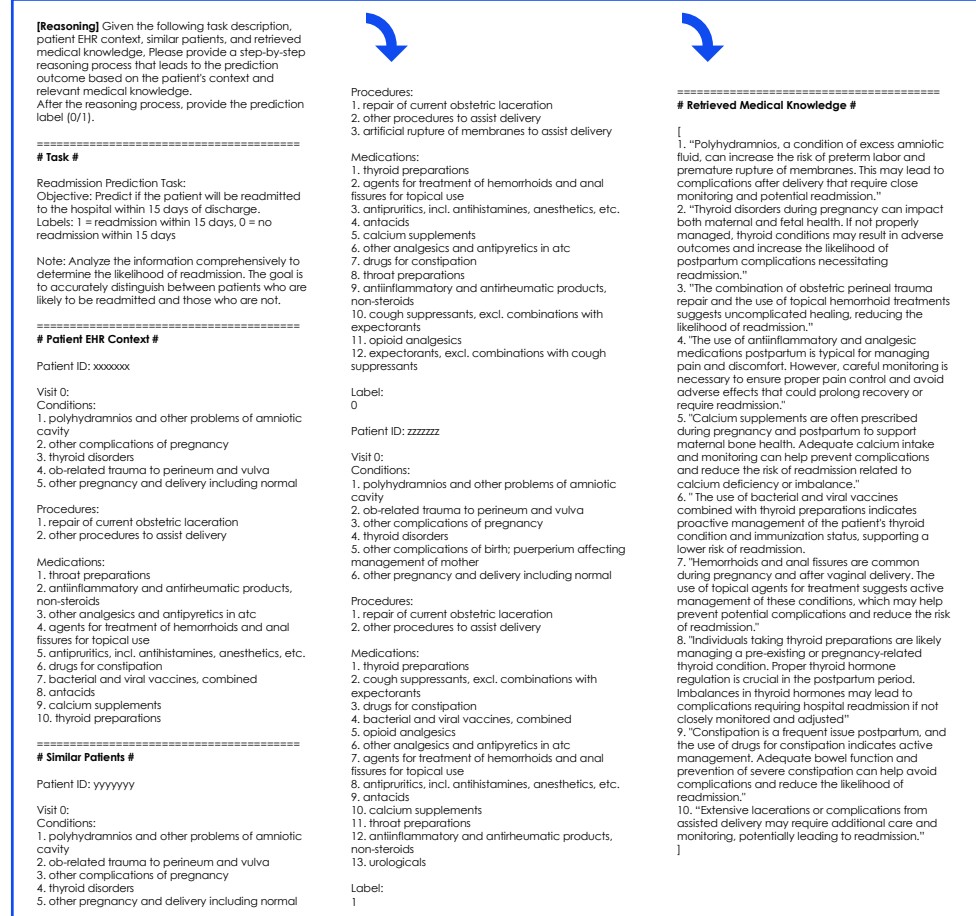

Figure 10: Case Study of the Fine-tuned KARE Model.

## F.2 ZERO-SHOT & FEW-SHOT COMPARISON

Table 7: Case Study of zero-shot and few-shot EHR prediction with LLM (Sonnet-3.5). Ground Truth: The patient will die in the next visit (Prediction = **1**). For the ethic concern, the patients involved are all from synthetic MIMIC-III by HALO (Theodorou et al., 2023).

| Case | Description |
|---|---|
| Input Prompt with EHR | Given the following task description and patient context, please make a prediction with reasoning based on the patient's context. |
| | **Task**: Mortality Prediction Task |
| | **Objective**: Predict the mortality outcome for a patient's subsequent hospital visit based solely on conditions, procedures, and medications. |
| | **Labels**: 1 = mortality, 0 = survival |
| | **Key Considerations**: |
| | Conditions: |
| | • Severity of diagnosed conditions (e.g., advanced cancer, severe heart failure, sepsis) |
| | • Presence of multiple comorbidities |
| | • Acute vs. chronic nature of conditions |
| | Procedures: |
| | • Invasiveness and complexity of recent procedures |
| | • Emergency vs. elective procedures |
| | • Frequency of life-sustaining procedures (e.g., dialysis, mechanical ventilation) |
| | Medications: |
| | • Use of high-risk medications (e.g., chemotherapy drugs, immunosuppressants) |
| | • Multiple medication use indicating complex health issues |
| | • Presence of medications typically used in end-of-life care |
| | Note: Focus on combinations of conditions, procedures, and medications that indicate critical illness or a high risk of mortality. Consider how these factors interact and potentially exacerbate each other. Only the patients with extremely very high risk of mortality should be predicted as 1. |
| | **Patient Context**: |
| | Patient ID: 29488 |
| | Visit 0: |
| | Conditions: |
| | - Deficiency and other anemia |
| | - Essential hypertension |
| | - Complication of device; implant or graft |
| | - Congestive heart failure; nonhypertensive - Cancer of prostate |
| | - Anxiety disorders |
| | - Thyroid disorders |
| | - Disorders of lipid metabolism |
| | - Conduction disorders |
| | - Mycoses |
| | - Other diseases of kidney and ureters |
| | - Cancer of esophagus |
| | Procedures: |
| | - Diagnostic cardiac catheterization; coronary arteriography |
| | - Other or procedures on vessels other than head and neck |
| | - Colorectal resection |
| | Drugs: |
| | - Other drugs for obstructive airway diseases, inhalants in ATC |
| | - Lipid modifying agents, plain |
| | - Antithrombotic agents |
| | - Angiotensin II receptor blockers (ARBs), plain |
| | Visit 1: |
| | Conditions: |
| | - Congestive heart failure; nonhypertensive |
| | - Cardiac dysrhythmias |
| | - Shock |
| | - Alcohol-related disorders |
| | - Diabetes mellitus with complications |
| | - E codes: adverse effects of medical care |
| | - Disorders of teeth and jaw |
| | Procedures: |
| | - Other vascular catheterization; not heart |
| | - Blood transfusion Drugs: |
| | - Other drugs for obstructive airway diseases, inhalants in ATC |
| | - Other analgesics and antipyretics in ATC |
| | - Drugs for peptic ulcer and gastro-oesophageal reflux disease (GORD) |
| | - Beta blocking agents |
| | - Potassium supplements |

**Table 7 – continued from previous page**

| Case | Description |
|---|---|
| | **Supplementary Information**: [PLACEHOLDER]
**Patient References**: [PLACEHOLDER]
**Format**:
# Reasoning #
[Your reasoning here]
# Prediction #
[Your prediction here (1/0)] |
| Case (1): Zero-Shot Prompting with EHR Base Context Only | As provided, Sonnet 3.5 gave the following incorrect prediction:
**Model Output**:
# Reasoning #
While this patient's condition is serious and deteriorating, there are several factors to consider:
**Severity of conditions:**

• The patient has multiple severe conditions, including congestive heart failure, cancer (prostate and esophagus), and cardiac issues.

• New complications have arisen in visit 1, such as shock and cardiac dysrhythmias, which are concerning.

• However, some of the most immediately life-threatening conditions (e.g., sepsis, advanced stage cancer, multi-organ failure) are not explicitly mentioned.

**Procedures:**

• The procedures in visit 1 (blood transfusion and vascular catheterization) suggest a decline in health, but they are also interventions aimed at stabilizing the patient.

• There's no mention of extreme life-sustaining procedures like mechanical ventilation or dialysis.

**Medications:**

• The medications listed are primarily for managing chronic conditions rather than end-of-life care.

• There's no mention of high-risk medications like chemotherapy drugs or strong opioids for palliative care.

**Disease progression:**

• While there's a clear decline between visits 0 and 1, the conditions in visit 1, while serious, are potentially manageable with appropriate medical intervention.

**Absence of certain high-mortality indicators:**

• There's no mention of conditions like sepsis, acute respiratory failure, or advanced stage metastatic cancer, which would significantly increase mortality risk.

Given the task description's note to only predict mortality (1) for patients with "extremely very high risk of mortality", this patient's condition, while serious, does not meet that extremely high threshold. The presence of active interventions (blood transfusion, medications for managing conditions) suggests ongoing treatment rather than purely palliative care.

# Prediction #
**0** (incorrect) |
| Case (2): Zero-Shot Prompting with Classic RAG-Augmented Context (Knowledge Source: PubMed Abstract) | **Supplementary Information**:

• Rank 1, Similarity: 0.7204270362854004 PMID: 314160 Title: Resection of posterior ventricular aneurysm, replacement of mitral valve, and coronary bypass. Authors: C M Chavez, G Rodriquez, J H Conn Abstract: A 58-year-old man had sudden and progressive heart failure after a severe myocardial infarction. Aggressive medical treatment consisting of diuretics, vasopressors, and digitalis failed to improve his condition significantly. Cardiac catheterization disclosed a critical stenosis in the left anterior descending branch of the left coronary artery, a large posterior left ventricul aneurysm, and severe mitral insufficiency. Intermittent third degree heart block developed after admission. Surgical correction resulted in a dramatic recovery, and three years after operation he is fully recovered and asymptomatic. Publication Year: 1979

• Rank 2, Similarity: 0.7123403549194336 PMID: 804881 Title: [Acute coronary thrombosis in a 28 year-old woman]. Authors: G Motté, M Vogel, G Coatantiec, P Mariette Abstract: A 28-year-old woman, with no previous cardiovascular history, was hospitalized for myocardial infarction complicated by bifascicular block followed by complete atrio-ventricular block with a regressive course. A coronary arteriography performed on the 10th day demonstrated a thrombosis of the anterior interventricular artery, the rest of the coronary network being normal. The influence of a dyslipidaemia and the taking of oral contraceptives was discussed as an aetiology. Publication Year: 1975 |

**Table 7 – continued from previous page**

| Case | Description |
| --- | --- |

- Rank 3, Similarity: 0.7102002501487732 PMID: 42254 Title: [Non arrhythmogenic sudden death as complication of coronary heart disease]. Authors: H Zilcher, D Glogar Abstract: In a cohort of 417 patients admitted consecutively to the Coronary Care Unit for acute myocardial ischemia (unstable angina pectoris in 121, acute myocardial infarction in 296 patients) 21 cases of non arrhythmogenic sudden death occurred within 24 hours after admission. 16 of these patients suffered from acute myocardial infarction and 5 from unstable angina pectoris. Cause of death was cardiac rupture in 12 and pump failure in 4 patients with acute myocardial infarction, whereas all patients with unstable angina pectoris died from pump failure. Patients with cardiac rupture within 24 hours after admission, had significantly higher systolic and diastolic blood pressure in comparison with the other groups and with patients dying from cardiac rupture on the third day, or later. All patients dying from pump failure with unstable angina pectoris and one of the patients dying from pump failure with acute myocardial infarction had beta blocker therapy. ... Publication Year: 1979

- (...)

- Rank 9, Similarity: 0.6921100616455078 PMID: 443259 Title: Chest pain, shock, arrhythmias and death in a young woman. Authors: Abstract: 9 figures form the core of this article describing and discussing a case of sudden death, 2 hours after a 30-year old woman presented at a hospital emergency with chest pains. She had taken no medications other than oral contraceptives (OCs) for 10 years. The patient was admitted to the coronary care unit where findings included a palpable blood pressure of 94 mm of Hg, a heart rate of 128/min, and a respiratory rate of 28/minute. Cyanosis was noted, jugular veins were distended, and there were rales over the lung bases bilaterally; cardiac sounds were soft and a third heart sound was audible. Arterial oxygen tension was 15 mm of Hg, and carbon dioxide tension was 42 mm of Hg; pH was 7.2. Ventricular tachycardia developed and ventricular fibrillation ensued. The patient was intubated and well oxygenated, external cardiac compression was performed, sodium bicarbonate, epinephrine, and calcium were administered, and electrical defibrillation was performed. After several attempts, the latter resulted in a slow idioventricular rhythm on the electrocardiogram, but neither the blood pressure nor pulse was detectable. ... Publication Year: 1979

- Rank 10, Similarity: 0.6920691132545471 PMID: 938220 Title: Left ventricular function during aortic surgery. Authors: R M Carroll, R B Laravuso, J F Schauble Abstract: Fourteen patients undergoing surgery for aneurysm or occlusive disease of the abdominal aorta were studied. Thirteen patients had a history of hypertension or myocardial infarction; two patients had chronic obstructive pulmonary disease. Tachycardia, hypertension, and elevated pulmonary artery occluded. (PAo) pressure occurred in response to laryngoscopy and intubation in two patients; elevation of PAo pressure in response to aortic cross-clamping occurred in two patients. In three of these four patients, electrocardiographic evidence of myocardial ischemia appeared. These events are important in a consideration of the occurence of myocardial infarction in patients undergoing abdominal aortic surgery. Satisfactory treatment of myocardial ischemia has been accomplished with the use of propranolol hydrochloride and sodium nitroprusside. Publication Year: 1976

**Model Output**:
# Reasoning #
Based on the patient's context and the given task description, I will analyze the key factors to predict mortality risk for the subsequent hospital visit:

**Conditions**:

- The patient has multiple serious conditions, including congestive heart failure (nonhypertensive), cardiac dysrhythmias, and shock. These are severe cardiovascular issues that significantly increase mortality risk.

- The presence of diabetes mellitus with complications adds to the overall complexity of the patient's health status.

- Alcohol-related disorders may complicate treatment and recovery.

**Procedures**:

- The patient underwent blood transfusion, which might indicate severe blood loss or anemia, potentially linked to the shock condition.

- Other vascular catheterization (not heart) was performed, possibly related to managing circulatory issues.

- The supplementary materials discuss these procedures but focus on their use in orthopedic surgeries.

**Medications**:

- The patient is on multiple medications, including beta-blocking agents and drugs for obstructive airway diseases, indicating complex health issues.

- Potassium supplements might be linked to managing electrolyte imbalances, possibly due to heart failure or other medications.

**Progression from previous visit**:

- The patient's condition appears to have worsened since the previous visit, with new serious conditions like shock and cardiac dysrhythmias appearing.

**Comorbidities**:

- The combination of cardiovascular issues, diabetes complications, and alcohol-related disorders creates a complex clinical picture with multiple interacting risk factors.

**Table 7 – continued from previous page**

| Case | Description |
|---|---|
| | However, there are some factors that suggest the patient may not be at the highest risk of mortality:
- The patient is receiving active treatment, including medications for their conditions.
- There's no mention of advanced cancer or other terminal illnesses.
- The procedures mentioned are not highly invasive or indicative of end-stage disease.
- There's no indication of mechanical ventilation or other life-sustaining procedures.

# Prediction #
**0** (incorrect) |
| Case (3): Zero-Shot Prompting with KARE-Augmented Context (Knowledge Source: KG Communities) | **Supplementary Information**:

• Patients with congestive heart failure combined with cardiac dysrhythmias and shock have a significantly increased risk of mortality due to impaired cardiac output and hemodynamic instability.

• The presence of esophageal cancer, especially when co-occurring with prostate cancer, indicates a higher burden of malignancy, which can exacerbate overall health decline.

• Complications of devices, implants, or grafts can lead to severe infections or organ dysfunction, which is life-threatening, especially in patients with weakened cardiac function.

• The combination of anemia and thyroid disorders can exacerbate cardiovascular conditions, leading to poorer outcomes in heart failure patients.

• Alcohol-related disorders can negatively impact medication adherence and exacerbate conditions like hypertension and diabetes, increasing the risk of death.

• The use of blood transfusions and vascular catheterization procedures indicates severe clinical interventions often linked to critical conditions and a higher risk of death.

• Polypharmacy, including beta-blockers, potassium supplements, and medications for obstructive airway diseases, reflects complex medical management, which can be associated with increased risk of adverse events.

• Patients with diabetes mellitus with complications face a greater risk of serious outcomes due to potential organ damage and delayed healing processes.

**Model Output**:
# Reasoning #
Based on the patient's context and the supplementary information:
**Severity of Conditions**:

• The patient has congestive heart failure, cardiac dysrhythmias, and shock, which together significantly impair cardiac function and increase mortality risk.

• The presence of two cancers (esophageal and prostate) indicates a high burden of malignancy, worsening the patient's overall prognosis.

• Complications of devices, implants, or grafts may lead to severe infections or organ dysfunction.

**Interactions Between Conditions**:

• Anemia and thyroid disorders can worsen heart failure outcomes.

• Alcohol-related disorders may impair medication adherence and liver function, complicating management of other conditions.

• Diabetes mellitus with complications further increases mortality risk due to potential organ damage.

**Procedures**:

• The need for blood transfusions and vascular catheterization suggests critical interventions due to severe anemia or hemodynamic instability.

• These procedures are associated with higher mortality rates in critically ill patients.

**Medications**:

• Polypharmacy with beta-blockers, potassium supplements, and other medications indicates complex management and high risk of adverse events.

• Medications for obstructive airway diseases suggest respiratory complications.

The provided information highlights that this combination of severe conditions, multiple comorbidities, and invasive procedures is associated with a very high risk of mortality.
Given the critical combination of severe cardiac conditions, multiple cancers, complex comorbidities, and the necessity for invasive procedures, the patient is at an extremely high risk of mortality and is likely to die in the next visit.

# Prediction #
**1** (correct) |

**Table 7 – continued from previous page**

| Case | Description |
|------|-------------|
| Case (4): Few-Shot Prompting with Positive and Negative Patient References and KARE-Augmented Context | **Supplementary Information**: (Same as in Case 3) 
 **Patient References**: |

**Patient References**:

- **Example 1 (Mortality = 1)**:
    - **Patient ID**: 10269
    - **Visit Information**:
        * **Conditions**:
            · Congestive heart failure; nonhypertensive
            · Sepsis
            · Acute respiratory failure
            · Shock
        * **Procedures**:
            · Mechanical ventilation
            · Blood transfusion
            · Vascular catheterization
        * **Medications**:
            · High-dose vasopressors
            · Broad-spectrum antibiotics

- **Example 2 (Mortality = 0)**:
    - **Patient ID**: 11445
    - **Visit Information**:
        * **Conditions**:
            · Essential hypertension
            · Type 2 diabetes mellitus without complications
            · Hyperlipidemia
        * **Procedures**:
            · Routine blood tests
            · Electrocardiogram
        * **Medications**:
            · Metformin
            · ACE inhibitors

**Model Output**:
\# Reasoning \#
Comparing the new patient to the provided references:
**Similarity to Example 1:**

- Both patients have congestive heart failure and shock.
- Both require invasive procedures like blood transfusions and vascular catheterization.
- The patient 10269 has acute respiratory failure and sepsis, while the new patient has multiple cancers and cardiac dysrhythmias.
- Both have complex comorbidities and are undergoing critical interventions, indicating a high risk of mortality.

**Difference from Example 2:**

- The negative reference has manageable chronic conditions without severe complications.
- No invasive procedures or critical conditions are present.
- The new patient has significantly more severe conditions and interventions.

**Patient's Conditions:**

- The combination of congestive heart failure, cardiac dysrhythmias, shock, and multiple cancers places the patient at a very high risk of mortality.
- Disorders linked to alcohol use may reduce compliance with prescribed medications and worsen chronic illnesses such as high blood pressure and diabetes, elevating mortality risk.
- Diabetes mellitus with complications and alcohol-related disorders further exacerbate health risks.

**Procedures and Medications:**

- Invasive procedures and polypharmacy indicate critical illness and complex management.
- Similar to the positive reference, these factors contribute to a higher mortality risk.

The provided knowledge and patient references highlight that patients with this combination of severe conditions and interventions are likely to die. Given the strong similarities to the positive reference and the critical nature of the patient's health status, it is highly likely that the patient will die in the next visit.

\# Prediction \#
**1** (correct)

| Case | Description |
|------|-------------|
| Analysis | - **Case (2)** demonstrates that when using traditional RAG methods with sparse retrieval from PubMed abstracts, the supplementary information contains several keywords matching the patient's conditions but fails to provide relevant information regarding the patient's high mortality risk. |

Continued on next page

**Table 7 – continued from previous page**

| Case | Description |
|---|---|
| | - **Case (3)** shows that KARE's dense and relevant knowledge retrieval provides comprehensive insights into the patient's condition, interactions between comorbidities, and the implications of procedures and medications, leading to a correct prediction.
- **Case (4)** illustrates that including patient references with KARE-augmented context further aids the model by providing concrete examples to compare against, enhancing reasoning and ensuring a correct prediction. |
| Conclusion | This case study emphasizes the importance of KARE's approach in providing dense, relevant knowledge that significantly improves the model's ability to make accurate predictions. By integrating comprehensive knowledge graphs and considering complex interactions between medical concepts, KARE enhances reasoning capabilities beyond what is achievable with base context or traditional RAG methods. |

# G  TEMPLATES, PROMPTS, AND EXAMPLES

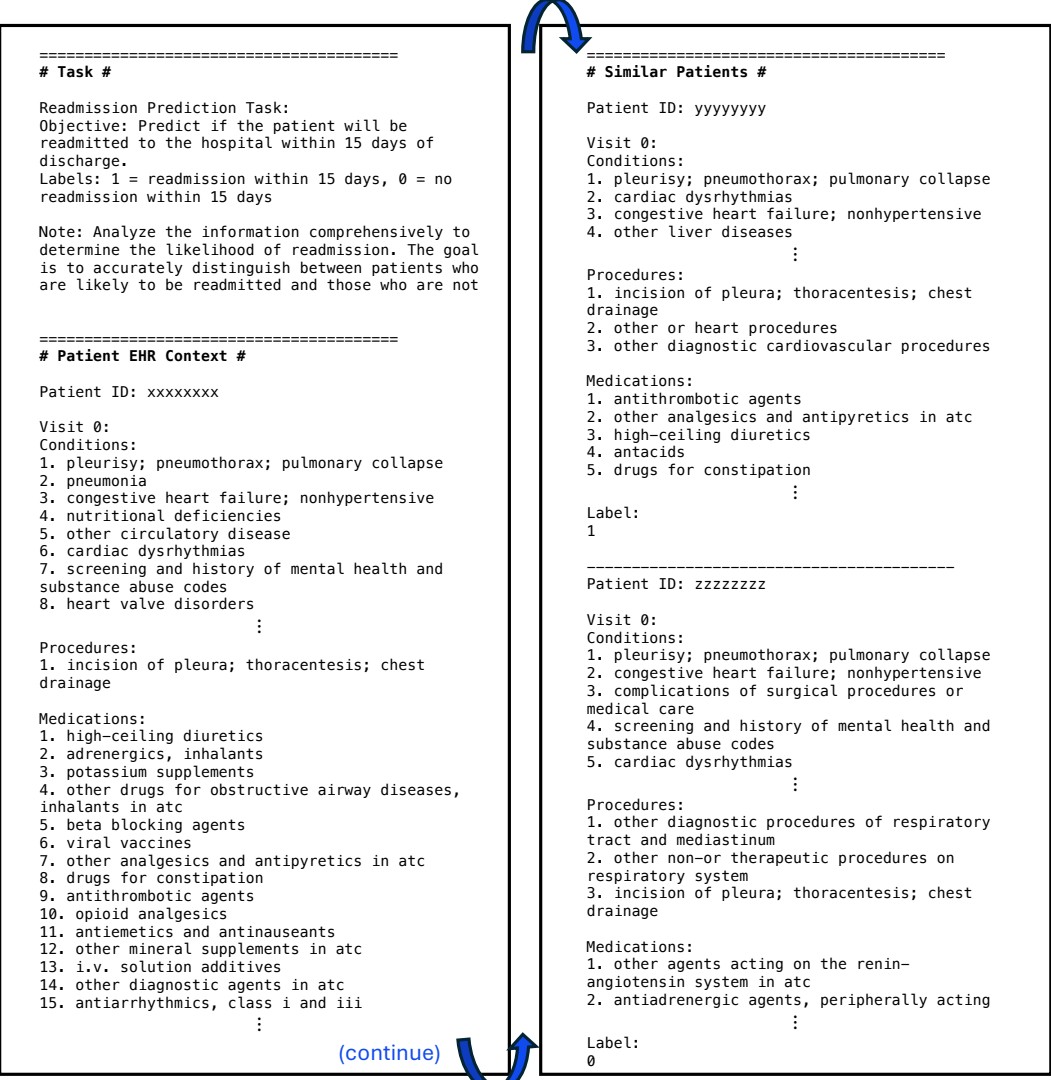

Figure 11: An example of the patient base context.

| **General Summarization** | **Theme-Specific Summarization** |
|---|---|
| You are a knowledgeable medical assistant tasked with generating a comprehensive summary of the medical concepts and relationships provided below.

Given a list of medical triples in the format (entity1, relationship, entity2), please create a single, coherent summary that captures the key medical knowledge represented by these concepts and their relationships. The summary should be written in the third person and include all the entity names for full context.

Focus on how these medical concepts are interconnected and their relevance to medical understanding or patient care. If there are any contradictions in the triples, please resolve them in the summary.

Please provide only the summary without any starting words or phrases, and without mentioning the individual triples.

The summary should be an integrated representation of the medical knowledge contained in the triples.

Example:
Triples:
- (Diabetes, is a risk factor for, Cardiovascular Disease)
- (Hypertension, is associated with, Diabetes)
- (Obesity, contributes to, Diabetes)

Summary:
Diabetes, hypertension, and obesity are closely interconnected medical conditions. Diabetes is a significant risk factor for developing cardiovascular disease. Hypertension, or high blood pressure, is often associated with diabetes. Obesity contributes to the development of diabetes, as excess body weight can lead to insulin resistance. Managing these conditions together is crucial for reducing the risk of serious complications and improving overall patient care.

Triples:
**{Formatted Triples}**

Summary: | You are a knowledgeable medical assistant tasked with generating a theme-specific summary of the medical concepts and relationships provided below.

Given a list of medical triples in the format (entity1, relationship, entity2) and a specific theme, please create a summary that focuses on how the knowledge represented by these triples is relevant to the given theme. The summary should highlight the key concepts, relationships, and implications that are most pertinent to the theme.

The summary should be written in the third person and include all the relevant entity names for context. Please provide only the summary without any starting words or phrases, and without mentioning the individual triples.

Examples:

Mortality Prediction:
Triples:
- (Diabetes, is a risk factor for, Cardiovascular Disease)
- (Hypertension, is associated with, Diabetes)
- (Obesity, contributes to, Diabetes)

Theme: Mortality prediction
**{Theme Description}**

Summary:
Diabetes, hypertension, and obesity are significant risk factors that can increase the likelihood of mortality. Diabetes is directly associated with an increased risk of cardiovascular disease, which is a leading cause of death. Hypertension and obesity, which often co-occur with diabetes, further compound these risks. Patients with these conditions require close monitoring and aggressive management to mitigate the risk of mortality.

Triples:
**{Formatted Triples}**

Theme: **{Theme}**

Summary: |

Figure 12: The prompts for (*left*) general and (*right*) theme-specific KG community summarization.

| **Summary Combination for Large Community** |
|---|
| You are a knowledgeable medical assistant tasked with generating a comprehensive summary of the medical concepts and relationships based on the individual summaries provided below.

Given the following summaries of a community, please create a single, coherent summary that captures the key medical knowledge represented by this community. The combined summary should include all the important entities and relationships mentioned in the individual summaries.

Focus on the main concepts, their relationships, and their relevance to medical understanding or patient care. The summary should be concise yet informative, providing a comprehensive overview of the community's medical knowledge.

Please ensure that the combined summary is written in the third person and includes all the relevant entity names for full context.

Please provide only the summary without any starting words or phrases.

Summaries:
**{summaries}**

Combined Summary: |

Figure 13: The prompt for summary combination for large communities.

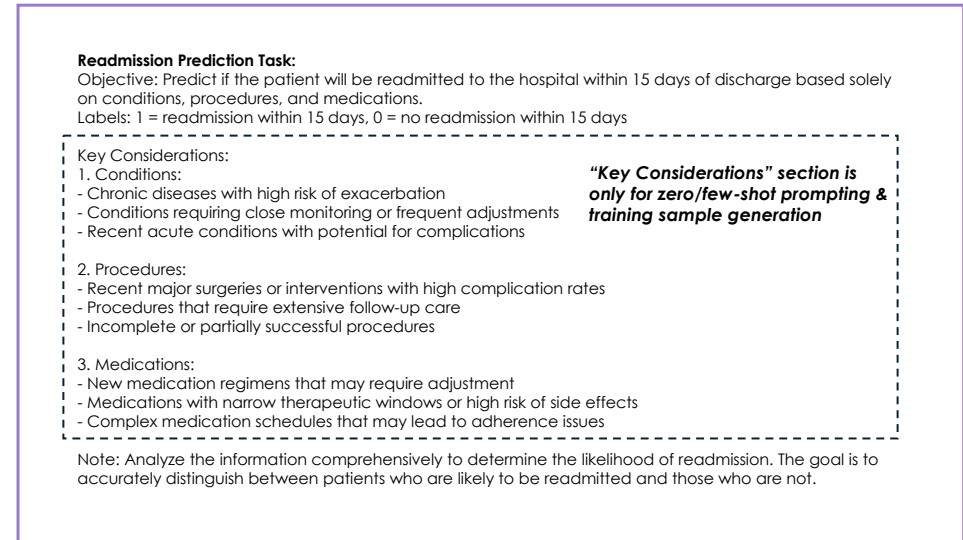

**KG Community**

Hypertension — cause → Chronic Kidney Disease ← cause — Diabetes

Chronic Kidney Disease — leads to → Anemia

Iron Supplementation — treat → Anemia

**Theme-Specific Summarization**

**Theme: Mortality**

"Diabetes and Hypertension are major risk factors for Chronic Kidney Disease (CKD). As CKD progresses, it can lead to complications such as Anemia, which can further increase morbidity and mortality. Early detection and management of Diabetes and Hypertension are crucial in preventing the development and progression of CKD, ultimately reducing the risk of premature death in these patient populations."

**Theme: Readmission**

"Chronic Kidney Disease (CKD), commonly caused by Diabetes and Hypertension, can lead to Anemia, characterized by a decreased number of red blood cells. Anemia may cause symptoms such as fatigue and shortness of breath, increasing the risk of hospital readmissions. To prevent readmissions in CKD patients, it is essential to manage the underlying causes, monitor hemoglobin levels, treat Anemia with Iron Supplementation, and ensure treatment adherence."

Figure 14: An example of theme-specific summarization of knowledge graph community.

**Task Description – Mortality Prediction**

**Mortality Prediction Task:**
Objective: Predict the mortality outcome for a patient's subsequent hospital visit based solely on conditions, procedures, and medications.
Labels: 1 = mortality, 0 = survival

Key Considerations:
1. Conditions:
- Severity of diagnosed conditions (e.g., advanced cancer, severe heart failure, sepsis)
- Presence of multiple comorbidities
- Acute vs. chronic nature of conditions

*"Key Considerations" section is only for zero/few-shot prompting & training sample generation*

2. Procedures:
- Invasiveness and complexity of recent procedures
- Emergency vs. elective procedures
- Frequency of life-sustaining procedures (e.g., dialysis, mechanical ventilation)

3. Medications:
- Use of high-risk medications (e.g., chemotherapy drugs, immunosuppressants)
- Multiple medication use indicating complex health issues
- Presence of medications typically used in end-of-life care

Note: Focus on combinations of conditions, procedures, and medications that indicate critical illness or a high risk of mortality. Consider how these factors interact and potentially exacerbate each other. Only the patients with extremely very high risk of mortality should be predicted as 1.

**Task Description – Readmission Prediction**

**Readmission Prediction Task:**
Objective: Predict if the patient will be readmitted to the hospital within 15 days of discharge based solely on conditions, procedures, and medications.
Labels: 1 = readmission within 15 days, 0 = no readmission within 15 days

Key Considerations:
1. Conditions:
- Chronic diseases with high risk of exacerbation
- Conditions requiring close monitoring or frequent adjustments
- Recent acute conditions with potential for complications

*"Key Considerations" section is only for zero/few-shot prompting & training sample generation*

2. Procedures:
- Recent major surgeries or interventions with high complication rates
- Procedures that require extensive follow-up care
- Incomplete or partially successful procedures

3. Medications:
- New medication regimens that may require adjustment
- Medications with narrow therapeutic windows or high risk of side effects
- Complex medication schedules that may lead to adherence issues

Note: Analyze the information comprehensively to determine the likelihood of readmission. The goal is to accurately distinguish between patients who are likely to be readmitted and those who are not.

Figure 15: Task descriptions of the mortality prediction and the readmission prediction tasks used in the paper. *Note*: For the fine-tuning process, we do not include the "Key Considerations" section in the input template.

Given the following task description, patient EHR context, similar patients, retrieved medical knowledge, and ground truth label, provide a step-by-step reasoning process that leads to the correct prediction:

```
=======================================
# Task #
{task_description}
=======================================
# Patient EHR Context #
{context}
=======================================
# Similar Patients #
{similar_patients}
=======================================
# Retrieved Medical Knowledge #
{medical_knowledge}
=======================================
# Ground Truth #
{ground_truth}
=======================================
```

Please provide a step-by-step reasoning process that leads to the correct prediction based on the patient's context, similar patients, and the retrieved relevant medical knowledge.

The reasoning chain should follow this structured format:

1. **Patient Overview**: Check the key information in the patient's context, with the Key Considerations from the task description in mind.
2. **Relevant Retrieved Medical Knowledge**: Highlight the retrieved medical knowledge pertinent to the patient's condition.
3. **Comparison with Similar Patients**: Analyze the similarities and differences between the patient and similar patients, explaining how these factors influence the prediction.
4. **Reasoning Towards Prediction**: Integrate the above information to logically reason towards the predicted outcome.
5. **Conclusion**: Summarize the reasoning and state the prediction without mentioning the ground truth.

The reasoning should be comprehensive, medically sound, and clearly explain how the patient's information leads to the predicted outcome.

**Important Notes:**
- Do not mention the ground truth label in the reasoning process.
- Use the relevant knowledge as needed.
- Analyze the similarities and differences between the patient and similar patients to justify the prediction.

After generating the reasoning chain, please review it and indicate your confidence in the reasoning chain at the end.

Options of confidence: [Very Confident, Confident, Neutral, Not Confident, Very Not Confident.]

**Output Format:**

# Reasoning Chain #

1. Patient Overview:
[YOUR OUTPUT]

2. Relevant Retrieved Medical Knowledge:
[YOUR OUTPUT]

3. Comparison with Similar Patients:
[YOUR OUTPUT]

4. Reasoning Towards Prediction:
[YOUR OUTPUT]

5. Conclusion:
[YOUR OUTPUT]

# Confidence #
[CONFIDENCE ("Very Confident", "Confident", "Neutral", "Not Confident", "Very Not Confident")]

Figure 16: Prompt used for reasoning chain generation for training sample.

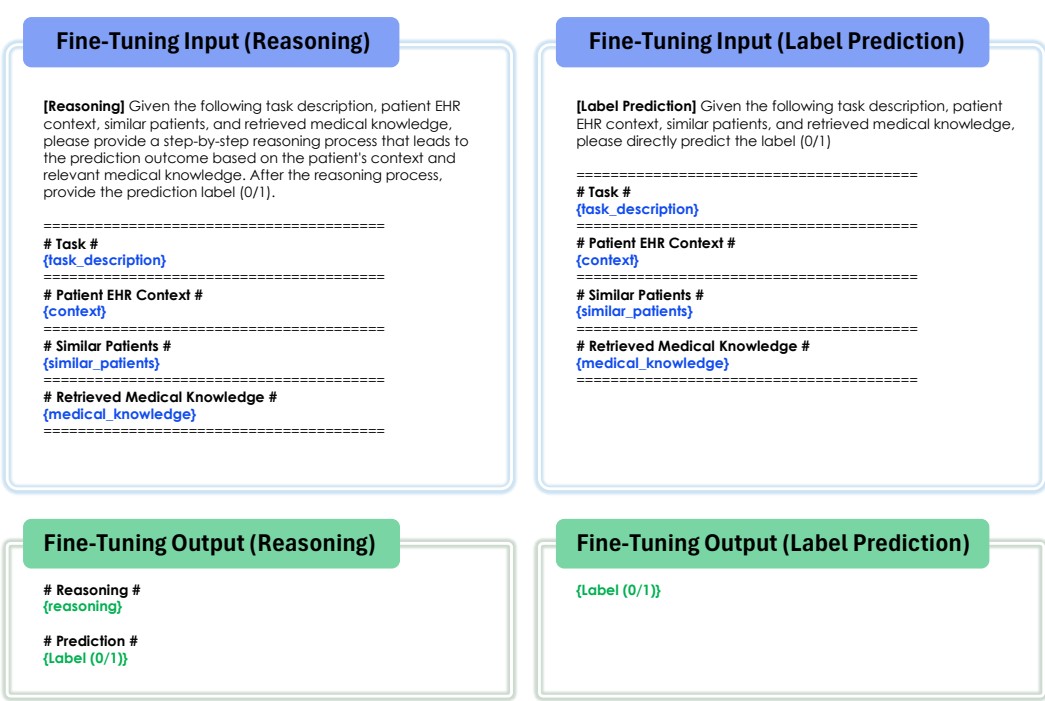

Figure 17: Template used for the input and output of fine-tuning. (To improve the reproducibility of KARE, we will publicize the processed data for fine-tuning the local LLM through PhysioNet)

## H  HUMAN EVALUATION OF REASONING CHAINS

We hired three MD students and an MD to conduct expert evaluation of the reasoning chain generated by KARE. We instructed them to use the same metrics introduced by Kwon et al. (2024) to evaluate 100 reasoning chains. All the evaluated samples are from test sets. Half of the chains were for mortality prediction, and the other half were for readmission prediction. For each task, 35 chains were positive (correctly predicted), and 15 chains were negative (incorrectly predicted). The definitions of the metrics and rating scales are described as follows:

- **CONSISTENCY**
  **Definition**: Consistency measures how well the generated rationale aligns with the presented data and the model's prediction. It evaluates whether the reasoning contains contradictions.
  - **5 (Excellent)**: The rationale is entirely consistent with the provided data and prediction. No contradictions are present.
  - **4 (Good)**: Minor inconsistencies are present but do not significantly detract from the overall coherence.
  - **3 (Fair)**: Some inconsistencies exist that partially undermine the rationale's reliability.
  - **2 (Poor)**: Significant inconsistencies exist, making the rationale difficult to trust.
  - **1 (Very Poor)**: The rationale is fundamentally contradictory to the data or prediction.

- **CORRECTNESS**
  **Definition**: Correctness assesses the medical validity of the knowledge and reasoning presented in the rationale.
  - **5 (Excellent)**: The rationale is entirely medically accurate and reflects evidence-based clinical knowledge.
  - **4 (Good)**: Minor inaccuracies are present but do not affect the overall clinical validity.
  - **3 (Fair)**: The rationale contains some medically incorrect statements that could mislead clinicians.
  - **2 (Poor)**: Multiple inaccuracies significantly reduce the credibility of the rationale.
  - **1 (Very Poor)**: The rationale is largely or entirely medically incorrect, making it unusable.

- **SPECIFICITY**
  **Definition**: Specificity evaluates how detailed and precise the reasoning is in addressing the clinical scenario.
  - **5 (Excellent)**: The rationale provides highly detailed and tailored insights specific to the patient case.
  - **4 (Good)**: The rationale is detailed but occasionally includes generalities.
  - **3 (Fair)**: The rationale is moderately specific, with noticeable generalizations.
  - **2 (Poor)**: The rationale is vague and lacks sufficient detail to guide clinical decision-making.
  - **1 (Very Poor)**: The rationale is overly generic and lacks any meaningful detail.

- **HELPFULNESS**
  **Definition**: Helpfulness measures the extent to which the rationale aids the prediction toward the correct diagnosis.
  - **5 (Excellent)**: The rationale strongly supports the correct prediction and adds valuable clinical insights.
  - **4 (Good)**: The rationale is helpful overall but lacks some critical insights.
  - **3 (Fair)**: The rationale provides some useful guidance but is incomplete or not compelling.
  - **2 (Poor)**: The rationale adds minimal value to the prediction and lacks actionable insights.
  - **1 (Very Poor)**: The rationale is unhelpful and does not contribute meaningfully to the diagnosis.

- **HUMAN-LIKENESS**
  **Definition**: Human-likeness measures how well the clinical rationale demonstrates insight and understanding of the patient description or diagnosis in a way that resembles human clinical reasoning.
  - **5 (Excellent)**: The rationale reflects deep clinical insight, contextual understanding, and reasoning that fully mimics human behavior.

- **4 (Good)**: The rationale generally matches human reasoning but lacks minor elements of nuanced understanding.
- **3 (Fair)**: The rationale demonstrates basic human-like reasoning but misses several critical elements of insight.
- **2 (Poor)**: The rationale shows limited resemblance to human reasoning and lacks essential clinical insight.
- **1 (Very Poor)**: The rationale fails to resemble human clinical reasoning and demonstrates no meaningful understanding of the patient description or diagnosis.

The evaluation results are shown in Fig. 18.

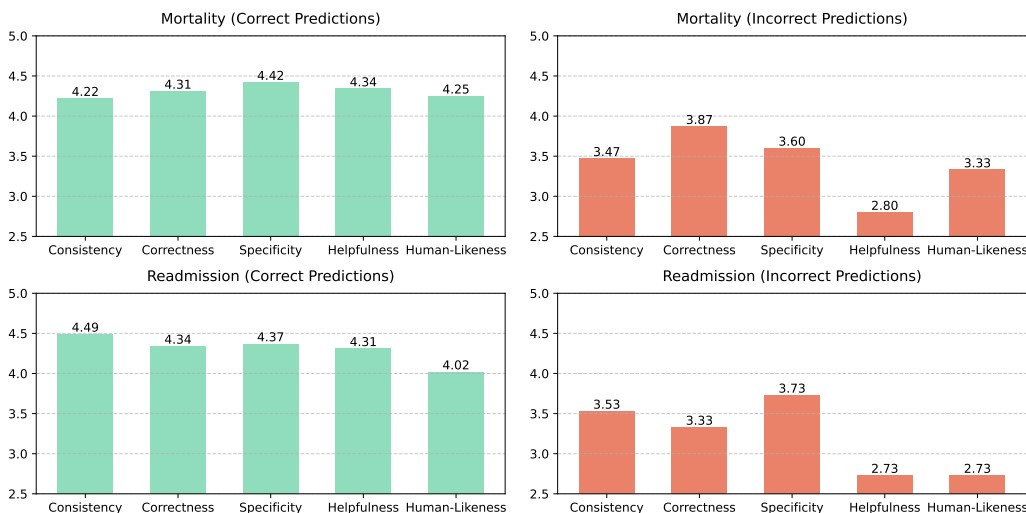

Figure 18: Evaluation of reasoning chains generated by KARE.

**Discussions:**

1. Reasoning chains leading to incorrect clinical predictions (negative cases) consistently score lower across all metrics. This highlights the critical role of high-quality reasoning chains in ensuring prediction accuracy. Enhancing the consistency and correctness of these chains could potentially improve the overall prediction performance.

2. Human-Likeness scores are notably lower for the readmission prediction task. According to the experts we consulted, this is primarily because determining whether a patient will be readmitted within 15 days is inherently challenging, even for experienced clinicians, given the limited information provided in the patient context. Factors like the absence of basic demographic details (e.g., gender and age) further complicate this task. Despite these limitations, KARE demonstrates a remarkable ability to outperform clinicians in predicting readmissions, showcasing its potential in information-scarce scenarios.

3. For the mortality prediction task, inconsistencies between the reasoning chains and the final predictions were observed in certain cases. For instance, some reasoning chains concluded that the patient would survive the next visit, yet the final predicted label was "1" (indicating mortality). These discrepancies negatively impacted the consistency scores. Incorporating an additional verification step to align the reasoning chain with the final prediction may help address this issue and enhance overall reliability.

# I  PARAMETERS & TRAINING TIME OF MODELS

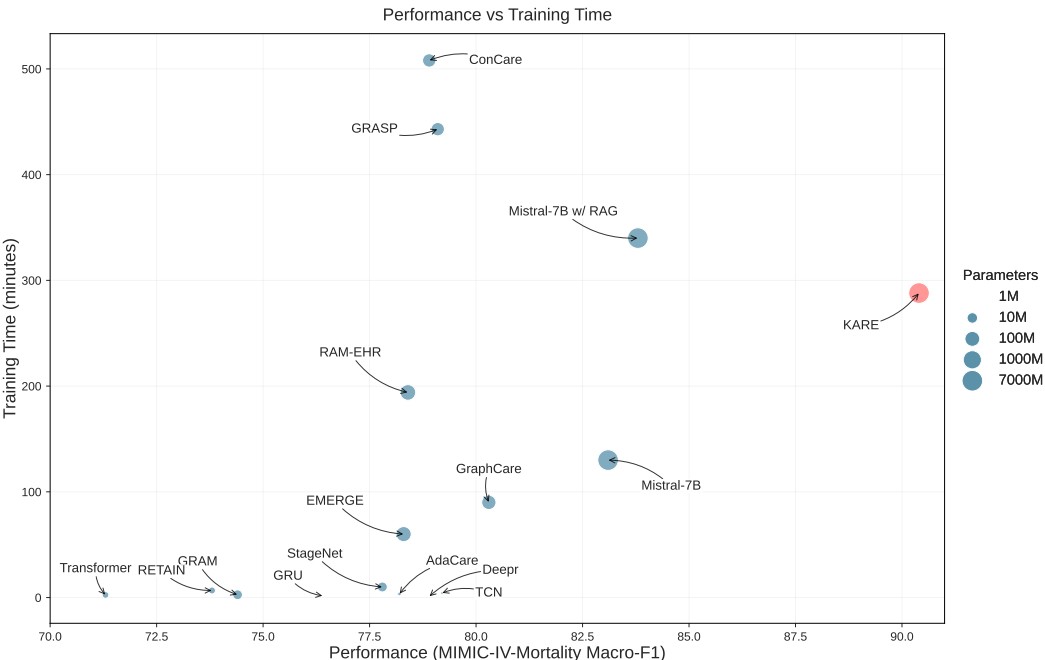

Figure 19: **Performance vs. Training Time.** Training time refers to the duration required for the model to achieve its optimal performance on the validation set.

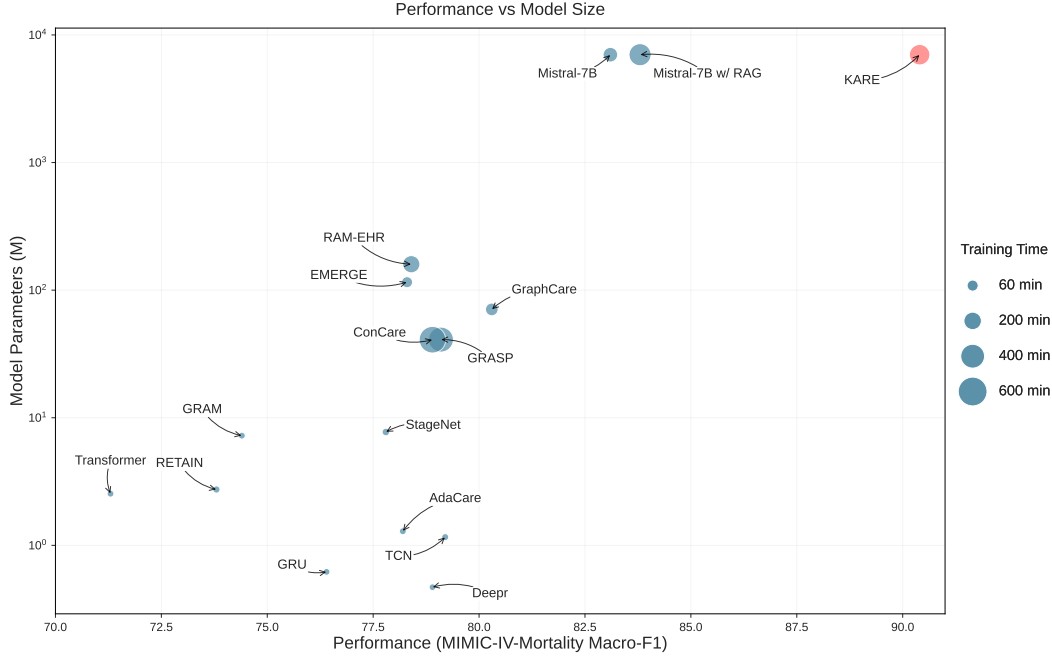

Figure 20: **Performance vs. Model Parameters.**

## J NOTATIONS

Table 8: Notations used in our paper.

| Notation | Description |
|---|---|
| $G', V', R', E'$ | Knowledge graph before semantic clustering and its components |
| $G, V, R, E$ | Refined knowledge graph after semantic clustering and its components |
| $G_{c_i}, V_{c_i}, E_{c_i}$ | Concept-specific knowledge graph for concept $c_i$ and its components |
| $G_{c_i}^{\text{KG}}$ | Subgraph of $G_{c_i}$ from a biomedical knowledge graph |
| $G_{c_i}^{\text{BC}}$ | Subgraph of $G_{c_i}$ extracted from a biomedical corpus |
| $G_{c_i}^{\text{LLM}}$ | Subgraph of $G_{c_i}$ extracted using a large language model |
| $G_p, V_{G_p}$ | Patient-specific knowledge graph for patient $p$ and its entities |
| $\mathcal{C}_l^m, C_{k,m}^l$ | Set of communities and the $k$-th community at level $l$ in run $m$ |
| $\mathcal{C}, C_k, C_{\text{best}}$ | Set of all communities, a community, and the best community |
| $V_{C_k}, S_{C_k}$ | Entities and summary of community $C_k$ |
| $\mathcal{B}_p, \mathcal{A}_p$ | Base and augmented context for patient $p$ |
| $S_p$ | Selected community summaries for patient $p$ |
| $\mathbf{e}_i, \mathbf{e}_j, e(\cdot)$ | Text embedding of entity $i$ / relation $j$ and embedding function |
| $H(v)$ | Hit count of node $v$ in previous selections |
| $\tau, \mathcal{T}_\tau$ | Healthcare prediction task (theme) and its representative terms |
| $y_{p,\tau}^*, y_{p,\tau}$ | Ground truth and predicted labels for patient $p$ and task $\tau$ |
| $\rho_{p,\tau,k}, \rho_{p,\tau}^{\text{best}}$ | The $k$-th and best reasoning chains for patient $p$ and task $\tau$ |
| $c_i, \mathbf{C}$ | A medical concept and the set of all concepts |
| $R_{c_i}$ | Top $X$ co-existing concepts for concept $c_i$ |
| $\phi_e, \phi_r$ | Mappings from original entities and relations to cluster representatives |
| $p_{ij}$ | Shortest path between concepts $c_i$ and $c_j$ |
| $L, M$ | Number of levels and runs in community detection |
| $Z_c, Z_s$ | Maximum triples per community, triples per summary |
| $\theta_e, \theta_r$ | Clustering thresholds for entities and relations |
| $\alpha, \beta, \lambda_1, \lambda_2, \lambda_3$ | Hyperparameters for context augmentation |

