# OpenReview forum: "Reasoning-Enhanced Healthcare Predictions with Knowledge Graph Community Retrieval"
_ICLR.cc/2025/Conference — ICLR 2025 Poster_

### Official Review · Reviewer_oKgq · 2024-10-30

**Soundness:** 3
**Presentation:** 3
**Contribution:** 3
**Rating:** 8
**Confidence:** 4

**Summary:**

In this manuscript, the authors propose KARE, a knowledge graph-enhanced large language model designed for predicting patient mortality and readmission, with an added aim to mitigate LLM hallucinations. The paper introduces a novel approach that effectively combines knowledge graphs with LLMs through a clustering method. Additionally, the authors present a reasoning-chain mechanism to enhance the LLM's inference capabilities and provide interpretable prediction results. Experimental results demonstrate that KARE achieves a marginal improvement in mortality and readmission predictions on the MIMIC-III and MIMIC-IV datasets. However, several major and minor issues need to be addressed. If author could well address my concern, I would like to update my ratings,

**Strengths:**

* The paper is well-written and structured.
* The authors propose a novel knowledge-graph learning framework combined with LLMs.
* The integration of knowledge augmentation and knowledge graphs is both novel and interesting.
* The experimental evaluation is thorough, conducted on both the MIMIC-III and MIMIC-IV datasets.

**Weaknesses:**

* The excessive use of symbols makes certain sections difficult to follow.
* The paper lacks a detailed description of longitudinal EHR processing with LLMs.
* The improvement in mortality prediction performance achieved by KARE is marginal.
* There is a lack of sensitivity analysis for hyperparameters, such as the number of top co-occurring concepts, top documents, and the criteria for selecting optimal values. Additional hyperparameters, such as maximum sequence length and maximum path count, should also be discussed, or at least identified explicitly for clarity.
* The paper would benefit from a more in-depth interpretability analysis to clarify the relationship between the generated reasoning, labels, and the knowledge graph.
* The paper lacks an evaluation of LLM hallucinations, despite the stated aim to address hallucination issues in clinical decision-making.
* An analysis of model parameters and time consumption is missing, which could provide valuable insights into the model’s computational efficiency and practical applicability
* There is a lack of references to related work.

  [1] . Kang, M., Lee, S., Baek, J., Kawaguchi, K., & Hwang, S. J. (2024). Knowledge-augmented reasoning distillation for small language models in knowledge-intensive tasks. Advances in Neural Information Processing Systems, 36.

  [2]. Niu, S., Ma, J., Bai, L., Wang, Z., Guo, L., & Yang, X. (2024). EHR-KnowGen: Knowledge-enhanced multimodal learning for disease diagnosis generation. Information Fusion, 102, 102069.

  [3]. Kwon, T., Ong, K. T. I., Kang, D., Moon, S., Lee, J. R., Hwang, D., ... & Yeo, J. (2024, March). Large language models are clinical reasoners: Reasoning-aware diagnosis framework with prompt-generated rationales. In Proceedings of the AAAI Conference on Artificial Intelligence (Vol. 38, No. 16, pp. 18417-18425).

**Questions:**

* In Section 3.2, how does the method handle cases where the same patient has more than two visits with different labels? Is there a specific design to distinguish similar symptoms in different patients and multiple visits from the same patient?
* In Section 3.2, how is the effectiveness of the relevance score (i.e., Formula (3)) validated? Would an ablation study on the components of Relevance(C_k) help to clarify this?
* How does the method utilize patient longitudinal visit information?
* For the baseline ML methods, it’s mentioned that most are implemented using PyHealth. Could you clarify the backbone model for each ML method? Also, is there a fair configuration in place for implementing language model-based encoders, such as ClinicalBERT, across these methods?
* The experimental results for mortality prediction show that the baseline ML methods, such as ConCare and TCN, perform closely to KARE, with some evaluation metrics even exceeding KARE’s. How should these results be interpreted?
* In experiment, sensitivity and specificity are indeed important metrics, but F1 score and accuracy are also widely used for assessing diagnostic accuracy. Interestingly, in ablation study, excluding similar patients often yields better or comparable results than including them, especially for mortality prediction on the MIMIC-III dataset. Do these results support the methodological choices made in Section 3.2?
* As you state that your base model is fine-tuned on Mistral-7B-Instruct-v0.3, I wonder about the performance of fine-tuning an LLM (Mistral or Llama) with RAG. Can your model still show substantial improvements compared to other LLM-based models beyond just zero-shot or few-shot settings? I am concerned that most of the current improvements may be attributed primarily to supervised fine-tuning.

---

> ### Author Response · Authors · 2024-11-16
> **Author Response to Reviewer oKgq (Part I)**
>
> **Author Response to Reviewer oKgq**
>
> Thank you for recognizing the strengths of our work! We address your concerns and answer your questions below. We also uploaded a revision and used blue to mark the new changes.
>
> ---
>
> ### For Weaknesses:
>
> > **[W1] The excessive use of symbols makes certain sections difficult to follow.**
>
> Thank you for this feedback. While we have provided a comprehensive notation table in Appendix I (Table 9), we will improve readability by:
>
> 1. Using more descriptive names and inline explanations in the main text
> 2. Adding illustrative examples when introducing new notation
> 3. Better visualizing symbol relationships in our figures
>
>
>
> > **[W2] The paper lacks a detailed description of longitudinal EHR processing with LLMs.**
>
> We respectfully note that our paper does address longitudinal EHR processing in several sections:
>
> 1. In Section 3.2, we process patient visits chronologically to construct the base context, integrating conditions, procedures, and medications across time. Figure 11 in Appendix G shows an example of the base context, which is structured to enable LLMs to clearly understand temporal relationships between visits.
>
> 2. When retrieving similar patients (Section 3.2), our framework considers complete visit histories to ensure meaningful temporal comparisons. Our LLM prompt templates (Figure 16) explicitly guide the model to analyze progression of conditions across visits.
>
> 3. Our dynamic graph retrieval approach (Algorithm 1) accounts for temporal relationships by incorporating visit-level recency in the relevance score (Equation 5).
>
> The longitudinal nature of EHR data is intrinsically handled in our approach through these integrated components, with our fine-tuning process (Section 3.3.2) ensuring the LLM learns to effectively reason about patient trajectories over time.
>
>
>
> > **[W3] The improvement in mortality prediction performance achieved by KARE is marginal.**
>
> We respectively disagree with this conclusion. Our improvements in mortality prediction are in fact substantial, particularly when considering the appropriate metrics for imbalanced datasets.
>
> As the mortality prediction datasets are imbalanced (5.42% and 19.16% positive labels (mortality=1) for MIMIC-III and MIMIC-IV respectively), we should not focus too heavily on metrics like accuracy. In fact, an untrained model that blindly predicts all patients will survive can still achieve 94.6% accuracy on MIMIC-III! This explains why ConCare, which overfits on this dataset, shows similar behavior.
>
> Instead, we should focus on metrics like Macro-F1 and Sensitivity to test the model's real prediction ability on imbalanced datasets. Particularly, **Sensitivity measures the model's ability to predict "whether this patient will die in the next visit" - a challenging task given the very few positive examples in the training data.** Our substantial improvements in these metrics (e.g., from 17.2% to 24.7% Sensitivity on MIMIC-III, from 57.8% to 73.2% Sensitivity on MIMIC-IV) demonstrate KARE's superior ability in identifying high-risk patients.
>
>
>
> > **[W4] There is a lack of sensitivity analysis for hyperparameters, such as the number of top co-occurring concepts, top documents, and the criteria for selecting optimal values. Additional hyperparameters, such as maximum sequence length and maximum path count, should also be discussed, or at least identified explicitly for clarity.**
>
> Thank you this suggestion. We note that our key hyperparameters are already documented in Section 3 and Appendix D:
>
> 1. KG Construction (Appendix B.1-B.3): top-20 co-existing concepts, top-3 documents per concept set, max path length=7, max paths=40
> 2. Community Detection (Section 3.1.3): maximum $Z_s$=20 triples per initial summary, $Z_c$=150 triples per community
> 3. Context Augmentation (Section 3.2): α=0.1, β=0.7, λ₁=0.2, λ₂=0.2, λ₃=0.3
> 4. Fine-tuning Parameters (Appendix D.3): full configuration provided in Table 6, including sequence length, learning rate, etc.
>
> Importantly, these parameters were determined through principled approaches rather than exhaustive search:
>
> - Community size thresholds ($Z_s$, $Z_c$) are based on LLM context window constraints
> - Clustering thresholds are optimized using silhouette scores
> - Context augmentation parameters are validated through LLM evaluation of retrieved information relevance and utility
> - Fine-tuning parameters follow standard recommendations for Mistral-7B instruction tuning
>
> This approach ensures parameter selection remains manageable while maintaining model performance.
>
> Nevertheless, we agree that adding sensitivity analysis would strengthen the paper and propose to include this in later revision.

---

> ### Author Response · Authors · 2024-11-16
> **Author Response to Reviewer oKgq (Part II)**
>
> > **[W5] The paper would benefit from a more in-depth interpretability analysis to clarify the relationship between the generated reasoning, labels, and the knowledge graph.**
> >
> > **[W6] The paper lacks an evaluation of LLM hallucinations, despite the stated aim to address hallucination issues in clinical decision-making.**
>
> We respectfully note that our paper provides substantial analysis of interpretability and hallucination mitigation:
>
> 1. Appendix F provides detailed case studies demonstrating how our model:
>    - Generates structured reasoning chains that explicitly link patient conditions and retrieved knowledge to predictions
>    - Shows how retrieved knowledge from our KG helps avoid hallucinations by grounding predictions in verified medical knowledge
>    - Compares reasoning quality between vanilla LLM predictions (which often hallucinate) and KARE's knowledge-grounded predictions
>
> 2. The effectiveness of our approach in reducing hallucinations is demonstrated through:
>    - Zero/few-shot experiments (Table 2) showing how KARE-augmented context improves prediction accuracy compared to base LLM responses
>    - Our multitask learning strategy (Table 4) which shows better performance compared to the single-task approach
>    - Case studies showing how KARE's predictions are consistently supported by concrete evidence from the KG, unlike baseline LLM approaches that may generate plausible but incorrect reasoning
>
> While we agree that additional analysis could be valuable, we believe our current evaluation demonstrates KARE's ability to generate reliable, knowledge-grounded reasoning for clinical predictions.
>
>
>
> > **[W7] An analysis of model parameters and time consumption is missing, which could provide valuable insights into the model’s computational efficiency and practical applicability**
>
> We agree that analyzing computational requirements is important. KARE has two distinct computational phases:
>
> 1. One-time Preprocessing:
>
> - KG Construction:
>   * From UMLS: 2.8 hours
>   * From LLM: 4.5 hours
>   * From PubMed: 8.3 hours (including 0.4h for concept embedding, 3.1h for retrieval, 4.8h for relation extraction)
>   * Total concept sets processed: 26,134
> - Community Processing:
>   * Leiden community detection (25 runs): 12.4 mins
>   * Summary generation for 147,264 summaries: 9.6 hours
>
> 2. Runtime Components:
>
> - Context augmentation with our dynamic retrieval: ~2s per patient
> - Model inference: ~1s per prediction
> - Hardware: 8 NVIDIA A100 GPUs for fine-tuning (~4.8 hours), single GPU for inference
>
> While preprocessing is computationally intensive, this is a one-time cost common to large-scale KG systems. Given KARE's significant performance improvements in critical healthcare predictions, we believe this computational investment is justified for real-world deployment.
>
>
>
> > **[W8] There is a lack of references to related work.**
>
> Thank you for letting us know these related papers! We have cited them in Lines 135 and 420 in our latest revision.

---

> ### Author Response · Authors · 2024-11-16
> **Author Response to Reviewer oKgq (Part III)**
>
> ---
>
> ### For Your Questions:
>
>
>
> > **[Q1] In Section 3.2, how does the method handle cases where the same patient has more than two visits with different labels?**
>
> We handle this case by treating a patient with $t$ visits as $t-1$ prediction instances, following standard practice in EHR prediction (e.g., PyHealth, GraphCare, RAM-EHR).
>
> Let's illustrate with a concrete example:
>
> **Patient ID: 12345**
>
> ```
> Visit 1: Conditions: c1, c2; Procedures: p1; Medications: m1, m2; readmission_label=0
> Visit 2: Conditions: c3, c4; Procedures: p2, p3; Medications: m3; readmission_label=1
> Visit 3: Conditions: c5; Procedures: p4, p5; Medications: m4, m5; readmission_label=1
> Visit 4: Conditions: c6; Procedures: p6; Medications: m6
> ```
>
> We treat this as three separate prediction instances:
>
> **Patient ID: 12345_1**
>
> - Input:
>   * Visit 1: Conditions: c1, c2; Procedures: p1; Medications: m1, m2
> - Readmission_Label: 0
>
> **Patient ID: 12345_2**
>
> - Input:
>   * Visit 1: Conditions: c1, c2; Procedures: p1; Medications: m1, m2
>   * Visit 2: Conditions: c3, c4; Procedures: p2, p3; Medications: m3
> - Readmission_Label: 1
>
> **Patient ID: 12345_3**
>
> - Input:
>
>   * Visit 1: Conditions: c1, c2; Procedures: p1; Medications: m1, m2
>   * Visit 2: Conditions: c3, c4; Procedures: p2, p3; Medications: m3
>   * Visit 3: Conditions: c5; Procedures: p4, p5; Medications: m4, m5
>
> - Readmission_Label: 1
>
>
>
> The intermediate labels are not used for prediction, as each instance only predicts the outcome of its final visit.
>
>
>
> > **[Q2] In Section 3.2, how is the effectiveness of the relevance score (i.e., Formula (3)) validated? Would an ablation study on the components of Relevance(C_k) help to clarify this?**
>
> Yes, we have conducted such an ablation study, as shown in **Figure 3 (LHS) on Page 10**. This study demonstrates how each component (node hits, coherence, recency, theme relevance, and DGRA) contributes to the model's performance. Node hits proves to be the most critical component, followed by DGRA and theme relevance. We carefully designed and validated this relevance score through extensive experiments.
>
>
>
> > **[Q3] How does the method utilize patient longitudinal visit information?**
>
> Please see our response to **[W2]** above (Part I). Thank you!
>
>
>
> > **[Q4] For the baseline ML methods, it’s mentioned that most are implemented using PyHealth. Could you clarify the backbone model for each ML method? Also, is there a fair configuration in place for implementing language model-based encoders, such as ClinicalBERT, across these methods?**
>
> The ML-based methods in our experiments are all implemented using the original architectures as described in their papers, without incorporating any language models. For fair comparison, we used PyHealth to implement most methods with consistent embedding size (256) across all models. Note that these methods **work directly with structured EHR codes** (conditions, procedures, and medications) rather than processing text information. We detail the baseline implementations in Appendix C.
>
> For GRAM and KerPrint which were not in PyHealth, we implemented them separately following their original codebases while maintaining the same embedding configuration for fairness.
>
> One-sentence summaries of these models can be found on PyHealth's documentation page [R1].
>
>
>
> [R1] https://pyhealth.readthedocs.io/en/latest/#machine-deep-learning-models
>
>
>
>
>
> > **[Q5] The experimental results for mortality prediction show that the baseline ML methods, such as ConCare and TCN, perform closely to KARE, with some evaluation metrics even exceeding KARE’s. How should these results be interpreted?**
>
> We respectively disagree that "*ConCare and TCN perform closely to KARE*", as ConCare achieved 0% and TCN achieved 9.3% on Sensitivity for mortality prediction on MIMIC-III, where sensitivity is the most important metric here to examine model's ability.
>
> As mentioned in our response to **[W3]**, for imbalanced datasets like MIMIC-III-Mortality and MIMIC-IV-Mortality (only 5.42% and 19.16% positive labels), the effectiveness of the model should be measured by its ability to correctly predict "this patient will die in the next visit", which is measured by sensitivity. High accuracy can be misleading - even blindly predicting all patients will survive would achieve 94.6% accuracy on MIMIC-III. Both ConCare and TCN have poor performance in identifying high-risk patients.
>
> As explained in **Lines 461-466**, the trade-off between sensitivity and specificity means that improving sensitivity (identifying mortality risk) can sometimes negatively impact specificity (predicting survival).

---

> ### Author Response · Authors · 2024-11-16
> **Author Response to Reviewer oKgq (Part IV)**
>
> > **[Q6] Interestingly, in ablation study, excluding similar patients often yields better or comparable results than including them, especially for mortality prediction on the MIMIC-III dataset. Do these results support the methodological choices made in Section 3.2?**
>
> As shown in Table 3, similar patient retrieval consistently improves performance across all tasks except MIMIC-III-Mortality. As explained in **Lines 475-480**, this exception occurs because MIMIC-III-Mortality has very few positive samples (5.42%), making it difficult to find truly similar patients for mortality cases since we need to retrieve both positive and negative examples without knowing the target patient's label.
>
> While similar patient retrieval generally shows positive impact, we note that:
>
> 1. It's not our primary innovation (adapted from EHR-CoAgent [R2])
> 2. Its contribution is smaller compared to retrieved knowledge and reasoning chain components
> 3. It provides consistent improvements in most scenarios when positive samples are sufficient (i.e. Readmission tasks in our case)
>
> [R2] Cui, Hejie, et al. "LLMs-based Few-Shot Disease Predictions using EHR: A Novel Approach Combining Predictive Agent Reasoning and Critical Agent Instruction." arxiv 2024.03
>
>
>
>
>
> > **[Q7] Can your model still show substantial improvements compared to other LLM-based models beyond just zero-shot or few-shot settings? I am concerned that most of the current improvements may be attributed primarily to supervised fine-tuning.**
>
> This is an interesting point! As shown in Table 3, when none of similar patients, retrieved knowledge, or reasoning is implemented (row 1), the fine-tuned LLM (Mistral) is not even competitive with ML-based methods. Additionally, in Table 2, we showed that when patient context is augmented with knowledge retrieved by our method, the performance is better than that with Classic RAG in the zero-shot setting.
>
> However, we did not include an ablation study for the fine-tuning model to answer: "Is the knowledge retrieved by our method better than classic RAG in the fine-tuning setting?" Thus, we add the results of such a study on the MIMIC-IV (*with no reasoning and similar patient retrieval applied*) as follows:
>
> |                         | Mortality    |              |                 |                 | Readmission  |              |                 |                 |
> | ----------------------- | ------------ | ------------ | --------------- | --------------- | ------------ | ------------ | --------------- | --------------- |
> | **Retrieved Knowledge** | **Accuracy** | **Macro F1** | **Sensitivity** | **Specificity** | **Accuracy** | **Macro F1** | **Sensitivity** | **Specificity** |
> | None                    | 92.2         | 83.1         | 65.0            | 96.2            | 56.1         | 46.7         | 23.1            | 76.2            |
> | Classic-RAG             | 92.5         | 83.8         | 63.2            | 97.6            | 58.8         | 52.1         | 46.7            | 57.5            |
> | Ours                    | 93.5         | 86.8         | 70.8            | 97.6            | 66.8         | 66.6         | 73.2            | 60.9            |
>
> As expected, the retrieved knowledge by Classic-RAG has lower effectiveness than ours in the fine-tuning setting, which addresses your concern.
>
>
>
>
>
> ---
>
> Again, we greatly appreciate your review and feedback. We have endeavored to address each of your concerns comprehensively. If any aspects require additional clarification or if you have further questions, we would be happy to discuss them.

---

> ### Comment · Reviewer_oKgq · 2024-11-18
>
> Thank you for your detailed response. I will maintain my current rating unless the following questions are adequately addressed.
> > For the response W5 and W6
>
> However, regarding hallucinations, I do not believe Tables 2 and 4 effectively demonstrate KARE's ability to mitigate hallucinations, particularly in the context of generated clinical reasonings. There is a lack of evaluation specifically addressing the quality of clinical reasoning generation and the assessment of reasoning hallucinations. Given that clinical reasoning is one of your key outputs and plays a critical role in clinical medical diagnosis, this aspect warrants more thorough analysis. Similar evaluations have been effectively conducted in other clinical reasoning generation studies.
>
> > For the response W7
>
> I suggest that the authors provide a figure to clearly illustrate the number of **model parameters** and **time consumption** for all baseline models.
>
>
> > For the question Q7
>
> Thank you for conducting additional experiments to address my concerns. However, I believe the authors should include some fine-tuned LLMs individually, such as Mistral or LLaMA2, as well as their combination with RAG, in Table 2. The current results in Table 2 do not appear entirely fair, as the LLM-based methods are evaluated solely in zero-shot or few-shot settings without fine-tuning.

---

> ### Author Response · Authors · 2024-11-20
> **New results & discussions based on your feedback.**
>
> Dear Reviewer oKgq,
>
> Thank you very much for your prompt and insightful feedback! Based on your comments, we have made the following updates and enhancements to our work:
>
> 1. Conducted a human evaluation of the reasoning chains generated by KARE.
> 2. Performed a comprehensive evaluation of model parameters and time consumption for all baseline models.
> 3. Added new results for fine-tuned LLMs in Table 2.
>
>
>
> Details of these updates are provided below:
>
> > **For [W5] and [W6]:**
>
> We hired *three MD students and one MD professional* to conduct a human evaluation of 100 reasoning chains generated by KARE for both correct and incorrect mortality/readmission predictions. All the evaluated samples are from test sets.
>
> The evaluation details have been included in **Appendix I of our latest revision**, with the results presented in **Fig. 18 on Page 45**. We have also provided detailed discussions under the figure, which can be summarized as follows:
>
> 1. The quality of the reasoning chains is crucial to the accuracy of the final prediction.
> 2. Both tasks (mortality and readmission prediction) are inherently challenging, even for experienced clinicians, due to limited patient information (e.g., missing age/gender data and coarse-grained medical concepts). Despite these challenges, KARE demonstrates superior performance compared to clinicians in information-scarce scenarios.
> 3. Some inconsistencies between the reasoning chains and the final predictions were observed in a small number of cases. Addressing this issue can potentially further improve the performance of KARE.
>
>
> To ensure the transparency, we anonymously share the reviewed samples and raw results at:
> https://drive.google.com/drive/folders/1h9qh9ZfO7LK3VGqoVSFLFaHUu6TrzqAB?usp=sharing
>
>
> > **For [W7]:**
>
> We have reviewed the parameters and training time consumption for all baseline models. The results are presented in **Fig. 19 and Fig. 20 in Appendix J (Page 46).**
>
> The findings indicate that while KARE has a larger parameter size, it achieves significantly superior performance compared to other models, particularly excelling over Mistral-7B with Classic RAG in both efficiency and predictive capability.
>
> We would also like to emphasize that performance and interpretability are the most critical aspects of clinical predictive models. While lightweight machine learning-based models are parameter-efficient, their performance is consistently suboptimal and lacks interpretability without mechanisms such as reasoning chains.
>
>
>
> > **For [Q7]**
>
> We understand your concern and have included the performance of the fine-tuned backbone model, Mistral-7B-Instruct-v0.3, as well as its integration with Classic RAG in **Table 2**.
>
>
>
> ---
>
> We hope these updates address your concerns and further clarify the strengths of our approach. Thank you again for your valuable feedback, and please feel free to let us know if you have any additional suggestions.

---

> > ### Comment · Reviewer_oKgq · 2024-11-21
> >
> > Thank you for your detailed response. My concerns have been thoroughly addressed, and I have updated my rating accordingly.

---

> > > ### Author Response · Authors · 2024-11-21
> > > **Thank you!**
> > >
> > > We're delighted that our responses addressed your concerns and are truly grateful for the increased rating! Thank you for your thoughtful feedback which helped strengthen our work significantly.

---

### Official Review · Reviewer_thia · 2024-10-31

**Soundness:** 3
**Presentation:** 3
**Contribution:** 2
**Rating:** 3
**Confidence:** 4

**Summary:**

The paper introduces KARE, a novel framework designed to enhance clinical decision support by addressing the limitations of Large Language Models (LLMs) in healthcare. While LLMs show potential, they suffer from hallucinations and lack the fine-grained medical knowledge necessary for high-stakes applications like diagnosis. Traditional retrieval-augmented generation (RAG) methods often retrieve sparse or irrelevant data, undermining accuracy. KARE improves upon this by integrating a multi-source knowledge graph (KG) with LLM reasoning. The KG is built from biomedical databases, clinical literature, and LLM-generated insights, structured using hierarchical community detection for precise information retrieval. Key innovations include dense medical knowledge structuring, dynamic retrieval of multi-faceted medical insights, and reasoning-enhanced predictions. KARE outperforms existing models in MIMIC-III and MIMIC-IV datasets, improving prediction accuracy by up to 15%, while also enhancing the interpretability and trustworthiness of clinical predictions.

**Strengths:**

1. The research team has achieved impressive results, exceeding established benchmarks. This progress marks a significant step forward in improving predictive model accuracy in medical data analysis.
2. The authors' workload is immense, and the experimental details are thoroughly outlined, which I greatly appreciate.

**Weaknesses:**

1. By reviewing your code and the details in the article, I can see that your workload is immense, however, the contribution of this article is incremental. My understanding is that it is essentially a combination of GraphRAG and GraphCare [1]. Furthermore, many key baselines were not cited. Since the authors mentioned that this paper focuses on RAG for EHR, some essential RAG algorithms should have been introduced, such as MedRetriever [2], and commonly used GraphRAG algorithms like KGRAG [3].
2. In the experiment or appendix section, I did not clearly see the formulas for Sensitivity and Specificity, nor were there any corresponding references, which is quite confusing to me. Moreover, using Accuracy as a metric in cases of highly imbalanced labels is unreasonable. For instance, in the MIMIC-III Mortality Prediction task, the positive rate is 5.42%. If I predict that all patients will survive, I can still achieve an accuracy of 94.58%. Previous works, such as GraphCare [1], have adopted AUROC and AUPRC as evaluation metrics.
3. The article is overly long and filled with detailed content, making it easy for readers to miss important points.

- [1] GraphCare: Enhancing Healthcare Predictions with Personalized Knowledge Graphs. ICLR 2024
- [2] MedRetriever: Target-driven interpretable health risk prediction via retrieving unstructured medical text. CIKM 2021
- [3] Biomedical knowledge graph-enhanced prompt generation for large language models. Arxiv 2023

**Questions:**

1. The authors used Claude 3.5 Sonnet as an expert model to generate training samples and augment knowledge graph. However, since Claude is a general-purpose model, could it lack some medical knowledge, potentially leading to biased training samples and cumulative errors? As mentioned in your summary: "Yet LLMs still suffer from hallucinations and lack fine-grained contextual medical knowledge, limiting their high-stakes healthcare applications such as clinical diagnosis." In the experiment section, there are many related LLMs in the medical domain. It would be better if the researcher could compare KARE with more related LLM-based baselines referred in [4].
2. I didn't see any examples of training samples in the code you provided. Can you provide us with some examples?
3. This parameter design is very challenging. In KARE, there are many hyperparameters, including but not limited to those in graph generation, summarization, model training, and model testing. Any slight change can lead to significant deviations in the model's results. Could you elaborate on how you adjust the parameters in such a large hyper-parameter space?


- [4] https://huggingface.co/blog/leaderboard-medicalllm.

**Details Of Ethics Concerns:**

No ethics concerns. To my knowledge, EHR data is fuzzified, for example, the patient's visit time information is also time offset. And the author also used local LLM, so I don't think there are any ethical concerns. The author has already provided details in Appendix A.

---

> ### Author Response · Authors · 2024-11-16
> **Author Response to Reviewer thia (Part I)**
>
> **Author Response to Reviewer thia**
>
> Thank you for recognizing the strengths of our work. We address your concerns and answer your questions below. We also uploaded a revision and used blue to mark the new changes.
>
> ---
>
> ### For Weaknesses:
>
> > **[W1.1] The contribution is incremental. It is a combination of GraphRAG and GraphCare.**
>
> We respectfully disagree with this characterization. KARE represents a novel framework that goes well beyond combining existing approaches. We address this from both technical and research impact perspectives:
>
> **Technical Contribution:**
>
> - The overlap of KARE and GraphRAG exists only in ***KG Partitioning using Leiden*** (first half of Section 3.1.3). All other components are distinct:
>
> |                                   | KARE (ours)                                                  | GraphRAG (Edge et al., 2024)                                 | Key Advantages of KARE                                       |
> | --------------------------------- | ------------------------------------------------------------ | ------------------------------------------------------------ | ------------------------------------------------------------ |
> | **KG Construction**    | 1. Sources: biomedical KG, corpus, and LLMs; 2. Knowledge extraction based on the medical concept co-existence in patient visits across EHR dataset | Only sourced from documents, without task-specific prior information, leading to unfocused structured knowledge | The constructed KG contains knowledge highly relevant to EHR prediction, as clinical knowledge can be found easily in sources where multiple concepts co-exist |
> | **KG Partitioning**               | Multiple runs (25 in our case)                               | Single run                                                   | A node can belong to multiple communities at the same hierarchical level, while in GraphRAG it exists in only one community. This is **very important** as medical concepts often co-exist with different sets of concepts in patient visits |
> | **Community Summarization** | Multiple theme-specific summaries for each community (themes: general/mortality/readmission in our case) | General summaries for communities                            | Communities can be interpreted differently for different tasks, enhancing effectiveness across multiple prediction tasks |
> | **Community Retrieval**           | Dynamic retrieval with: 1. Node hits tracking; 2. Decay factors for previously retrieved information; 3. Context coherence; 4. Temporal recency; 5. Theme relevance; 6. Iterative selection (Algorithm 1) | Parallel processing of community chunks with helpfulness scoring | 1. Dynamically avoids redundant information retrieval through hit tracking and decay;  2. Healthcare-specific metrics ensure clinical relevance |
>
> - The overlap with GraphCare exists only in ***Patient KG Construction (Equation 2)***, where KARE uses the patient KG solely as a reference (to compute node hits and recency) for information retrieval:
>
> |                                     | KARE (ours)                                                  | GraphCare (Jiang et al., 2024)                               | Key Advantages of KARE                                                 |
> | ----------------------------------- | ------------------------------------------------------------ | ------------------------------------------------------------ | ------------------------------------------------------------ |
> | **KG Construction**      | 1.Sources: biomedical KG, corpus, and LLMs; 2. Knowledge extraction guided by medical concept co-existence in patient visits across EHR dataset | Sourced from LLMs and KGs, without prior EHR dataset information, leading to inclusion of task-irrelevant knowledge | More focused medical knowledge due to concept co-existence guidance in extraction |
> | **Input Feature / Patient Context** | The most important context (community summaries) referred to the patient KG. | Entire patient KG, containing sparse and random medical knowledge | Input features focus on essential information captured by graph communities (real-world associated knowledge) |
>
> In conclusion, while KARE shares some basic concepts with GraphRAG and GraphCare, it contributes significant task-specific innovations for EHR prediction. A simple combination of these methods would perform poorly (worse than most traditional ML methods) on these tasks.

---

> ### Author Response · Authors · 2024-11-16
> **Author Response to Reviewer thia (Part II)**
>
> **Research Impact on Clinical Prediction:**
>
> Our work addresses a critical gap in current clinical prediction research. Recent studies [R1, R2] have concluded that LLMs perform poorly in clinical prediction tasks, even after fine-tuning. Their findings are consistent with our results in Table 3, where row 1 shows that fine-tuned LLM without retrieval and reasoning performs worse than traditional ML methods (Table 2).
>
> However, our work demonstrates that this limitation can be overcome. By incorporating knowledge retrieval and reasoning in the fine-tuning process, we show significant improvements in LLM performance for clinical prediction tasks. This suggests a promising direction for leveraging LLMs in healthcare applications when properly augmented with medical knowledge and reasoning capabilities.
>
>
>
> In summary, KARE represents a fundamental advance in clinical prediction, not an incremental combination of existing methods. Its novel technical components and strong empirical results demonstrate how to effectively harness LLMs for healthcare applications.
>
>
>
> [R1] Chen et al. "ClinicalBench: Can LLMs Beat Traditional ML Models in Clinical Prediction?", arxiv 2024.11
>
> [R2] Liu et al. "Large Language Models Are Poor Clinical Decision-Makers: A Comprehensive Benchmark" EMNLP 2024
>
>
>
> > **[W1.2] Some RAG algorithms like MedRetriever and KGRAG should be introduced.**
>
> We have added citations for these two papers in the related work section (highlighted in blue). Additionally, we have tested MedRetriever's performance and added its results to Table 2 as follows:
>
> | Dataset & Task        | Accuracy | Macro F1 | Sensitivity | Specificity |
> | --------------------- | -------- | -------- | ----------- | ----------- |
> | MIMIC-III-Mortality   | 93.2     | 53.3     | 11.3        | 95.2        |
> | MIMIC-III-Readmission | 63.2     | 62.7     | 66.3        | 59.1        |
> | MIMIC-IV-Mortality    | 89.5     | 77.9     | 55.6        | 95.2        |
> | MIMIC-IV-Readmission  | 63.0     | 62.1     | 69.4        | 55.8        |
>
>
>
> > **[W2.1] Lacking formulas for Sensitivity and Specificity.**
>
> We apologize for not explicitly including the formulas. Sensitivity and Specificity are standard metrics for evaluating ML-based classification problems:
>
> - Sensitivity = TP/(TP + FN)  [True Positive Rate]
>
> - Specificity = TN/(TN + FP)  [True Negative Rate]
>
> where TP = True Positives, TN = True Negatives, FP = False Positives, and FN = False Negatives.
>
> These metrics are particularly crucial in our healthcare setting. **Sensitivity measures the model's ability to correctly identify high-risk patients** (e.g., those who will die or be readmitted), while **Specificity measures its ability to correctly identify low-risk patients**. In our paper (lines 461-466), we highlighted the importance of the sensitivity, and explained why specificity of KARE is not always the best.
>
> The overfitting case you mentioned (*"in the MIMIC-III Mortality Prediction task, the positive rate is 5.42%. If I predict that all patients will survive, I can still achieve an accuracy of 94.58%"*) can be observed in ConCare's performance on this task, where it failed to learn the ability to predict the patients who will die, as shown by its Sensitivity of 0.
>
>
>
> > **[W2.2] Should adopt metrics like AUROC and AUPRC for imbalanced labels.**
>
> **AUROC and AUPRC cannot be directly measured for LLM predictions** because, although LLMs compute next-token probabilities internally, these probabilities are: (1) distributed over the entire vocabulary rather than just binary classes, (2) dependent on how different LLMs encode the same label ("0"/"1") using different tokens or combinations, and (3) not directly comparable to the binary class probabilities output by ML models.
>
> Therefore, **we use sensitivity and specificity which effectively evaluate performance on imbalanced datasets** using only the final predictions.
>
> On the highly imbalanced MIMIC-III/IV mortality task (positive rate = 5.42%/19.16%), KARE achieves significantly higher sensitivity (24.7%/73.2%) compared to baselines while maintaining high specificity (98.3%/99.8%). This demonstrates our model's superior ability to identify high-risk patients - the most critical capability for mortality prediction.
>
> Our metric choice (accuracy, macro F1, sensitivity, specificity) aligns with other recent LLM-based EHR prediction works like EHR-CoAgent [R3].
>
> ***We have included the discussion of metrics in Appendix E in the latest revision.***
>
> [R3] Cui, Hejie, et al. "LLMs-based Few-Shot Disease Predictions using EHR: A Novel Approach Combining Predictive Agent Reasoning and Critical Agent Instruction." arxiv 2024.03

---

> ### Author Response · Authors · 2024-11-16
> **Author Response to Reviewer thia (Part III)**
>
> ---
>
> ### For Your Questions:
>
> > **[Q1] It would be better if the researcher could compare KARE with more (medical) related LLM-based baselines in [4].**
>
> Thanks for raising this important concern. We address this from two perspectives:
>
> 1. **Choice of Claude 3.5 Sonnet:** Recent evaluations from the same research group show that: (1) Figure 1 in [R4] demonstrates Sonnet 3.5 has similar capability as GPT-4 on medical tasks, and (2) Figure 5 in [R5] shows GPT-4 outperforms specialized medical LLMs like Meditron3-70b and OpenBioLLM-70b. This suggests Sonnet 3.5 has equivalent or superior capabilities compared to medical-specific LLMs for the tasks.
>
> 2. **Mitigation of Hallucination Risk:** Our framework specifically addresses the hallucination concern through:
>    - Multi-source knowledge verification (biomedical KG, literature, LLM)
>    - Community-based knowledge organization that preserves verified relationships
>    - Dynamic retrieval that prioritizes verified medical knowledge
>    - We hired medical experts to evaluate the KARE-generated reasoning chains toward the prediction. This new study is detailed in  **Appendix I of our latest revision**, with the results presented in **Fig. 18 on Page 45**. The result shows a high consistency between KARE's reasoning with experts'.
>
> While comparing with additional medical LLMs would be valuable, reproducing our entire pipeline with different base LLMs during the rebuttal period would be impractical due to computational constraints and time limitations.
>
> [R4] https://huggingface.co/blog/mpimentel/comparing-llms-medical-ai
>
> [R5] Kanithi, Praveen K., et al. "Medic: Towards a comprehensive framework for evaluating llms in clinical applications." arxiv 2024.09
>
>
>
> > **[Q2] I didn't see any examples of training samples in the code you provided. Can you provide us with some examples?**
>
> Sure, we shared some examples of the training data in this anonymous folder: https://drive.google.com/drive/folders/18bWak-xCmLh7oTtSCqg9MnWk6A2Tj8gQ?usp=drive_link
>
>
>
> >**[Q3] This parameter design is very challenging. Could you elaborate on how you adjust the parameters in such a large hyper-parameter space?**
>
> Almost all hyperparameters in our framework can be determined through empirical observation rather than exhaustive grid search. For example:
>
> - Community size thresholds are determined by observing the LLM's context window size constraints
> - Clustering thresholds are optimized using silhouette scores, with the optimality verified through inspection of the resulting cluster qualities
> - Context augmentation parameters are selected by generating several sets of augmented context and leveraging the LLM's ability to evaluate the relevance and utility of the retrieved information
> - For LLM fine-tuning, we use standard hyperparameters recommended for instruction fine-tuning of Mistral-7B models, with a cosine learning rate schedule implemented through the TRL package.
>
> Therefore, despite the seemingly large parameter space, the tuning process remains manageable as most parameters can be determined through principled observation and validation rather than exhaustive search.
>
> ---
>
> Again, we greatly appreciate your review and feedback. We have endeavored to address each of your concerns comprehensively. If any aspects require additional clarification or if you have further questions, we would be happy to discuss them.

---

> > ### Comment · Reviewer_thia · 2024-11-25
> >
> > Dear Authors,
> >
> > Thank you for your hard work on this submission and your detailed rebuttal. While I appreciate your efforts, based on my experience in this domain, I find that KARE does not offer significant novelty or contributions compared to existing approaches like GraphCare and GraphRAG.
> >
> > Regarding the introduction of AUROC and AUPRC, it is feasible to implement by calculating the probabilities of "die" or "live" tokens from the model output logits and then applying softmax, or averaging the results over multiple runs to obtain stable probabilities.
> >
> > Moreover, your explanation for hyperparameter tuning remains unconvincing to me.
> >
> > Given these considerations, I have decided to keep my rating unchanged.
> >
> > Best regards,
> > Reviewer

---

> > > ### Author Response · Authors · 2024-11-26
> > > **Kindly Seeking Your Further Feedback**
> > >
> > > Dear Reviewer thia,
> > >
> > > We write to follow up on our new response addressing your concerns in short:
> > >
> > > 1. KARE is novel mainly because: (1) both GraphRAG and GraphCare lack reasoning capabilities, targeting different tasks; (2) GraphCare's KG contains random and irrelevant relationships, while KARE builds the KG according to the co-existence patterns of medical concepts; (3) the only overlap between KARE and GraphRAG is using Leiden for graph partitioning during KG construction (Step 1); (4) KARE addresses a critical gap in the research community, and we demonstrate that LLMs are not poor clinical predictors - **they just need highly relevant knowledge and unified-format rationale**, which is provided by our framework.
> > >
> > > 2. Your proposed AUROC computation method for LLMs faces fundamental technical challenges: (1) A recent work, ClinicalBench, uses the same approach you recommended, but their results (e.g., Table 1) show AUROC values that correlate well with F1 scores for traditional ML methods but exhibit random behavior for LLM-based approaches, demonstrating the unreliability of such computation; (2) until a theoretically well-justified calibration approach is developed, AUROC for LLMs on binary classification tasks cannot be accurately computed.
> > >
> > > 3. Regarding hyperparameter tuning, we have shown that: (1) KG construction parameters are determined by computational constraints; (2) knowledge retrieval parameters are validated through efficient LLM-based utility assessment; (3) LLM training involves only two main tunable parameters, making the optimization process manageable.
> > >
> > > As the discussion period has been extended, we welcome any additional questions or suggestions for experiments that could help address remaining concerns. We are committed to ensuring the rigor and clarity of our work.
> > >
> > >
> > >
> > > Best regards,
> > >
> > > The Authors

---

> ### Author Response · Authors · 2024-11-24
> **Kindly Seeking Your Feedback on Our Response**
>
> Dear Reviewer thia,
>
> Thank you again for your detailed review! As the reviewer-author discussion phase concludes shortly, we wanted to ensure you've seen our comprehensive response above addressing your main concerns:
>
> - **Regarding the perceived incremental contribution**: We showcased that KARE is fundamentally different from GraphRAG/GraphCare through detailed comparison tables highlighting the minimal technical overlap. Recent studies concluded that LLMs perform poorly in clinical prediction tasks, even after fine-tuning. Our work demonstrates how to overcome this limitation through knowledge retrieval and reasoning in the fine-tuning process - representing a fundamental advance rather than an incremental combination.
>
> - **On metrics and evaluation**: We've clarified our metric choices (why not AUROC/AUPRC) and formulas in ***Appendix E***. The sensitivity/specificity metrics are particularly crucial for imbalanced healthcare datasets, where KARE shows significant improvements in identifying high-risk patients. We have also added MedRetriever's performance to Table 2.
>
> - **Concerning LLM reliability**: We've added a new human evaluation study in ***Appendix I***, where medical professionals assessed 100 reasoning chains generated by KARE. The results demonstrate KARE's effectiveness in generating reliable clinical reasoning.
>
> - **For training samples and hyperparameter tuning**: We've shared example training data in an anonymous folder and explained our principled approach to parameter selection.
>
> We believe these updates substantially strengthen our work. As the discussion phase is ending soon, we would greatly appreciate your feedback on whether any concerns remain unaddressed. We are happy to provide further clarifications or improvements if needed.
>
> Best regards,
>
> The Authors

---

> ### Author Response · Authors · 2024-11-25
> **Round2 Response to Reviewer thia (Part I)**
>
> Dear Reviewer thia,
>
>
>
> Thank you for your continuous review of our work and response. We would like to provide further clarifications based on your remaining concerns:
>
>
>
> > **(1)  I find that KARE does not offer significant novelty or contributions compared to existing approaches like GraphCare and GraphRAG**
>
>
>
> We respectively disagree with the assessment of KARE's novelty for several fundamental reasons:
>
> (1) GraphCare and GraphRAG lack reasoning processes, with GraphRAG being designed for dataset-level summarization rather than clinical predictions. The reasoning in KARE provides crucial interpretability for clinical decision.
>
> (2) GraphCare's knowledge graph ignores the co-existence patterns of medical concepts in EHR data. Integrating EHR-irrelevant knowledge is very likely unhelpful or even harmful to prediction performance. KARE's approach ensures captured relationships are clinically meaningful, **and largely mitigates the needs of validations from medical experts**. Also, patient graph in KARE is not the input feature, but just a reference for knowledge retrieval. Moreover, GraphCare is not an LLM-based method.
>
> (3) GraphRAG's original implementation fundamentally **does not work** for clinical prediction tasks. This is clearly demonstrated by our experiments: Figure 3 shows the effectiveness of our DGRA algorithm, and Table 4 validates our multitask setting - contributions that have been possibly overlooked in your assessment.
>
> (4) Importantly, while recent works [R1, R2] conclude that LLMs are poor clinical decision-makers, KARE provides an effective approach to significantly boost LLM performance, with detailed component analysis in Table 3. This addresses a critical gap in the research community.
>
> We believe that if using "existing techniques" negates novelty, groundbreaking works like BERT would have been rejected for using "just another transformer encoder." In our work, the only technique we adopted from GraphRAG is **Graph Partitioning using Leiden** (which we also found to be effective for clinical prediction when applied multiple times to ensure diversity). Therefore, we strongly disagree with characterizing KARE as incremental to any existing work.
>
>
>
>
> > **(2) Regarding the introduction of AUROC and AUPRC, it is feasible to implement by calculating the probabilities of "die" or "live" tokens from the model output logits and then applying softmax, or averaging the results over multiple runs to obtain stable probabilities.**
>
> The suggested approach of "calculating probabilities of 'die' or 'live' tokens from model output logits" is technically infeasible for several fundamental reasons:
>
> Unlike traditional ML models with two output neurons for binary classification, LLMs:
>
> - Have a vocabulary of 50K+ tokens where concepts like "death" can be expressed through numerous tokens ("die", "died", "deceased", "passing", etc.)
> - Distribute probabilities across the entire vocabulary
> - Have no clear mapping between token probabilities and binary class probabilities
>
> **Even if we tried to aggregate probabilities for related tokens**:
>
> - There's no principled way to identify all relevant tokens for each class
> - Token probabilities sum to 1.0 across ALL possible next tokens, not just those relevant to classification
> - Real LLM examples demonstrate why normalization is problematic: when a model outputs probabilities like P("die")=0.15 and P("live")=0.13, even if the prediction is clearly "die", normalizing these probabilities would artificially make them appear similarly likely (≈0.54 vs 0.46)
> - This forced normalization completely distorts the model's actual prediction confidence and makes the resulting probabilities incomparable to ML model probabilities where class probabilities naturally sum to 1.0
>
> Given these fundamental issues, **AUROC/AUPRC cannot be accurately computed for LLMs on binary classification tasks until a theoretically well-justified calibration approach is developed**. Using these metrics without proper theoretical foundations would result in **unfair and potentially misleading comparisons** between LLMs and traditional ML models.
>
> **ClinicalBench's [R1] Table 1 is an evidence** for this: while the AUROC varies consistently with F1 for traditional ML methods, it's quite random for LLM-based methods, showing the unreliability of such computation method.
>
> These limitations explain why recent LLM-based clinical prediction works (e.g., EHR-CoAgent) rely on final predictions rather than  deriving unreliable probability scores. **In our work, we use the same metrics as EHR-CoAgent** [R3].
>
> [R1] Chen et al. "ClinicalBench: Can LLMs Beat Traditional ML Models in Clinical Prediction?", arxiv 2024.11
>
> [R2] Liu et al. "Large Language Models Are Poor Clinical Decision-Makers: A Comprehensive Benchmark" EMNLP 2024
>
> [R3] Cui, Hejie, et al. "LLMs-based Few-Shot Disease Predictions using EHR: A Novel Approach Combining Predictive Agent Reasoning and Critical Agent Instruction." arxiv 2024.03

---

> > ### Author Response · Authors · 2024-11-25
> > **Round2 Response to Reviewer thia (Part II)**
> >
> > > **(3) Your explanation for hyperparameter tuning remains unconvincing to me.**
> >
> > Your concern about hyperparameter sensitivity appears to be based on an incorrect assumption. Our experiments demonstrate robust performance - Figure 3 shows that even removing entire knowledge sources like UMLS has minimal impact on final performance, indicating our method is not highly sensitive to parameter changes.
> >
> > We should clarify that we do not claim our hyperparameter settings are optimal. Many choices in KG construction were practically constrained by computational resources. For example, we process 1/10 of PubMed abstracts (~3M out of 30M) simply because dense retrieval from the full corpus would be computationally prohibitive.
> >
> > For context augmentation (knowledge retrieval), the parameters are tuned within the range [0, 1], with several samples evaluated by the LLM for retrieval utility under each setting. This process can be efficiently completed in a short amount of time.
> >
> > For model training, our hyperparameter space is actually quite limited. The configuration file (https://anonymous.4open.science/r/KARE-Anonymous/finetune/recipes/config_full_mortality.yaml) shows only two main tunable parameters:
> >
> > - learning_rate: tuned between 1e-7 and 1e-4
> > - gradient_accumulation_steps: tuned from 1 to 8
> >
> > Other parameters like per_device_train_batch_size are fixed at 1 due to GPU memory constraints.
> >
> > This limited parameter space, combined with our demonstrated robustness to major component changes, suggests that hyperparameter tuning is not a significant concern for reproducing our results.
> >
> >
> >
> > ---
> >
> > We appreciate your continued engagement with our work and hope these clarifications address your concerns. ***Please let us know if you want us to conduct any additional experiments or provide further clarifications.***
> >
> >
> >
> > Sincerely,
> >
> > The Authors

---

> ### Author Response · Authors · 2024-12-01
>
> Dear Reviewer thia,
>
> Thank you for your thoughtful initial review. We've provided detailed responses to your new comments in our [Round 2 Response](https://openreview.net/forum?id=8fLgt7PQza&noteId=8AHRqsPG1J). Would you be able to review our responses and share your assessment? As the discussion period closes in **two days**, we are still available to address any remaining concerns you may have.
>
> Best regards,
> \
> The Authors

---

### Official Review · Reviewer_pYH3 · 2024-11-02

**Soundness:** 4
**Presentation:** 3
**Contribution:** 2
**Rating:** 6
**Confidence:** 3

**Summary:**

This paper introduces KARE, a framework integrating LLM reasoning with KG retrieval to improve healthcare predictions. KARE combines structured multi-source medical knowledge with dynamic, patient-specific context augmentation to provide accurate and interpretable clinical predictions. Evaluated on MIMIC-III and MIMIC-IV datasets for mortality and readmission predictions, KARE demonstrates improved accuracy and interpretability over conventional models.

**Strengths:**

- Pros
  - Interesting topic for enhancing healthcare predictions by combining the reasoning capabilities of LLM.
  - Clear and well-motivated reasoning in the paper.
  - Comprehensive ablation studies validate the contributions of each model component.
  - Well-written and structured.

**Weaknesses:**

- Cons
  - The novelty of this paper is relatively limited, appearing to be incremental compared to GraphRAG [1].
  - In Section 3.3.1, the authors select the reasoning chain with the highest confidence as training data. However, according to conclusions from some existing studies [2,3], the reasoning chain with the highest confidence is not necessarily the most reliable.
  - MedRetriever [4] also adopts a retrieval-augmented approach for healthcare prediction, but this paper lacks a comparative analysis with MedRetriever.
  - The definitions and calculation methods for Sensitivity and Specificity need to be clarified more thoroughly, and metrics such as AUROC and AUPRC should be added.
  - Although Amazon Bedrock provides strict compliance standards and privacy protection measures, relying on it to generate reasoning chains for distillation may limit the generalizability of this approach in real healthcare scenarios with high privacy protection requirements.

[1] Edge D, Trinh H, Cheng N, Bradley J, Chao A, Mody A, Truitt S, Larson J. From local to global: A graph rag approach to query-focused summarization. arXiv preprint arXiv:2404.16130. 2024 Apr 24.

[2] Yang, Haoyan, et al. "Can We Trust LLMs? Mitigate Overconfidence Bias in LLMs through Knowledge Transfer." arXiv preprint arXiv:2405.16856 (2024).

[3] Xiong, M., Hu, Z., Lu, X., LI, Y., Fu, J., He, J. and Hooi, B., Can LLMs Express Their Uncertainty? An Empirical Evaluation of Confidence Elicitation in LLMs. In The Twelfth International Conference on Learning Representations.

[4] Ye M, Cui S, Wang Y, Luo J, Xiao C, Ma F. Medretriever: Target-driven interpretable health risk prediction via retrieving unstructured medical text. InProceedings of the 30th ACM International Conference on Information & Knowledge Management 2021 Oct 26 (pp. 2414-2423).

**Questions:**

See Weaknesses Above.

---

> ### Author Response · Authors · 2024-11-16
> **Author Response to Reviewer pYH3 (Part I)**
>
> **Author Response to Reviewer pYH3**
>
> Thank you for recognizing the strengths of our work. We address your concerns and answer your questions below. We also uploaded a revision and used blue to mark the new changes.
>
> ---
>
> > **[W1] The novelty of this paper is relatively limited, appearing to be incremental compared to GraphRAG.**
>
> KARE and GraphRAG address fundamentally different problems: KARE focuses on clinical prediction with reasoning, while GraphRAG tackles query-focused document summarization. This difference in goals drives substantially different technical innovations tailored to each domain's unique challenges. While both methods use the Leiden algorithm for community detection, their technical approaches differ significantly:
>
> |                                   | KARE (ours)                                                  | GraphRAG (Edge et al., 2024)                                 | Key Advantages of KARE                                       |
> | --------------------------------- | ------------------------------------------------------------ | ------------------------------------------------------------ | ------------------------------------------------------------ |
> | **KG Construction**    | 1. Sources: biomedical KG, corpus, and LLMs; 2. Knowledge extraction based on the medical concept co-existence in patient visits across EHR dataset | Only sourced from documents, without task-specific prior information, leading to unfocused structured knowledge | 1. Domain-specific KG construction integrating multiple medical knowledge sources; 2. Knowledge organization driven by real-world clinical patterns; 3. Better handling of medical terminology variations |
> | **KG Partitioning**               | Multiple runs (25 in our case) with different random seeds to capture diverse concept relationships | Single run of community detection                            | 1. A medical concept can belong to multiple communities at the same level, reflecting its different clinical contexts; 2. More robust representation of complex medical relationships |
> | **Community Summarization** | Multiple theme-specific summaries for each community (themes: general/mortality/readmission in our case) | General summaries for communities                            | Communities can be interpreted differently for different tasks, enhancing effectiveness across multiple prediction tasks |
> | **Community Retrieval**           | Dynamic retrieval with: 1. Node hits tracking; 2. Decay factors for previously retrieved information; 3. Context coherence; 4. Temporal recency; 5. Theme relevance; 6. Iterative selection (Algorithm 1) | Parallel processing of community chunks with helpfulness scoring for query-focused document summarization | 1. Dynamically avoids redundant information retrieval through hit tracking and decay;  2. Healthcare-specific metrics ensure clinical relevance |
>
> The key novelty of KARE lies in its integration of domain knowledge, clinical reasoning, and prediction capabilities to address the specific challenges of healthcare applications. Our extensive experiments demonstrate that these healthcare-specific innovations lead to substantial improvements over existing methods in clinical prediction tasks.

---

> ### Author Response · Authors · 2024-11-16
> **Author Response to Reviewer pYH3 (Part II)**
>
> > **[W2] In Section 3.3.1, the authors select the reasoning chain with the highest confidence as training data. However, according to conclusions from some existing studies [2,3], the reasoning chain with the highest confidence is not necessarily the most reliable.**
>
> Thank you for sharing those two interesting papers! We also found another paper [R1] showing similar findings. While our experimental results demonstrate meaningful performance gains through confidence-based reasoning chain selection (see table below), we acknowledge that relying solely on verbalized confidence for chain selection may not be optimal.
>
> | Reasoning Chain Selection (within Three Runs)? | Task (on MIMIC-IV) | Accuracy | Macro F1 | Sensitivity | Specificity |
> | ---------------------------------------------- | ------------------ | -------- | -------- | ----------- | ----------- |
> | N                                              | Mortality          | 93.7     | 89.5     | 71.8        | 99.5        |
> | Y                                              | Mortality          | 94.1     | 90.4     | 73.2        | 99.8        |
> | N                                              | Readmission        | 73.0     | 72.6     | 80.5        | 65.8        |
> | Y                                              | Readmission        | 73.9     | 73.8     | 85.6        | 63.7        |
>
> In future work, we plan to explore more robust approaches for reasoning chain selection, including uncertainty quantification methods proposed in [R2] and other recent works. This could further enhance our framework's performance while addressing the current limitation of relying on verbalized confidence.
>
> [R1] Tanneru, Sree Harsha, Chirag Agarwal, and Himabindu Lakkaraju. "Quantifying uncertainty in natural language explanations of large language models." in PMLR.
>
> [R2] Lin, Zhen, Shubhendu Trivedi, and Jimeng Sun. "Generating with confidence: Uncertainty quantification for black-box large language models." in TMLR
>
>
>
> > **[W3] MedRetriever [4] also adopts a retrieval-augmented approach for healthcare prediction, but this paper lacks a comparative analysis with MedRetriever.**
>
> We have added citations for MedRetriever in the related work section (highlighted in blue). Additionally, we have tested MedRetriever's performance and added its results to Table 2 as follows:
>
> | Dataset & Task        | Accuracy | Macro F1 | Sensitivity | Specificity |
> | --------------------- | -------- | -------- | ----------- | ----------- |
> | MIMIC-III-Mortality   | 93.2     | 53.3     | 11.3        | 95.2        |
> | MIMIC-III-Readmission | 63.2     | 62.7     | 66.3        | 59.1        |
> | MIMIC-IV-Mortality    | 89.5     | 77.9     | 55.6        | 95.2        |
> | MIMIC-IV-Readmission  | 63.0     | 62.1     | 69.4        | 55.8        |

---

> ### Author Response · Authors · 2024-11-16
> **Author Response to Reviewer pYH3 (Part III)**
>
> > **[W4.1] The definitions and calculation methods for Sensitivity and Specificity need to be clarified more thoroughly**
>
> We apologize for not explicitly including the calculation methods. Sensitivity and Specificity are standard metrics for evaluating ML-based classification problems:
>
> - Sensitivity = TP/(TP + FN)  [True Positive Rate]
>
> - Specificity = TN/(TN + FP)  [True Negative Rate]
>
> where TP = True Positives, TN = True Negatives, FP = False Positives, and FN = False Negatives.
>
> These metrics are particularly crucial in our healthcare setting. **Sensitivity measures the model's ability to correctly identify high-risk patients** (e.g., those who will die or be readmitted), while **Specificity measures its ability to correctly identify low-risk patients**. In our paper (lines 461-466), we highlighted the importance of the sensitivity, and explained why specificity of KARE is not always the best.
>
>
>
> > **[W4.2] metrics such as AUROC and AUPRC should be added.**
>
> **AUROC and AUPRC cannot be directly measured for LLM predictions** because, although LLMs compute next-token probabilities internally, these probabilities are: (1) distributed over the entire vocabulary rather than just binary classes, (2) dependent on how different LLMs encode the same label ("0"/"1") using different tokens or combinations, and (3) not directly comparable to the binary class probabilities output by ML models.
>
> Given these fundamental issues, AUROC/AUPRC cannot be accurately computed for LLMs on binary classification tasks until a theoretically well-justified calibration approach is developed. Using these metrics without proper theoretical foundations would result in unfair and potentially misleading comparisons between LLMs and traditional ML models.
>
> A recent work ClinicalBench [R4] applies a method extracting tokens from the model output logits and then applying softmax. However, their presented Table 1 is an evidence of the unreliability of such computation method: while the AUROC varies consistently with F1 for traditional ML methods, it's quite random for LLM-based methods.
>
> Therefore, **we use sensitivity and specificity which effectively evaluate performance on imbalanced datasets** using only the final predictions. Our metric choice (accuracy, macro F1, sensitivity, specificity) aligns with other recent LLM-based EHR prediction works like EHR-CoAgent [R3].
>
> ***We have included the discussion of metrics in Appendix E in the latest revision.***
>
>
>
> > **[W5] Although Amazon Bedrock provides strict compliance standards and privacy protection measures, relying on it to generate reasoning chains for distillation may limit the generalizability of this approach in real healthcare scenarios with high privacy protection requirements.**
>
> We appreciate the reviewer's thoughtful comment about privacy considerations when using Amazon Bedrock for reasoning chain generation. We would like to clarify several points:
>
>
> 1. For real-world deployment, our framework is designed to be platform-agnostic. Healthcare organizations can:
>
>    - Deploy their own local LLMs within their secure infrastructure for reasoning chain generation
>
>    - Utilize private cloud solutions that meet their specific compliance requirements
>
>    - Implement privacy-preserving APIs that sanitize or anonymize sensitive information before processing
>
> 2. The core innovation of KARE lies in its knowledge graph community retrieval and reasoning enhancement architecture, not in the specific platform used for reasoning chain generation. The principles and methodology can be implemented using any compliant infrastructure that meets an organization's privacy requirements.
>
>
> [R3] Cui, Hejie, et al. "LLMs-based Few-Shot Disease Predictions using EHR: A Novel Approach Combining Predictive Agent Reasoning and Critical Agent Instruction." arxiv 2024.03
>
> [R4] Chen et al. "ClinicalBench: Can LLMs Beat Traditional ML Models in Clinical Prediction?", arxiv 2024.11
>
> ---
>
> Again, we greatly appreciate your review and feedback. We have endeavored to address each of your concerns comprehensively. If any aspects require additional clarification or if you have further questions, we would be happy to discuss them.

---

> ### Author Response · Authors · 2024-11-24
> **Following Up: Key Points Addressed in Our Response**
>
> Dear Reviewer pYH3,
>
> Thank you again for your detailed review. As the reviewer-author discussion phase concludes shortly, we wanted to highlight our responses to your key concerns:
>
> - **Regarding novelty compared to GraphRAG**: We demonstrated through detailed comparison tables that KARE is fundamentally different, with minimal technical overlap. Additionally, recent studies [R4, R5] have shown that LLMs perform poorly in clinical prediction tasks, even after fine-tuning. Our work provides a novel solution by incorporating knowledge retrieval and reasoning in the fine-tuning process, representing a fundamental advance rather than an incremental contribution.
>
> - **Concerning high-confidence reasoning chain selection**: We acknowledge the limitations raised by your cited papers. While our experimental results show meaningful performance gains through confidence-based selection, we've proposed exploring more robust approaches for reasoning chain selection in future work.
>
> - **On metrics and evaluation**: We've clarified our metric choices (why not AUROC/AUPRC) and formulas in ***Appendix E***. The sensitivity/specificity metrics are particularly crucial for imbalanced healthcare datasets, where KARE shows significant improvements in identifying high-risk patients. We have also added MedRetriever's performance to Table 2.
>
> - **Regarding privacy and deployment**: We clarified that KARE is platform-agnostic - healthcare organizations can deploy their own local LLMs, use private cloud solutions, or implement privacy-preserving APIs.
>
>
> We hope these updates address your concerns adequately. As we approach the end of the discussion period, we welcome any additional feedback you may have.
>
> Best regards,
>
> The Authors
>
>
> ---
> [R4] Chen et al. "ClinicalBench: Can LLMs Beat Traditional ML Models in Clinical Prediction?", arxiv 2024.11
>
> [R5] Liu et al. "Large Language Models Are Poor Clinical Decision-Makers: A Comprehensive Benchmark" EMNLP 2024

---

> > ### Comment · Reviewer_pYH3 · 2024-11-25
> >
> > Thanks for the detailed reply and the additional experiments, I have revised my score.

---

> > > ### Author Response · Authors · 2024-11-25
> > > **Thank you!**
> > >
> > > We are happy that our responses addressed your concerns and deeply appreciate the improved rating! Thank you for your thoughtful feedback, which has greatly contributed to enhancing our work.

---

### Official Review · Reviewer_tZi7 · 2024-11-04

**Soundness:** 3
**Presentation:** 4
**Contribution:** 4
**Rating:** 6
**Confidence:** 4

**Summary:**

The paper presents KARE, a framework that enhances healthcare predictions by combining knowledge graph (KG) community-level retrieval with large language model (LLM) reasoning. It addresses LLM limitations like hallucinations and inadequate medical knowledge, which can affect clinical diagnosis. KARE builds a comprehensive knowledge graph from biomedical databases, clinical literature, and LLM insights, organized through hierarchical community detection and summarization to improve retrieval precision and relevance. Key innovations include: (1) a dense medical knowledge structuring approach for accurate information retrieval; (2) a dynamic retrieval mechanism that enriches patient contexts with multi-faceted insights; and (3) a reasoning-enhanced prediction framework that produces accurate and interpretable clinical predictions.

**Strengths:**

1. I really enjoyed the LLM-Driven Reasoning, producing both accurate and interpretable clinical predictions that enhance trust.
2. The model's multi-source knowledge graph ensures relevant information retrieval, addressing LLM limitations like hallucinations and sparse data.
3. KARE's retrieval mechanism enriches patient data with multi-faceted insights, enhancing EHR representation learning

**Weaknesses:**

1. Methodology Clarity: Some aspects of the methodology lack clarity, such as how the different knowledge graphs (KGs) are connected in Equation (1). Specifically, the approach for creating edges between nodes in different KGs, like G^KG and G^BC, is not well explained.

2.  The experimentation could be broadened to include more general tasks, such as diagnosis prediction or drug recommendation. Was there a reason these broader tasks were not considered?

3.  The results lack standard deviations or confidence intervals, which would help indicate the reliability of the reported performance.

4. Ablation Study Design: The ablation study could be more informative if it involved removing each feature individually, rather than adding features one at a time.
5. The provided anonymous GitHub code link cannot be opened.

**Questions:**

please refer to the weaknesses

---

> ### Author Response · Authors · 2024-11-16
> **Author Response to Reviewer tZi7 (Part I)**
>
> **Author Response to Reviewer tZi7**
>
> Thank you for recognizing the strengths of our work. We address your concerns and answer your questions below. We also uploaded a revision and used blue to mark the new changes.
>
> ---
>
> > **[W1] How the different knowledge graphs (KGs) are connected in Equation (1). Specifically, the approach for creating edges between nodes in different KGs, like G^KG and G^BC, is not well explained.**
>
> The union operation in Equation (1) is a straightforward concatenation of triple lists from the three sources  ($G^{KG}$, $G^{BC}$, and $G^{LLM}$) - we do not create additional edges between nodes from different sources. Each triple maintains its original relationships within its source. We showcase the detailed extraction process and examples for each source in Appendix B.4 (Figures 7-9) of our revision. The semantic clustering step (Section 3.1.2) later helps consolidate equivalent entities/relations across sources while preserving the original graph structures.
>
>
>
> > **[W2] The experimentation could be broadened to include more general tasks, such as diagnosis prediction or drug recommendation. Was there a reason these broader tasks were not considered?**
>
> We focused on mortality and readmission prediction primarily because the reasoning chains generated by the teacher LLM are most reliable for binary classification tasks. For multi-class tasks like length-of-stay prediction and multi-label tasks like drug recommendation, the large label space makes it challenging to generate consistent and reliable reasoning chains.
>
> Given KARE's strong performance on the current binary tasks and its task-agnostic design, we are confident it will generalize well to binary diagnosis prediction tasks, and we will include these results in our future revision.
>
>
>
> > **[W3] The results lack standard deviations or confidence intervals, which would help indicate the reliability of the reported performance.**
>
> Thank you for this valuable suggestion! We have added a comprehensive performance table with standard deviations as **Table 8 in Appendix H of our latest revision**. The small standard deviations (e.g., accuracy of 73.9±0.4% and macro F1 of 73.8±0.5% for MIMIC-IV readmission prediction) demonstrate the statistical reliability of our results. We keep Table 2 in the main text for better readability, as it already contains extensive results across 24 models, 2 datasets, and 4 metrics. For transparency, we also specify in Table 2 that ML-based methods are averaged over 30 runs, LM+ML methods over 10 runs, and LLM-based methods over 3 runs.

---

> ### Author Response · Authors · 2024-11-16
> **Author Response to Reviewer tZi7 (Part II)**
>
> > **[W4] Ablation Study Design: The ablation study could be more informative if it involved removing each feature individually, rather than adding features one at a time.**
>
> Thank you for the suggestion! We have added a new case showing the performance of having no reasoning but retrieved knowledge (the second row below). The table below shows the ablation study of retrieved knolwedge and reasoning (w/o similar patient retrieval):
>
> **MIMIC-III:**
>
> |                         |               | Mortality    |              |                 |                 | Readmission  |              |                 |                 |
> | ----------------------- | ------------- | ------------ | ------------ | --------------- | --------------- | ------------ | ------------ | --------------- | --------------- |
> | **Retrieved Knowledge** | **Reasoning** | **Accuracy** | **Macro F1** | **Sensitivity** | **Specificity** | **Accuracy** | **Macro F1** | **Sensitivity** | **Specificity** |
> | N                       | N             | 90.4         | 53.0         | 11.4            | 94.3            | 57.6         | 57.6         | 50.5            | 66.3            |
> | **Y**                   | **N**         | 93.8         | 60.9         | 21.6            | 97.2            | 67.4         | 66.5         | 68.1            | 65.0            |
> | N                       | Y             | 93.1         | 58.4         | 15.8            | 97.5            | 65.5         | 64.7         | 62.3            | 67.7            |
> | Y                       | Y             | 95.3         | 64.6         | 24.7            | 98.3            | 72.8         | 72.6         | 74.7            | 70.6            |
>
> **MIMIC-IV:**
>
> |                         |               | Mortality    |              |                 |                 | Readmission  |              |                 |                 |
> | ----------------------- | ------------- | ------------ | ------------ | --------------- | --------------- | ------------ | ------------ | --------------- | --------------- |
> | **Retrieved Knowledge** | **Reasoning** | **Accuracy** | **Macro F1** | **Sensitivity** | **Specificity** | **Accuracy** | **Macro F1** | **Sensitivity** | **Specificity** |
> | N                       | N             | 92.2         | 83.1         | 65.0            | 96.2            | 56.1         | 46.7         | 23.1            | 76.2            |
> | **Y**                   | **N**         | 93.5         | 86.8         | 70.8            | 97.6            | 66.8         | 66.6         | 73.2            | 60.9            |
> | N                       | Y             | 93.3         | 85.4         | 67.3            | 97.5            | 64.7         | 62.1         | 69.3            | 55.9            |
> | Y                       | Y             | 93.8         | 89.6         | 74.5            | 98.8            | 72.2         | 71.9         | 81.1            | 64.0            |
>
> The new results further highlight the effectiveness of the integration of clinical knowledge retrieved by our approach.
>
> As the result for the new case (the second row) is based on one-time run, we will add it later to Table 3 after we finish 3 runs with different seeds.
>
>
>
> > **[W5] The provided anonymous GitHub code link cannot be opened.**
>
> Thank you for bringing this to our attention. We've verified that the anonymous GitHub link (https://anonymous.4open.science/r/KARE-Anonymous) is functional. Alternatively, you can also access the complete codebase in our uploaded supplementary materials.
>
>
>
> ---
>
> Again, we greatly appreciate your review and feedback. We have endeavored to address each of your concerns comprehensively. If any aspects require additional clarification or if you have further questions, we would be happy to discuss them.

---

> ### Author Response · Authors · 2024-12-01
>
> Dear Reviewer tZi7,
>
> We appreciate your thorough initial review and have provided comprehensive responses to your comments in our rebuttal. With only **two days** remaining in the discussion period, would you be able to review our response? We're eager to address any outstanding concerns you might have.
>
> Best regards,
> \
> The Authors

---

### Official Review · Reviewer_83dM · 2024-11-04

**Soundness:** 2
**Presentation:** 3
**Contribution:** 2
**Rating:** 5
**Confidence:** 4

**Summary:**

The paper presents KARE, a framework designed to improve healthcare predictions by combining KG community-level retrieval with LLM reasoning. It builds a medical KG from diverse sources, organizes it into meaningful communities, and dynamically augments patient data with relevant information. Extensive tests on MIMIC-III and MIMIC-IV datasets demonstrate significant performance improvements for mortality and readmission prediction.

**Strengths:**

1. The proposed method improves the interpretability and trustworthiness of clinical predictions by generating reasoning chains.
2. The experiments are extensive. The framework is compared to many baselines and shows a sufficient performance gain in both tasks.
3. The visualizations are well-structured and the writing is easy to follow.
4. The problem setting is clearly defined.

**Weaknesses:**

1. The framework is overly complex, making it difficult to optimize and reproduce. Moreover, any noise introduced in the earlier steps (e.g., entity and relation extraction, LLMs for relationship suggestions and reasoning chains generation, etc.) could affect subsequent stages and degrade the final performance.
2. Given its complexity, the framework's efficiency should be evaluated. It involves building three different KGs for each medical concept, conducting community detection 25 times, and generating multiple summaries for each community (~60k communities in total). This process likely requires significant time and resources, making it inefficient.
3. The contribution is a bit limited compared to GraphRAG; the primary difference is mentioned in line 213 that they run GraphRAG multiple times for better diversity.
4. Acronyms like LLM, KG, EHR, and RAG are introduced multiple times throughout the paper.
5. It would be better to include case studies for the three KGs generated by the biomedical KG, biomedical corpus, and LLMs.

**Questions:**

1. In Section 3.1.1, three different KGs are generated for each medical concept, and the final KG is to integrate the three KGs together. How do you handle the conflicts among the three KGs?

---

> ### Author Response · Authors · 2024-11-16
> **Author Response to Reviewer 83dM (Part I)**
>
> **Author Response to Reviewer 83dM**
>
> Thank you for recognizing the strengths of our work. We address your concerns and answer your questions below. We also uploaded a revision and used blue to mark the new changes.
>
> ---
>
> > **[W1] The framework is overly complex, making it difficult to optimize and reproduce; Noise in early steps can affect subsequent stages and degrade the final performance.**
>
> We acknowledge the reviewer's concern about potential noise propagation through our pipeline, though we note this complexity is inherent to GraphRAG-like approaches that aim to combine graph-based knowledge retrieval with language models. Our empirical results and ablation studies demonstrate that the framework is robust to potential noise in early stages:
>
> 1. Our ablation study on knowledge sources (Figure 3, RHS) shows that even when removing entire knowledge sources, the model maintains strong performance. The relatively small performance drops (e.g., removing UMLS-derived $G_{KG}$ causes minimal degradation) suggest that noise from any single source has limited impact on final predictions.
> 2. The dynamic context augmentation with multiple selection metrics (node hits, coherence, recency, theme relevance) helps filter out noisy or irrelevant information. As shown in Figure 3 (LHS), each metric contributes to the final performance, with node hits being most critical - suggesting our framework effectively identifies and utilizes relevant knowledge while being resilient to noise.
>
> Furthermore, to enhance the reproducibility of our work, we will try to publicize our LLM training data through PhysioNet.
>
> We note that KARE pioneered an effective approach for fine-tuning LLMs on EHR-based prediction tasks, while recent studies [R1, R2] did not explore this direction. Given the critical importance of prediction accuracy in healthcare applications, we believe the complexity in one-time data preparation is justified by the significant performance gains (10.8-15.0% on MIMIC-III and 12.6-12.7% on MIMIC-IV) over leading methods.
>
> [R1] Chen et al. "ClinicalBench: Can LLMs Beat Traditional ML Models in Clinical Prediction?", arxiv 2024.11
>
> [R2] Liu et al. "Large Language Models Are Poor Clinical Decision-Makers: A Comprehensive Benchmark" EMNLP 2024
>
>
>
> > **[W2] Given its complexity, the framework's efficiency should be evaluated.**
>
> While KARE requires significant preprocessing, this is a one-time cost that we believe is justified by its superior performance in critical healthcare predictions. Specifically:
>
> KG Construction (one-time):
>
> - From biomedical KG (UMLS): 2.8 hours
> - From LLM: 4.5 hours
> - From biomedical corpus (PubMed Abstract): 8.3 hours
>   * Concept set embedding (single A6000 GPU): 0.4 hours
>   * Document retrieval (single A6000 GPU): 3.1 hours
>   * Relation extraction by LLM: 4.8 hours
>   * Total concept sets processed: 26,134
>
> Community Processing (one-time):
>
> - Conducting community detection 25 times using Leiden: 12.4 mins
>   * Graph partitioning: 1.1 mins
>   * Community organization: 11.3 mins
> - Generating 147,264 summaries from 59,832 communities: 9.6 hours
>
> Note that intensive computational requirements for KG indexing are a common challenge when working with large-scale KGs, as  evidenced by community discussions [R3].
>
> Given the critical nature of healthcare applications and KARE's significant performance improvements, we believe this one-time computational cost is completely acceptable for real-world deployment.
>
>
> [R3] (1) https://github.com/microsoft/graphrag/issues/453, (2) https://github.com/microsoft/graphrag/issues/746

---

> ### Author Response · Authors · 2024-11-16
> **Author Response to Reviewer 83dM (Part II)**
>
> > **[W3] The contribution is a bit limited compared to GraphRAG; the primary difference is mentioned in line 213 that they run GraphRAG multiple times for better diversity.**
>
> Actually, the overlap of KARE and GraphRAG exists only in ***KG Partitioning using Leiden*** (first half of Section 3.1.3). All other components are distinct:
>
> |                                   | KARE (ours)                                                  | GraphRAG (Edge et al., 2024)                                 | Key Advantages of KARE                                       |
> | --------------------------------- | ------------------------------------------------------------ | ------------------------------------------------------------ | ------------------------------------------------------------ |
> | **KG construction**    | 1. Sources: biomedical KG, corpus, and LLMs; 2. Knowledge extraction based on the medical concept co-existence in patient visits across EHR dataset | Only sourced from documents, without task-specific prior information, leading to unfocused structured knowledge | The constructed KG contains knowledge highly relevant to EHR prediction, as clinical knowledge can be found easily in sources where multiple concepts co-exist |
> | **KG Partitioning**               | Multiple runs (25 in our case)                               | Single run                                                   | A node can belong to multiple communities at the same hierarchical level, while in GraphRAG it exists in only one community. This is very important as medical concepts often co-exist with different sets of concepts in patient visits |
> | **Community Summarization** | Multiple theme-specific summaries for each community (themes: general/mortality/readmission in our case) | General summaries for communities                            | Communities can be interpreted differently for different tasks, enhancing effectiveness across multiple prediction tasks |
> | **Community Retrieval**           | Dynamic retrieval with: 1. Node hits tracking; 2. Decay factors for previously retrieved information; 3. Context coherence; 4. Temporal recency; 5. Theme relevance; 6. Iterative selection (Algorithm 1) | Parallel processing of community chunks with helpfulness scoring | 1. Dynamically avoids redundant information retrieval through hit tracking and decay;  2. Healthcare-specific metrics ensure clinical relevance |
>
> These fundamental differences and healthcare-oriented implementation contribute to KARE's significant performance improvements over existing methods, demonstrating its novel contribution to the field of clinical predictions.
>
>
>
>
>
> > **[W4] Acronyms like LLM, KG, EHR, and RAG are introduced multiple times throughout the paper.**
>
> Thank you for noting this. We have revised the paper to introduce each acronym only at its first appearance in both abstract and main text (as per standard academic practice), and removed all subsequent reintroductions in the main text.
>
>
>
>
>
> > **[W5] It would be better to include case studies for the three KGs generated by the biomedical KG, biomedical corpus, and LLMs.**
>
> We agree with this suggestion and have added examples of KG extraction from the biomedical KG, biomedical corpus, and LLMs in **Appendix B.4 (Figures 7-9)** in our latest revision.
>
>
>
> ---
>
> > **[Q1] In Section 3.1.1, three different KGs are generated for each medical concept, and the final KG is to integrate the three KGs together. How do you handle the conflicts among the three KGs?**
>
> The potential conflicts among the three KGs ($G_{KG}$, $G_{BC}$, $G_{LLM}$) are handled through semantic clustering (Section 3.1.2). By embedding all entities/relations in a shared space and applying agglomerative clustering, we merge semantically similar elements across sources. Each cluster (new entity/relation) is represented by its central element.
>
>
>
> ---
>
> Again, we greatly appreciate your review and feedback. We have endeavored to address each of your concerns comprehensively. If any aspects require additional clarification or if you have further questions, we would be happy to discuss them.

---

> ### Author Response · Authors · 2024-11-24
> **Kindly Seeking Your Thoughts on Our Response**
>
> Dear Reviewer 83dM,
>
> Thank you for your detailed review. With the reviewer-author discussion period drawing to a close, we wanted to highlight our responses to your key concerns:
>
> - **Regarding framework complexity and noise propagation**: We demonstrated through ablation studies that KARE is robust to potential noise in early stages - removing entire knowledge sources causes minimal performance degradation, and our dynamic context augmentation effectively filters irrelevant information.
>
> - **On computational efficiency**: We've provided a comprehensive breakdown of computational requirements in our response. While the one-time preprocessing involves thorough knowledge integration from multiple sources, the runtime components are highly efficient (~1s for inference per prediction). Given KARE's significant performance improvements in critical healthcare predictions, we believe this computational profile is well-suited for real-world deployment.
>
> - **Comparison with GraphRAG**: We showcased that KARE is fundamentally different from GraphRAG through detailed comparison tables highlighting the minimal technical overlap.
>
> - **Concerning KG construction**: We've added examples of KG extraction from different sources in ***Appendix B.4*** (Figures 7-9).
>
> We hope these clarifications address your concerns adequately. As we approach the end of the discussion period, we welcome any additional feedback you may have and are ready to provide further clarifications if needed.
>
> Best regards,
>
> The Authors

---

> ### Author Response · Authors · 2024-12-01
>
> Dear Reviewer 83dM,
>
> Thank you for your detailed initial review. We have carefully addressed your concerns in our rebuttal. Would you be able to review our response and provide any additional thoughts? As the discussion period closes in **two days**, we are still available to address any remaining concerns you may have.
>
> Best regards,
> \
> The Authors

---

### Author Response · Authors · 2024-11-22
**Response to All Reviewers**

Dear Reviewers,

We sincerely appreciate your thorough feedback and constructive suggestions. Based on your comments, we have made comprehensive revisions to our paper, with all changes highlighted in blue.

First, we thank the reviewers for recognizing the key strengths of our work:

* The framework effectively addresses critical challenges in healthcare predictions (`All Reviewers`)
* The experimental evaluation is comprehensive and thorough (Reviewers `83dM`, `pYH3`, `thia`, `oKgq`)
* The LLM-driven reasoning significantly enhances both accuracy and interpretability (Reviewers `83dM`, `pYH3`, `tZi7`, `oKgq`)
* The integration of knowledge graphs with LLM reasoning presents a novel and promising approach (Reviewers `tZi7`, `oKgq`)
* The visualization and presentation demonstrate clarity and strong structure (Reviewers `83dM`, `pYH3`, `oKgq`)

The revised paper incorporates all reviewers' suggestions, including enhanced analysis explanations, new experimental results, and additional experimental details.

Key updates in our revision include:

* Integration of additional related works as suggested by reviewers (`pYH3`, `thia`, `oKgq`)

* Analysis of metric selection rationale (excluding AUROC/AUPRC) and detailed definitions of Sensitivity and Specificity in **Appendix E** (`pYH3`, `thia`)

* Addition of MedRetriever performance results in Table 2 (`pYH3`, `thia`)

* Inclusion of KG construction case studies from different sources in **Appendix B.4** (Figures 7, 8, and 9) (`83dM`)

* Performance with standard deviation in **Appendix H** (`tZi7`)

* Human evaluation results of KARE-generated reasoning chains in **Appendix I** (`thia`, `oKgq`)

* Comparative analysis of model parameters and training time in **Appendix J** (`oKgq`)



**We have addressed each reviewer's concerns and questions thoroughly in our detailed individual responses**.

As the reviewer-author discussion phase concludes shortly, we welcome your review of our responses and any additional feedback for improvement. We greatly appreciate your continued participation in this discussion.

---

### Meta-Review · Area_Chair_Ankd · 2024-12-20

**Metareview:**

This paper introduces KARE, a novel framework designed to enhance clinical decision-making by combining knowledge graph (KG) community retrieval with reasoning capabilities of large language models (LLMs). Traditional retrieval-augmented generation (RAG) models often retrieve sparse or irrelevant data, hindering healthcare predictions. KARE overcomes these limitations by structuring a multi-source KG from biomedical databases, clinical literature, and LLM-generated insights, then leveraging hierarchical graph community detection to retrieve precise and contextually relevant information.

Key Contributions:
1. Dense Medical Knowledge Structuring: Enables accurate retrieval of context-specific medical data.
2. Dynamic Knowledge Retrieval: Enriches patient-specific contexts with detailed and relevant insights.
3. Reasoning-Enhanced Prediction Framework: Combines enriched contexts with LLM reasoning to deliver interpretable and precise clinical predictions.

Results:
- Outperforms leading models in mortality and readmission predictions on the MIMIC-III (by 10.8-15.0%) and MIMIC-IV (by 12.6-12.7%) datasets.
- Demonstrates improved prediction accuracy and trustworthiness due to LLM-driven reasoning.

Innovations:
- Hierarchical KG community organization for effective retrieval.
- Multi-faceted insights addressing LLM hallucination and sparse retrieval issues.

Additional Notes:
The paper includes extensive experiments, human evaluations, and detailed responses to reviewer concerns. Discussions highlight the importance of choosing appropriate metrics, such as sensitivity and specificity, over AUROC/AUPRC for LLM-based predictions. Despite some critical reviews, KARE is presented as a significant step forward in leveraging AI for clinical applications.

The paper introduces KARE, a framework that combines knowledge graph (KG) retrieval and large language model (LLM) reasoning to enhance healthcare predictions. KARE addresses LLM limitations, such as hallucinations and insufficient medical knowledge, by integrating structured data from biomedical sources, clinical literature, and LLM insights into a unified KG. The KG is organized through hierarchical community detection and summarization, enabling precise and contextually relevant retrieval for improved predictions.

Key Features:
1. Dense Medical Knowledge Structuring: Ensures accurate and relevant information retrieval by embedding entities and relations into a shared semantic space.
2. Dynamic Context Augmentation: Enhances patient-specific data with multi-source insights, improving electronic health record (EHR) representation learning.
3. LLM-Driven Reasoning Framework: Produces accurate, interpretable predictions to boost trust in clinical applications.

Innovations:
- Integration of multi-source medical knowledge with KG community detection.
- A dynamic retrieval mechanism enriching patient data with multi-faceted insights.
- An interpretable, reasoning-based prediction framework for critical tasks like mortality and readmission prediction.

**Additional Comments On Reviewer Discussion:**

1. Conflict Handling in KG Integration
2. Methodology Clarifications
3. Computational Efficiency
4. Experimentation Scope
5. Ablation Study Improvements
6. Statistical Reliability
7. Code Accessibility

The authors seek additional feedback as the discussion period concludes, ensuring remaining concerns are addressed. The revisions reflect significant enhancements to methodology clarity, result reliability, and ablation study design.

---

### Decision · Program_Chairs · 2025-01-22

Accept (Poster)